# Whole-body physics simulation of fruit fly locomotion

Roman Vaxenburg[1], Igor Siwanowicz[1], Josh Merel[2], Alice A. Robie[1], Carmen Morrow[1], Guido Novati[3], Zinovia Stefanidi[1,4], Gert-Jan Both[1], Gwyneth M. Card[1,5], Michael B. Reiser[1], Matthew M. Botvinick[3,6], Kristin M. Branson[1], Yuval Tassa[3 ✉] & Srinivas C. Turaga[1 ✉]

The body of an animal influences how its nervous system generates behaviour[1]. Accurately modelling the neural control of sensorimotor behaviour requires an anatomically detailed biomechanical representation of the body. Here we introduce a whole-body model of the fruit fly *Drosophila melanogaster* in a physics simulator[2]. Designed as a general-purpose framework, our model enables the simulation of diverse fly behaviours, including both terrestrial and aerial locomotion. We validate its versatility by replicating realistic walking and flight behaviours. To support these behaviours, we develop phenomenological models for fluid and adhesion forces. Using data-driven, end-to-end reinforcement learning[3,4], we train neural network controllers capable of generating naturalistic locomotion[5–7] along complex trajectories in response to high-level steering commands. Furthermore, we show the use of visual sensors and hierarchical motor control[8], training a high-level controller to reuse a pretrained low-level flight controller to perform visually guided flight tasks. Our model serves as an open-source platform for studying the neural control of sensorimotor behaviour in an embodied context.

Animal behaviour emerges from sensorimotor feedback loops that integrate signals from the brain, body and environment[1,8]. The body determines how neural motor commands translate into movement and how sensory feedback is generated in response. Therefore, a detailed biomechanical understanding of the body is crucial for modelling the neural control of movement. Here we introduce a physics-based simulation framework for an anatomically detailed model of the fruit fly *D. melanogaster*, designed to support the modelling of diverse sensorimotor behaviours. We validate our model by demonstrating realistic locomotion—both walking and flight—using reinforcement learning (RL). This general-purpose simulation provides a platform for future studies of brain–body interactions across a broad range of fruit fly behaviours.

Our work follows previous physics-based models of the worm[9], hydra[10], rodent[11] and fruit fly[12–16]. The Grand Unified Fly[13] pioneered sensorimotor closed-loop visually guided flight using a simplified body model and hand-designed controller. More recent work has revealed the basis of muscle actuation of the wing hinge[16]. In parallel, NeuroMechFly[14,15] introduced an anatomically detailed fruit fly model capable of walking and grooming, pairing a heuristically designed low-level walking controller with a learnt high-level controller to generate sensory-guided behaviours[15].

Our work unifies flight and walking in a single physics-based model, enhancing realism in body mechanics, physics interactions and control. We developed an anatomically detailed fly body model in the open-source MuJoCo[2] physics engine, incorporating high-resolution imaging to reconstruct a female *Drosophila* (Fig. 1). To accurately simulate both flight and walking, we introduced a computationally efficient phenomenological fluid dynamics model to approximate aerodynamic forces from wing flapping and adhesion actuators to model foot–surface interactions.

Using high-speed kinematic tracking[17–19], we trained closed-loop RL controllers capable of replicating naturalistic fly movements. These controllers, trained for both flight (Fig. 2) and walking (Fig. 3), operate using only high-level steering commands. Finally, we demonstrate the reuse of a pretrained low-level flight controller for vision-guided flight tasks (Fig. 4). Through inverse kinematics, we further show that our model supports a broad behavioural repertoire beyond locomotion, including grooming.

## Body model geometry

We used confocal fluorescence microscopy to capture high-resolution images of the entire adult female fly body (Fig. 1a and Methods; see also the supplementary datasets available at Figshare (ref. 20)). Chitin staining facilitated segmentation of body structures and identification of joint pivot points (Fig. 1b). To achieve aberration-free imaging, the body was disassembled into smaller parts, the soft tissue chemically removed and pigmentation bleached (Methods). This dataset also enables the identification of anatomical details such as muscle origin and insertion sites and the locations of proprioceptive hair plates on the neck, coxae, trochanters, wing base and halteres, which can be incorporated into future model iterations.

We manually segmented 67 body components using Fiji[21], then simplified their meshes in Blender[22] by reducing the vertex count to

[1]HHMI Janelia Research Campus, Ashburn, VA, USA. [2]Fauna Robotics, New York City, NY, USA. [3]Google DeepMind, London, UK. [4]Machine Learning in Science, Tübingen University and Tübingen AI Center, Tübingen, Germany. [5]Columbia University, New York City, NY, USA. [6]Gatsby Computational Neuroscience Unit, University College London, London, UK. ✉e-mail: tassa@google.com; turagas@janelia.hhmi.org

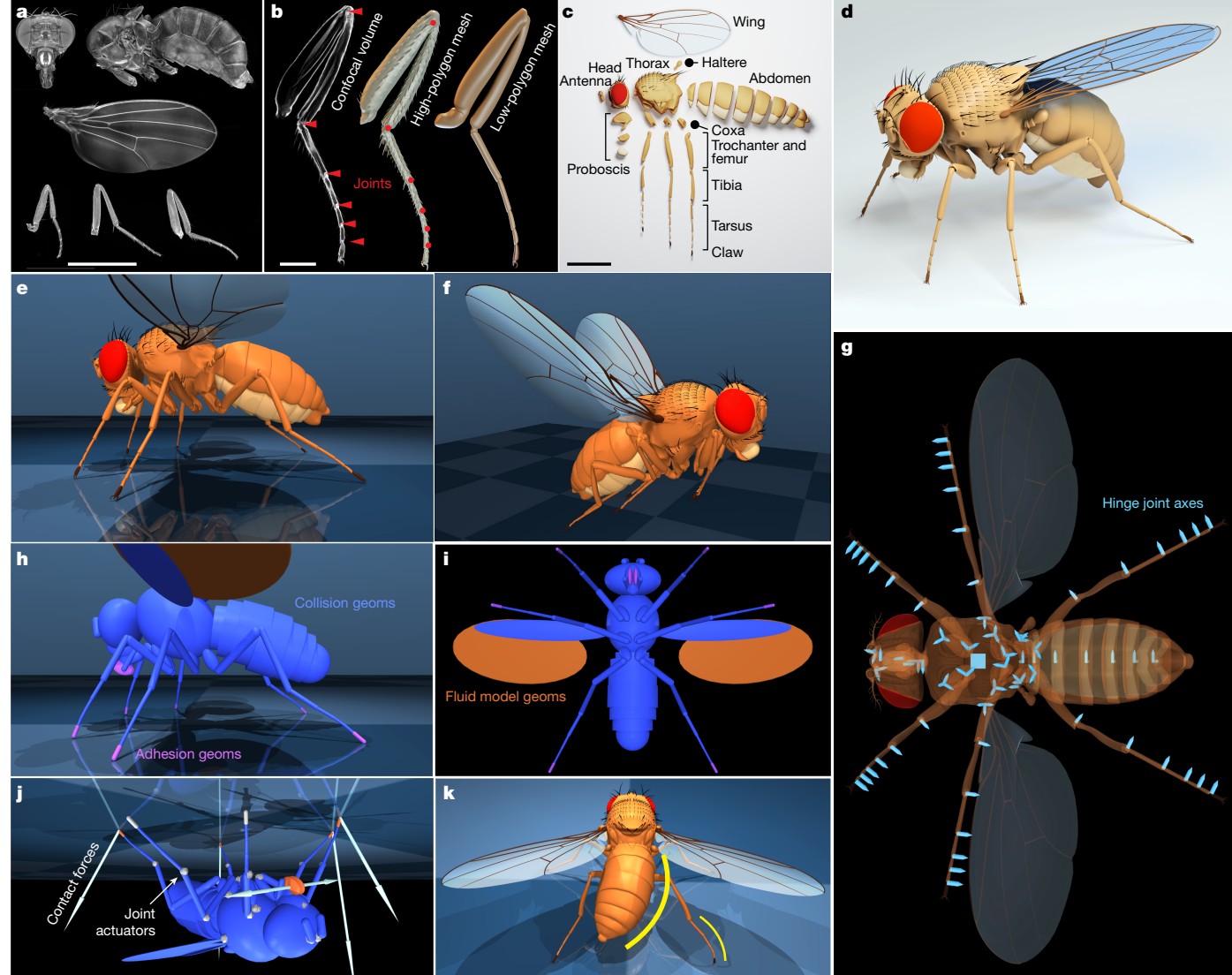

**Fig. 1 | Constructing the female fruit fly body model. a**, Compilation of six datasets representing a single fly. Maximum intensity projections of confocal stacks showing head, thorax with abdomen, wings and legs. Scale bar, 1 mm. **b**, Left, a partial projection of the midleg confocal volume with the joints between the femur, tibia and tarsal segments indicated. Middle, a 3D mesh extracted from the volume. Right, a low-polygon leg model. Scale bar, 0.2 mm. **c**, An exploded low-polygon fly model (around 20,000 faces) showing all body segments. Scale bar, 1 mm. **d**, The geometric fly model assembled in Blender. **e**, The complete physics fly model in MuJoCo simulator in the default rest pose. **f**, Fly model in a flight pose with retracted legs. **g**, DoFs. Translucent bottom view with light-blue arrows indicating hinge joint axes pointing in the direction of positive rotation. Groups of three hinge joints effectively form ball joints. Cube: 6-DoF free joint required for free CoM motion in the simulator and is not a part of fly's internal DoFs. **h,i**, Side view (**h**) and bottom view (**i**) of the geometric primitive (geom) approximation of body segments used for efficient collision detection and physics simulation. Blue, collision detection geoms; purple, geoms that have associated adhesion actuators; orange, wing ellipsoid geoms for simulating flight with the advanced fluid force model. **j**, Visualization of actuator forces generated when the model fly hangs upside down. The adhesion actuators of the front-right, middle-left and hind-right legs are actively gripping the ceiling (orange); the labrum (mouth) adhesors are also active; other actuators are inactive (white). The arrows visualize net contact forces proportional and opposite to the applied adhesion forces. **k**, Exaggerated posture showing the coordinated activation of the abdominal abduction and tarsal flexion actuators. Abdominal joints and tarsal joints (yellow) are each coupled with a single tendon actuator that simultaneously actuates multiple DoFs.

enable efficient computational modelling while preserving key morphological features (Fig. 1b,c). In Blender, we assembled the components into a full-body model and defined the kinematic tree by linking them at 66 identified joint locations (Fig. 1b,d and Extended Data Fig. 1), yielding 102 degrees of freedom (DoFs) in accordance with the literature (Fig. 1g). Biological joints were modelled as single hinge joints (1 DoF) or as combinations of two or three hinge joints (2 and 3 DoFs, respectively). These joint models are simplified approximations, particularly for complex articulations such as the neck joint, wing hinge and thorax–coxa articulation[16,23,24]. Joint angles corresponding to the resting pose (Fig. 1d,e) and the flight pose (Fig. 1f) were estimated through visual inspection of videography.

## Modelling body physics

The Blender model of the fly's body geometry was imported into the MuJoCo physics engine through a multi-step process (Supplementary Information). First, we generated primitive 'geom' representations of each body part (Fig. 1h,i) to enable efficient physics simulation and collision detection. Second, we measured the mass of each body

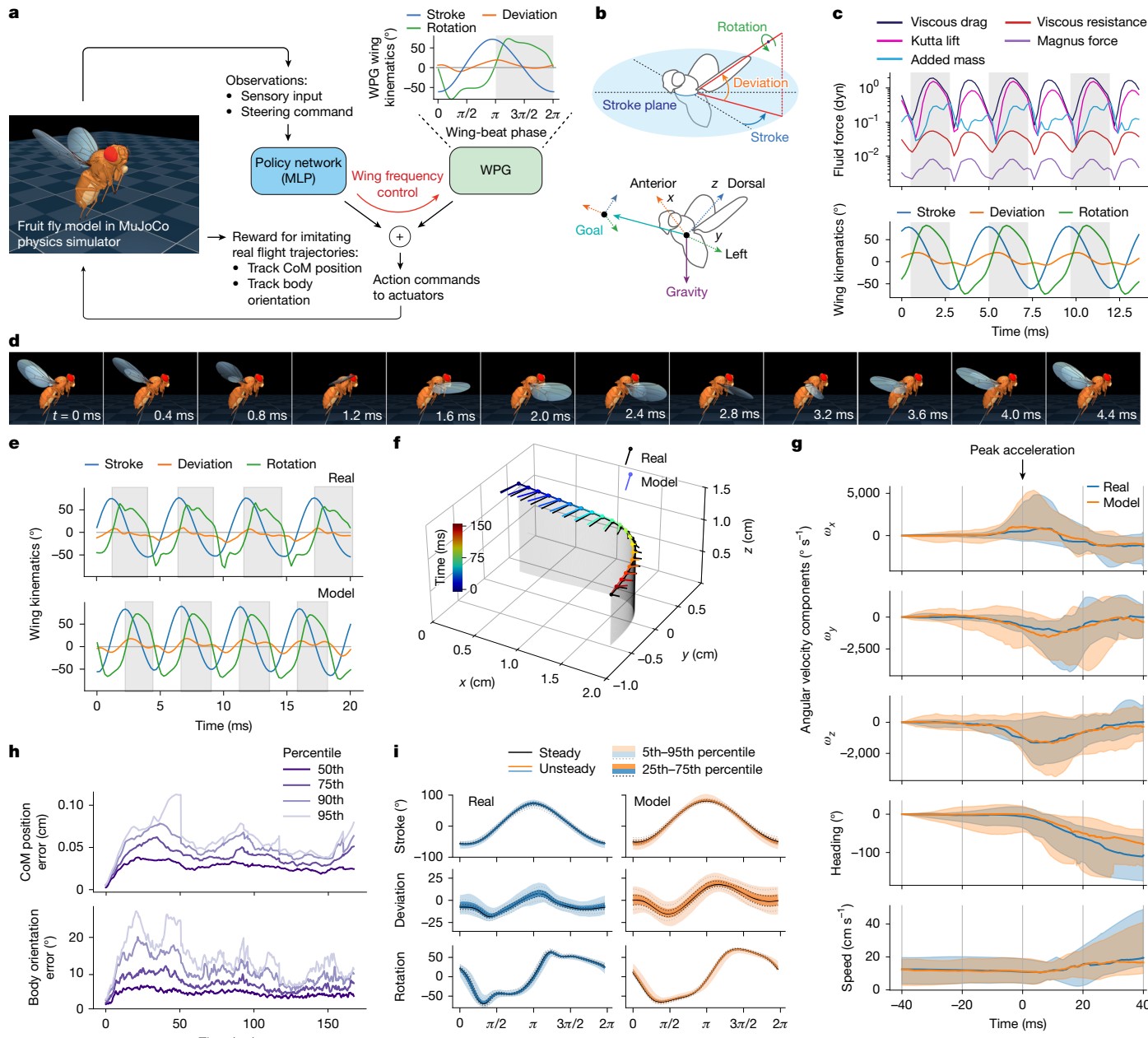

**Fig. 2 | Flight imitation. a**, Overview of RL set-up. A single policy network is trained to imitate the CoM position and body orientation across a dataset of 216 trajectories of freely flying *Drosophila* (around 43 s in total). The flight controller consists of a trainable MLP and a WPG. The motor command is the sum of the MLP and WPG outputs. Top right, one period of the fixed baseline wing-beat pattern produced by the WPG. Grey stripe indicates wing downstroke. **b**, Top, wing coordinate system and wing angle definition. Bottom, body coordinate system and example model sensory inputs: the direction to the goal CoM position and the gravity direction. **c**, Fluid model forces exerted on the left wing, and the corresponding wing kinematics, during a stable horizontal flight at 30 cm s⁻¹. **d**, Filmstrip of the model flying straight at 30 cm s⁻¹ during one full wing-beat cycle. **e**, Wing kinematics during a saccade manoeuvre produced by the model and real fly. **f**, Wings produce body movements through

a phenomenologically modelled fluid. The real (black) and model (coloured) fly body pose while traversing a test trajectory. Circles, heads; lines, tails. **g**, Median and percentiles of body angular velocity, heading and speed for real and model flies during test saccades. The trajectories are aligned to peak acceleration at $t = 0$. Roll and pitch angular velocities ($\omega_x$ and $\omega_y$) are similarly important in model flies' and real flies' turns. A small divergence between model and real occurs after the saccade. Solid lines, medians; shading, 25th–75th percentiles. **h**, Percentiles of errors between the model and the corresponding real fly's body CoM, and orientation for 56 test trajectories. **i**, Wing angles during steady (small body acceleration) and unsteady (large body acceleration) wing beats for model and real flies in the test set. Large body accelerations are achieved by similarly small alterations to the median wing-beat pattern.

part (Methods and Supplementary Table 1). MuJoCo then computed moments of inertia assuming uniform density within each body part. Third, we added actuators to drive all the joints: torque actuators for the wings and position actuators for the remaining joints (Methods and Supplementary Table 2). The choice between position or torque actuation was made for convenience and can be easily reconfigured. Position

actuators, in particular, can facilitate faster training in deep RL[25]. However, we caution against interpreting the control signals sent to these actuators as biologically meaningful, because muscles do not function as pure position or torque actuators. Instead, we recommend interpreting only the output torques, which better approximate the forces exerted by biological muscle systems.

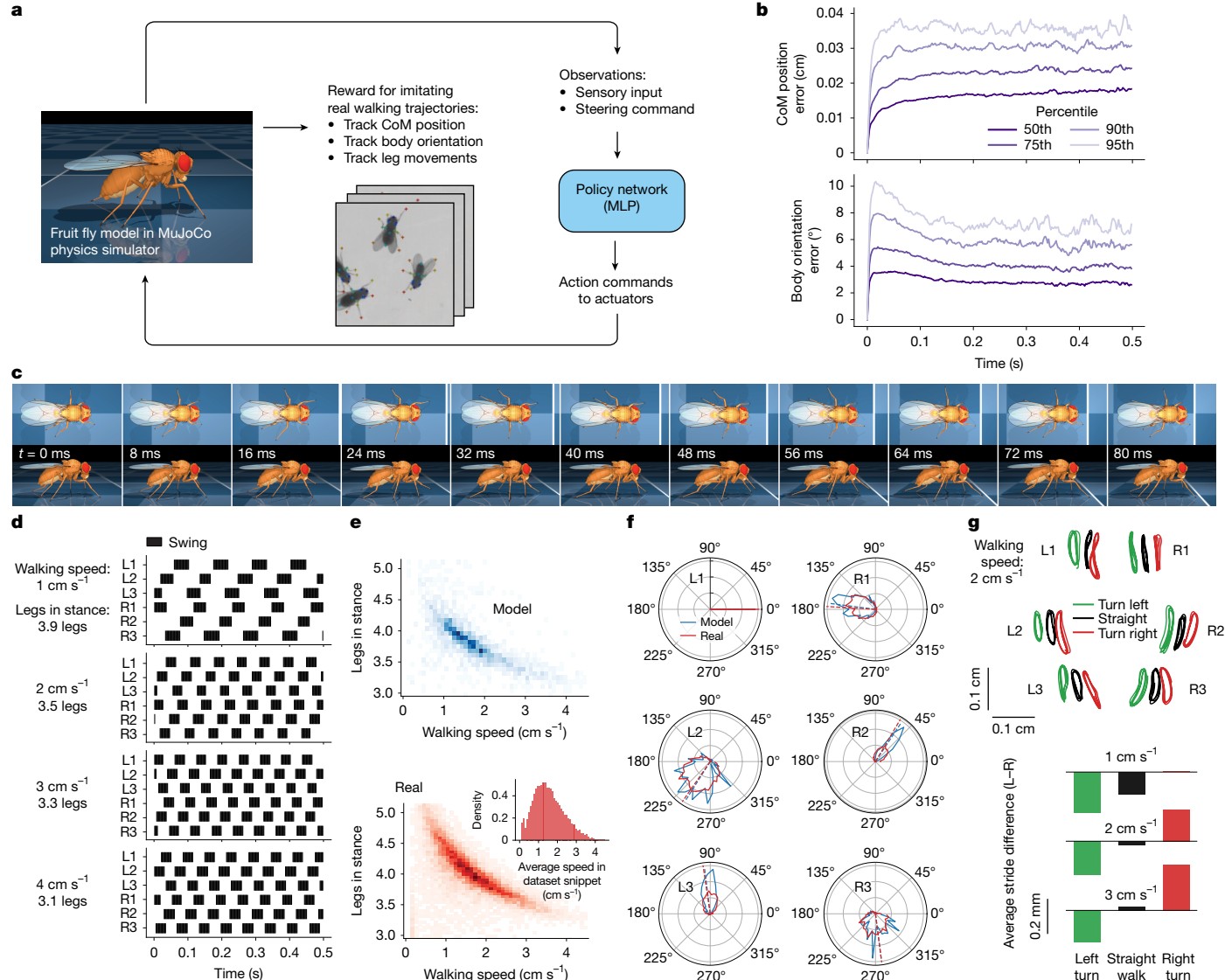

**Fig. 3 | Walking imitation. a**, Overview of RL set-up. A single policy network is trained to imitate a dataset of 13,000 snippets (around 64 min in total) of freely walking real *Drosophila*. Full body movements are imitated, including tracking CoM position, body orientation and detailed leg movements. **b**, Percentiles of errors between the model and the corresponding real fly's body CoM, and orientation for 3,200 test walking trajectories. **c**, Filmstrip of the model walking straight at 2 cm s⁻¹ during one full leg cycle, with 8-ms steps between frames. **d**, Gait diagrams of the fly model tracking synthetic fixed-speed straight-walking trajectories at four speeds. Black stripes indicate the swing phase of leg motion. For each speed, the average number of legs simultaneously in stance position (on the ground) is indicated. **e**, Number of legs simultaneously in stance position averaged over walking snippet versus average walking speed in snippet.

Top, model tracking test set trajectories. Bottom, entire walking dataset. Inset, the distribution of average walking speeds per snippet in the dataset. **f**, Distributions of swing onset phases of all legs relative to the front left leg L1 in walking trajectories with a mean speed in the range [1.2, 1.7] cm s⁻¹. Blue, fly model tracking test set trajectories; red, entire walking dataset. Dashed lines indicate circular medians. **g**, Learnt turning strategy. Top, *xy* projection of leg-tip trajectories in egocentric reference frame for model walking straight (black), turning left (green) and turning right (red), at a constant speed (2 cm s⁻¹). Leg-tip trajectories are shifted horizontally for clarity. Bottom, difference between (egocentric) left and right leg-tip swing length, averaged over all legs, at various walking speeds.

Joint limits were determined using inverse kinematics to match a range of observed poses from videography, including grooming postures that demonstrate the fly's remarkable flexibility (Methods and Extended Data Fig. 2). Typically, each actuator controlled a single DoF. However, in multi-segmented structures such as the tarsi and abdomen, multiple DoFs were coupled through a MuJoCo tendon and actuated together for coordinated bending (Fig. 1k). To model the adhesive properties of insect tarsi, which allow flies to walk on walls and ceilings[26], we introduced adhesion actuators in MuJoCo. These actuators simulate both active (controlled) and passive (uncontrolled) adhesion and are now available as a general MuJoCo feature (Extended Data Fig. 3a). Besides adding

adhesion actuators to the tarsal tips (Fig. 1h,j), we also added them to the labrum (mouth) to enable modelling of feeding and courtship behaviours[27].

Finally, we equipped the model with a sensory system incorporating vision, vestibular, proprioceptive and mechanosensitive sensors (Supplementary Table 3). Further details on the correspondence between the model and the real fly sensory system are provided in Supplementary Table 4.

The resulting model is a fully functional, biomechanical simulation of the entire fly body. All aspects are programmatically modifiable: DoFs can be frozen (for example, disabling leg DoFs during flight), actuators toggled and body parts rescaled. Sensors can also be customized,

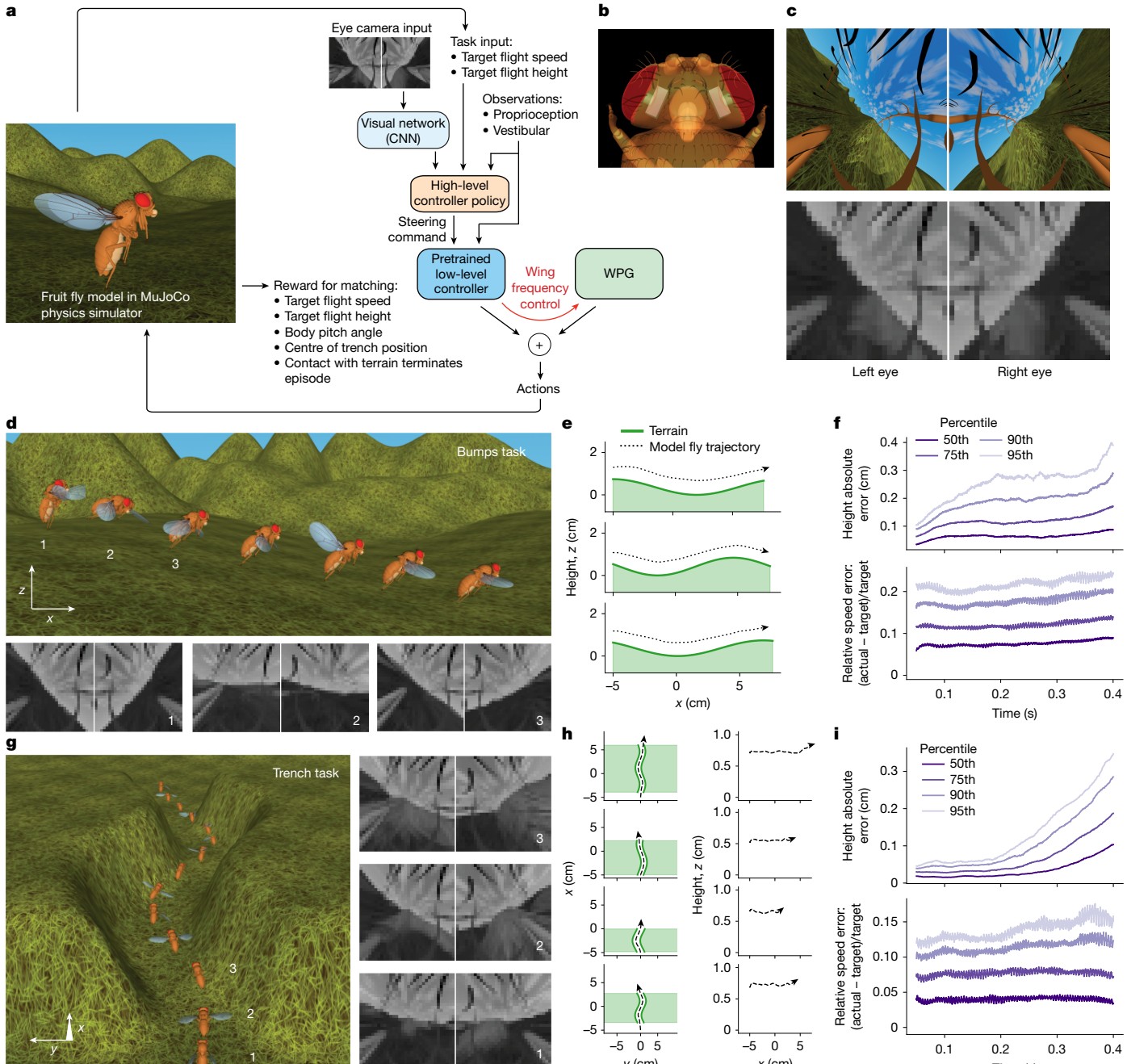

**Fig. 4 | Hierarchical controller reuse for vision-guided flight: altitude control (bumps) and obstacle avoidance (trench) tasks. a**, Overview of RL set-up. The policy uses vision to carry out flight at a given target speed and height while avoiding collision with terrain. In the bumps task, the terrain is a sequence of sine-like bumps across the flight path. The fly model must constantly adjust the altitude to maintain a constant target height above the bumpy terrain. In the trench task, the terrain is a narrow sine-shaped trench, requiring the fly to manoeuvre left and right. The terrain, target speed and height are randomly changed at each training episode. As in the flight imitation task, the flight controller combines policy network and WPG. The policy network consists of a CNN to process visual input from eye cameras; a high-level 'navigator' controller network; and reuses a low-level flight controller pretrained with the flight imitation task in Fig. 2. The high-level controller and the CNN are trained end-to-end with RL. The weights of the pretrained low-level controller are kept unchanged. **b**, Translucent top view of the head of the fly model, showing the placement of MuJoCo eye cameras. **c**, Top, high-resolution eye camera view for fly in **a**. Bottom, corresponding downsampled greyscale frames used as visual input. **d**, Top, time-lapse of flight produced by trained bumps-task policy. Bottom, example visual input frames captured by eye cameras. **e**, Side view of representative fly model trajectories in the bumps task. **f**, Percentiles of height and speed errors for 1,000 test bumps episodes. **g**, Left, time-lapse of flight produced by trained trench-task policy. Right, example visual inputs. **h**, Top-down view of representative fly model trajectories (left) and their corresponding flight height (right) in the trench task. **i**, Percentiles of height and speed errors for 1,000 test trench episodes.

including their activation and temporal filtering properties. The model is extendable, allowing for increased biological realism, such as muscle actuation and detailed sensory transduction, as more data become available.

## Modelling body–environment interactions

Beyond simulating the fly's body physics, the MuJoCo engine also models its interactions with the environment, including forces from

physical contacts, fluid interactions (air) and gravity. Contacts are simulated both between different fly body parts and between the fly and its surroundings, which is crucial for determining ground reaction forces during walking (Fig. 1j and Extended Data Fig. 3a).

Fluid interactions mainly account for forces generated by wing movement through air. Accurately simulating fluid dynamics is computationally demanding, so we developed a phenomenological, stateless, quasi-steady-state approximation (Supplementary Information). Our model extends a previous approach[28] to three dimensions, estimating the forces and torques on ellipsoid-shaped bodies moving through an incompressible quiescent fluid. It approximates five fluid dynamics phenomena: added mass[29,30], viscous drag[31], viscous resistance[32], Magnus lift[33] and Kutta lift[34]. The resulting forces and torques are polynomial functions of fluid parameters (density and viscosity) and ellipsoid parameters (shape, size and linear and angular velocities), with a slender ellipsoid approximation for the wings (Fig. 1i). This fluid model is a MuJoCo feature designed for our study, but can be applied to other models.

To accurately simulate flight, our phenomenological model must approximate the total aerodynamic forces acting on real wings, regardless of the underlying mechanism. This includes contributions from unmodelled phenomena, such as passive forces arising from wing flexibility[35]—which we do not explicitly simulate owing to our rigid-body approach—as well as complex fluid interactions such as turbulence and vortices. To compensate for these omissions, we optimized the coefficients of the fluid force components until stable hovering was achieved (Methods).

In our model, Kutta lift (generated by fluid circulation around the wing) and viscous drag (opposing wing movement through the fluid) were the dominant forces (Fig. 2c). Notably, these coefficients did not require fine-tuning, as flight performance remained stable even with 20% coefficient variation (Extended Data Fig. 4). However, our quasi-steady-state approximation captures only the time-averaged effects of turbulence and other transient fluid phenomena, without explicitly modelling their dynamic contributions.

Simulating behaviour through numerical integration of the fly's passive and active dynamics, along with its environmental interactions, presents a considerable computational challenge. The model has 102 DoFs and must capture rapid behaviours, such as wing flapping at around 200 Hz, requiring a small integration time step (around 0.1 ms). On a single core of an Intel Xeon E5-2697 v3 CPU, simulating 10 ms of flight and walking (Figs. 2a and 3a) took 421.5 ms and 68.65 ms, respectively—fast enough for RL of motor control. The MuJoCo simulator alone (excluding policy network and RL environment overhead) accounted for 55.5 ms and 23.2 ms for flight and walking, respectively (Supplementary Information and Supplementary Table 5).

## Imitation learning of locomotion

We used deep RL to train our fruit fly model to generate realistic locomotor behaviours. An artificial neural network served as the sensorimotor controller, processing sensory input and generating motor control signals in a closed loop. At each time step, MuJoCo simulated sensory signals, which were fed into the neural network. The network then computed actuator control signals, which MuJoCo used to simulate the resulting forces on the body.

To generate realistic locomotion, we used imitation learning[3,4], a data-driven approach that trains neural controllers to replicate observed behaviours. Specifically, we trained the network to match the trajectories of the real fly's centre of mass (CoM) and body segments during free locomotion, as measured from video. To ensure generalization beyond the training trajectories, we developed steerable low-level controllers[8]. These networks, although not intended as exact models of the ventral nerve cord, perform an analogous role by converting high-level descending commands into fine-grained motor control

signals. In our model, the high-level commands specify the desired change in the fly's CoM position and orientation (6 DoFs) at each time step, whereas the low-level motor control signals drive the actuators.

We trained two steerable neural network controllers—one for flight (Fig. 2) and one for walking (Fig. 3). Using video-derived trajectories of real flies (Methods), we set the CoM trajectory as the high-level steering command and designed a reward function that incentivized the model to match both the CoM trajectory and the body part positions at each time step. The reward was maximized when the model tracked the real fly's CoM while replicating its limb and wing movements, enabling the emergence of naturalistic locomotion patterns.

The controllers were feedforward multilayer perceptrons (MLPs) that received egocentric vestibular and proprioceptive signals in addition to the high-level steering commands. They were trained using Distributional Maximum a posteriori Policy Optimization (DMPO)[36,37], an off-policy, model-free RL algorithm with a distributional critic[38]. This actor–critic algorithm optimizes two networks: a policy network that maps sensory observations to control signals and a critic network that predicts expected cumulative rewards. We used the DMPO implementation from the acme RL library[39]. Training required approximately $10^9$ simulation steps (walking) and $10^8$ steps (flight), with $10^8$ and $10^7$ policy network updates, respectively. To reduce training time from weeks to days or hours, we developed a multi-CPU and GPU parallelization scheme[40] using Ray[41], a general-purpose distributed computing framework (Methods).

## Flight

We trained a steerable flight controller (Fig. 2a) using imitation learning on high-speed videography data of freely flying *Drosophila hydei* performing spontaneous saccades[6] and forced evasion manoeuvres[5]. These datasets contained 272 individual flight trajectories (around 53 s in total) that captured the CoM and wing kinematics during various manoeuvres, including turns, speed and altitude changes, sideways and backward flight and hovering (Methods). Although *D. hydei* is larger than *D. melanogaster*, their body and wing kinematics are expected to be similar[42], and this dataset represents the best available source of free-flight data. A single controller network was trained to imitate all 216 trajectories from the training set, enabling stable flight and generalization to new trajectories (Fig. 2).

Our flight controller design was inspired by the observation that real flies control their flight mainly through small deviations from a nominal wing-beat pattern[5,6]. Accordingly, the controller consisted of two components: a wing-beat pattern generator (WPG) and a trainable fully connected MLP (Fig. 2a). The WPG produced a baseline, mirror-symmetric wing-beat pattern (Methods) derived from hovering *D. melanogaster* wing kinematics[13,43] (Fig. 2a). The policy network controlled both the base frequency of the WPG and small deviations from its baseline, allowing the model to reproduce the full range of flight behaviours. Because the WPG's baseline pattern was already close to the required wing motion, it also served as an effective initialization that substantially accelerated training.

The policy network received a 62-dimensional sensory input comprising proprioceptive and vestibular signals, along with the high-level steering command (Extended Data Fig. 5 and Supplementary Table 6). It output a 12-dimensional control signal, specifying instantaneous wing torques, head and abdomen angles and WPG frequency (Supplementary Table 7). To speed up training (Extended Data Fig. 6), the legs were retracted to their typical flight position (Fig. 1f) and their DoFs were frozen. Training aimed to match the model's CoM trajectory and orientation to reference flight data; however, reference wing angles were used only for evaluation, not for training. Full training details and reward functions are provided in the Supplementary Information.

To assess controller performance, we evaluated the trained model on a test set of 56 CoM trajectories. The model fly accurately tracked

the target CoM trajectory, with a median position error of 0.25 mm and a median orientation error of less than 5° (Fig. 2h), as illustrated in an example trajectory (Fig. 2f). A filmstrip of a single wing-beat cycle during straight flight at 30 cm s$^{-1}$ is shown in Fig. 2d.

The model fly was trained to match the target CoM trajectories of real flies, but its wing kinematics were only weakly constrained (to approximate the baseline WPG by DMPO action penalty; Methods). This set-up enabled us to compare its wing trajectories with those of real flies to evaluate the accuracy of the physics simulation and behavioural realism. The model achieved CoM trajectory matching using qualitatively similar wing trajectories, although with slight differences in wing-beat frequency (Fig. 2e). Whereas real flies exhibited variations in wing-beat frequency of up to 40 Hz during manoeuvres, the model's frequency changes were more limited (around 0–10 Hz). Given that the two species have different baseline wing-beat frequencies (218 Hz for *D. melanogaster* and 192 Hz for *D. hydei*), we did not attempt a direct quantitative comparison.

Like real flies, the model relied on small left–right wing-stroke asymmetries to generate large accelerations during saccades (Fig. 2i). By analysing flight trajectories involving both steady (low-acceleration) and unsteady (high-acceleration) flight, we confirmed that minimal differences in wing stroke were sufficient to generate large changes in CoM trajectory, consistent with previous observations[5,6,13,43] (Fig. 2g,i). The model also replicated key features of real fly turning manoeuvres, including characteristic changes in median angular velocity, heading and speed[6,42].

Finally, we examined how the phenomenological fluid model generated forces to support flight (Fig. 2c). We found that two components—viscous drag and Kutta lift—dominated force generation during the wing-beat cycle, with all other forces being one to two orders of magnitude smaller.

## Walking

We trained a steerable closed-loop walking controller (Fig. 3a) using imitation learning. High-speed (150 fps) top-view videography captured groups of freely walking female fruit flies in a circular arena[44]. Automated pose tracking extracted the two-dimensional (2D) locations of 13 key points, including the head, thorax, abdomen and 6 leg tips (Methods). Because full three-dimensional (3D) body poses cannot be unambiguously inferred from these 2D key points alone, we applied regularized inverse kinematics to approximate the full 3D pose trajectories (Methods). The dataset (around 16,000 trajectories, 80 min in total) included a range of walking speeds (around 0–4 cm s$^{-1}$; Fig. 3e inset), turning and standing still.

Unlike flight, in which all manoeuvres are generated by small deviations from a common baseline wing-beat pattern[5,42], flies exhibit diverse gait patterns depending on walking speed[7], varying limb coordination and ground contact. This gait variability precluded the use of a simple pattern generator. Instead, we trained a fully connected MLP controller (Fig. 3a) without predefined structure. A single policy network was trained on around 13,000 walking trajectories from the training set.

Walking requires controlling considerably more DoFs than flight does (59 DoFs versus 12), encompassing leg movements (including adhesion), abdomen and head. Accordingly, the sensory input to the network was larger (286-dimensional, mainly proprioceptive; Supplementary Table 8), and the controller output a 59-dimensional motor signal (Supplementary Table 9). Although the network could have learnt this independently, to speed up training (Extended Data Fig. 6), the wings were folded and their actuation disabled to reduce complexity. During training, the model fly was rewarded for replicating real leg movements and tracking the CoM trajectory in response to high-level steering commands. Because we lacked direct leg adhesion measurements, adhesion was not explicitly included in the reward function, but the model learnt to activate adhesion naturally when legs

were in stance phase; that is, on the ground (Methods and Extended Data Fig. 3). Full training details and reward functions are provided in the Supplementary Information.

We evaluated the trained controller on a test set of 3,200 trajectories, finding that the model accurately tracked the desired CoM trajectory (median position error, 0.4 cm; median orientation error, 4°; Fig. 3b). A single walking cycle at 2 cm s$^{-1}$ is shown in Fig. 3c. Examining stance and swing phase durations at different speeds (Fig. 4d), we found that, as in real flies, the model fly kept at least three legs in stance at any time, with more legs in stance at slower speeds[7]. At 4 cm s$^{-1}$, an average of 3.1 legs were in stance, increasing to 3.9 at 1 cm s$^{-1}$. Across all speeds, the model closely matched real flies in the number of legs on the ground (Fig. 3e).

We assessed leg coordination by computing phase delays between each leg's swing onset relative to the left foreleg, L1, and found good agreement with real flies (Fig. 3f). When commanded to turn at speeds of 1–3 cm s$^{-1}$ with a 1-cm turning radius (Fig. 3g), the model decreased stride length on the turning side while increasing stride length on the opposite side, consistent with real fly behaviour[7]. However, unlike real flies, the model exhibited an asymmetric forelimb modulation, adjusting the front-leg stride length more during left turns than right turns.

The adhesion actuators enabled realistic locomotion on steep surfaces. To test this, we trained the model to traverse hilly terrain with varying slopes—an environment designed to be impossible to navigate without adhesion (Methods and Extended Data Fig. 3). The model learnt to adjust adhesion forces dynamically according to terrain steepness. Within MuJoCo's Coulomb friction model, adhesion forces act normal to the surface, pushing the fly legs towards the surface and increasing friction resistance to slip. The model applied stronger adhesion with the forelegs and midlegs on upward slopes and relied on the hind legs to prevent slipping on downward slopes. For further details, see Methods and Extended Data Fig. 3.

## Hierarchical vision-guided flight

Fruit flies are highly visual insects, with large compound eyes and optic lobes comprising about a third of their brain. To reflect this, we incorporated visual sensors into our model in addition to proprioceptive and vestibular sensors. The eyes were modelled using MuJoCo camera sensors (Fig. 4b), rendering a 32 × 32-pixel grid with a 150° field of view. This resolution approximates *Drosophila* vision, with an inter-ommatidial angle of 4.6° (ref. 45 and Fig. 4c). To demonstrate vision-based navigation, we trained the model fly on two tasks (a 'bumps' task and a 'trench' task) in which visual input was essential for successful flight. Figure 4c illustrates an example of low-resolution visual input alongside a high-resolution counterpart rendered (for visualization only) during flight.

We reused the general-purpose steerable low-level flight policy from the flight imitation task (Fig. 2) as part of a hierarchical vision-guided flight controller, trained by end-to-end RL (Fig. 4a). The controller consists of a fixed pretrained low-level policy (including the WPG) that directly controls wing motion and a high-level navigator policy that issues low-dimensional steering commands. The high-level controller received a 62-dimensional proprioceptive and vestibular sensory signal, along with a low-dimensional visual feature representation extracted by a convolutional network (CNN) from the 6,144-dimensional visual input. In addition, it received a 2D task-specific input: target flight height and speed (Supplementary Table 10). The low-level controller received the 62-dimensional proprioceptive and vestibular signal, plus the high-level steering command, but not the visual or task-specific inputs. As in the flight imitation task, the low-level controller produced a 12-dimensional control output, specifying wing torques, head and abdomen angles and WPG frequency (Supplementary Table 11).

To preserve learnt flight dynamics, the low-level controller's weights were frozen while the CNN and high-level MLP were jointly trained to maximize task reward. In both tasks, terrain conditions, as well as

target height and speed, were randomized in each training and test episode (Supplementary Table 12). Contact with terrain resulted in early episode termination (failure). The model fly started from a speed of zero, requiring it to accelerate to the target speed at the beginning of each trial. Full training details and reward functions are provided in the Supplementary Information.

### Bumps task

Fruit flies use visual cues to estimate altitude, which allows them to maintain a stable height over uneven terrain[46]. To model this visually guided altitude control, we created a virtual world with a randomly generated sinusoidal terrain profile. The model fly was rewarded for flying straight at a constant target velocity while maintaining a stable altitude above the ground (Fig. 4d). After training, it successfully learnt to use visual input to regulate altitude, achieving a median height error of 0.045 cm and a median speed error of 2.2 cm s$^{-1}$ after the initial acceleration phase (Fig. 4e,f).

### Trench task

In a second task, we trained the model fly to navigate a narrow trench without colliding with its walls. The virtual trench had a sinusoidal curving profile with a fixed width and depth (Fig. 4g). The model fly was rewarded for maintaining a constant forward speed and altitude, whereas collisions resulted in early episode termination and loss of future rewards. Successful navigation required the fly to use vision to detect and avoid the trench walls. After training, the model fly reliably navigated the entire trench while maintaining the target height and speed, with a median height error of 0.032 cm and a median speed error of 0.16 cm s$^{-1}$ after the initial acceleration phase (Fig. 4h,i).

## Discussion

Animal behaviour emerges from the interplay between the nervous system, body and environment. Here, we have demonstrated realistic locomotion—both walking and flight—using an anatomically detailed whole-body model of the fruit fly. This advance was made possible by improved physics simulation of body–environment interactions, and by deep RL, which approximated the nervous system through an artificial neural network trained to mimic real fly behaviour. Our model consists of 67 rigid-body components with 102 DoFs, actuated through torques at the joints. Using the MuJoCo physics engine, we simulated rigid-body collisions and fluid interactions with air. Deep RL and imitation learning were then used to train a closed-loop neural controller that generates realistic body movements for both walking and flight across arbitrary trajectories. All components—body model, physics simulation and pretrained controllers—are released as open-source software.

This work integrates measurements across spatial and temporal scales, combining microscopy of static anatomy with high-speed videography of dynamic locomotion. Our model simulates both the forces generated by the body and the sensory information available to it through idealized actuators and sensors. We see this open-source platform as a foundation for further refinement. Imaging techniques such as confocal microscopy (Extended Data Fig. 1), micro-computed tomography[14] and synchrotron X-ray holographic nano-tomography[47] can be used to provide whole-body musculoskeletal measurements that can inform anatomically detailed muscle actuation models, including for the neck[24], wing hinge[16] and coxa[47,48] joints. On the sensory side, our idealized sensors could be enhanced using mappings of proprioceptive organs in the leg and wing[47,48], and new eye maps could refine the spatial positioning of individual ommatidia[45]. In addition, model-based pose-tracking algorithms could extract more precise kinematics from high-speed videography[49,50].

Accurately incorporating muscle actuation across the whole body will require substantial effort. First, all muscles and their insertion sites must be identified to determine their respective DoFs. Second,

each muscle must be modelled in the physics simulator, approximating complex muscle and tendon wrapping to account for the limited capabilities of existing high-performance physics engines[2,51]. This step requires experimentation to determine the best trade-offs between anatomical accuracy and computational efficiency. In highly complex regions, such as the wing hinge and neck joints, for which full anatomical fidelity is impractical, virtual muscles might provide a more feasible approach by mapping muscle activations to joint torques. Third, system identification is needed to constrain muscle dynamics, ideally using correlated muscle activity and kinematic measurements[16]. As a first step, inverse dynamics and imitation learning could estimate muscle parameters, as was done for the non-muscle actuators in this work. Although recent work on the wing hinge[16] provides a roadmap for implementing virtual muscles, scaling this approach to the entire body remains difficult. Similar challenges are involved in accurately modelling proprioception across the body.

Future work can integrate connectomic maps of the entire fruit fly nervous system[52–58] to better model the neural circuits that underlie sensorimotor behaviour. Our model predicts sensory inputs and motor outputs on a moment-by-moment basis, which can be integrated with connectomic data detailing individual sensory and motor neuron mappings at the resolution of individual DoFs[55]. Recent work[59] has shown that connectome-constrained networks, combined with characterizations of their input–output functions, enable predictions of neural activity at single-neuron resolution. Using imitation learning, our model can be combined with connectomic and behavioural data to investigate neural mechanisms that underlie sensory–motor behaviours such as escape responses to looming stimuli[60], gaze stabilization[61] and ventral nerve cord control of locomotion.

In the long term, combining our whole-body model with a complete nervous system connectome, comprehensive behavioural measurements and connectome-constrained deep neural network modelling[59,62] could enable the development of whole-animal models of the entire body and nervous system of the adult fruit fly.

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

## Methods

### Preparation of anatomical samples

The five-to-six-day-old flies ($w^{1118}$;+;+, backcrossed to M. Heisenberg's CantonS for ten generations) were anaesthetized on ice, briefly washed with ethanol and dissected under PBS-T (PBS + 0.1% Triton X-100). Disassembling the fly into manageable elements allowed us to use high-magnification, high-numerical aperture (NA) objectives that have—in relation to the size of a fly's body—short working distances, but have the benefit of higher axial resolution than the lower-magnification and lower-NA objectives. Heads, wings, thoraces with abdomens, forelegs, midlegs and hind legs were transferred to individual tubes. All body parts except the wings were incubated with 0.25 mg ml$^{-1}$ trypsin in PBS-T for 48 h at 37 °C to remove the soft tissues. The cuticle was then bleached in 20% $H_2O_2$ for 24 h, and the exoskeleton and tendons were stained overnight with Congo Red (0.5 mg ml$^{-1}$; Sigma-Aldrich, C676-25G), a bright and comparatively photostable chitin-binding dye that stains both soft, membranous and hard, sclerotized cuticle. It also shows affinity to tendons and fine tendrils, which is very convenient for identifying muscles' origins and insertion sites, even in the absence of soft tissues. The dataset also enables the identification of locations of the proprioceptive hair plates of the neck, coxae, trochanters, wing base and halteres—information that can be incorporated into future versions of the model. The samples were dehydrated in ethanol and mounted in methyl salicylate (Sigma-Aldrich, M6752), which has a refractive index very close to that of glass, facilitating imaging throughout the relatively thick and bulky samples without degradation of the signal. Serial optical sections were obtained on a Zeiss 880 confocal microscope at 2 μm with a Plan-Apochromat 10×/0.45 NA objective, 1-μm intervals with a LD-LCI 25×/0.8 NA objective or 0.3 μm with a Plan-Apochromat 40×/1.3 NA objective. The 560-nm laser line was used to excite Congo Red.

### Blender model of body geometry

Three-dimensional meshes were extracted from the confocal stacks using Fiji's 3D viewer plug-in[21] and imported into Blender[22]. A 3D model was constructed from meshes representing the head, thorax and abdomen, wing and foreleg, midleg and hind leg of a single female fly. Appendage meshes were mirrored across the body's medial plane (Extended Data Fig. 1a). This model was used as the reference for creating a simplified lower-polygon-count model, in which the total number of vertices was reduced from 22.6 million to 20,000 (Extended Data Fig. 1b). This simplified model consisted of 67 articulated body segments (Extended Data Fig. 1d): 9 body axis segments (head, thorax and 7 abdominal segments), proboscis (4 segments), antennae, wings, halteres (6 segments in total) and legs (coxa, femur, tibia, 4 tarsal segments and tarsal claws; 6 × 8 segments). The exact positions of joints, articulations and axes of joints' rotation were determined with high confidence from confocal microscopy data (Fig. 1b and Extended Data Fig. 1c). The model was posed in the rest position and rigged in Blender by creating constraints defining movement of the body segments with respect to each other. Each of the 67 body segments was assigned (parented to) a control element called 'bone', forming a hierarchical kinematic tree system resembling a skeleton called 'armature' (Extended Data Fig. 1d).

### MuJoCo model of body physics

The Blender model was then exported to MuJoCo using the dm_control exporter (https://github.com/google-deepmind/dm_control). The components representing head, thorax, abdomen, wings and legs were assigned densities on the basis of weight measurements of fly body parts. The masses of body parts were obtained from 52 female flies weighed in bulk in two batches of 22 and 30 to minimize the measurement error. The flies were weighed with the Meter Toledo XS104 analytical balance with a readability of 0.1 mg and a linear deviation of 0.2 mg. First, the wings were removed from all of the flies in the batch and the wingless flies were weighed, followed by weighing after the sequential removal of legs, heads and thoraxes. The values were subtracted from the whole-body weight. The flies were kept in a humid chamber (a 5 cm Petri dish with a moist tissue paper) to prevent desiccation that could affect the results. The measured masses were: head, 0.15 mg; thorax, 0.34 mg; abdomen, 0.38 mg; legs (each), 0.0162 mg; wings (each), 0.008 mg. This corresponds to a total fly mass of 0.983 mg (Supplementary Table 1). The full body length of the model is 0.297 cm, and the wing-span is 0.604 cm.

Joint limits were at first determined using Blender's inverse kinematics tool. We started with fairly tight joint limits and then used reference images of extreme articulated postures (mostly from grooming behaviours) to increase joint limits as required, until all reference poses could be achieved. We then refined the leg joint limits using automated inverse kinematics fitting of the model to 392 frames from manually annotated grooming behaviour videos (more details below). The sensory-system details in the model's default configuration are shown in Supplementary Table 3. DoFs were actuated using torque or position actuators, with certain DoFs (abdomen and tarsi) coupled by tendons (Supplementary Table 2). For position actuators, control ranges were set to be equal to the corresponding joint ranges. For more details on building the fly MuJoCo model, see the Supplementary Information.

### Analysis of leg DoFs

To verify our approximation of the leg DoFs and leg joint ranges, we applied the following procedure. We recorded two-camera videos[63] of several free *Drosophila* individuals during grooming behaviour. We then uniformly sampled and annotated individual frames of the fly postures during grooming, giving us 3D coordinates of five key points for each leg: the four leg joints (body–coxa, coxa–femur, femur–tibia, tibia–tarsus) plus the tarsal tip. We annotated all six legs per frame regardless of which legs were actively involved in grooming in the frame. This provided us with data for legs both in grooming positions and in rest (standing) positions. We only observed grooming with T1 and T3 legs and we collected a total of 392 frame annotations. Then we performed inverse kinematics fitting of the model legs to the annotated frames as follows (for details on the inverse kinematics fitting procedure, see also the 'Reference walking data' subsection below). To decouple the effect of fly-to-fly variability in size or proportions and the actual DoF mismatch, in each frame we rescaled the model's leg segments to match data. We then fitted simultaneously all five key points per leg, separately for each leg, and computed the absolute fitting error (distance) for each of the five key points for each leg. Extended Data Figure 2 shows the distributions of the inverse kinematics fitting errors for each key point and each leg: Extended Data Fig. 2a shows the errors for leg fits in rest position, and Extended Data Fig. 2b shows errors in grooming positions. The median errors per leg are generally small, below 1% of the fly body length, and there is no significant difference between the rest position and grooming position fits. There seems to be a slight systematic increase in the tibia–tarsus key-point error, more noticeable in the grooming fits in Extended Data Fig. 2b, which is not surprising because grooming leg positions tend to be more intricate than the rest position. We also used the fitted poses to verify and adjust the joint limits of the fly model.

### Distributed RL

For each locomotion task, we trained a policy network using a distributed RL set-up[40,64] powered by Ray, an open-source general-purpose distributed computing package[41]. The distributed training configuration is shown in Extended Data Fig. 7. Multiple CPU-based actors run in parallel in separate MuJoCo environment instances, generate experiences and log them into a replay buffer. A single GPU-based learner samples training batches from the replay buffer and updates the policy and critic network weights. The critic network is a part of the training process only and is not used by the fly model directly. Each actor explores the environment and generates experiences using its

own copy of the policy network whose weights are periodically synchronized with the current learner policy. For learner policy updates, we used the off-policy actor–critic DMPO agent, a distributional extension[38] of the MPO agent[36,37]. We used dm_control[65] to set up the RL environments and for MuJoCo Python bindings. We used the DMPO agent implemented in acme[39] and the replay buffer implemented in reverb[66], and the Adam optimizer[67]. The hyperparameters of the distributed set-up and of the DMPO agent are shown in Supplementary Tables 13 and 14.

To guarantee stability, we ran MuJoCo physics simulations at time steps four to ten times smaller than those used to sample action commands from the policies[65] (see the physics and control time-step values in Supplementary Tables 15 and 16). The policies were stochastic during training (outputting distribution over actions) to facilitate exploration. The distributions were Gaussian, independent for each action dimension and parameterized by mean and standard deviation. The policy network architectures for each task are provided in Supplementary Tables 17–19. At test time, the policies were reverted to be deterministic by using the means of the predicted action distributions. The actions output by the policy networks were in the canonical range [−1,1]. The actions were then rescaled to match their corresponding proper ranges in the fly model; for example, joint limits for position actuators or force limits for force actuators. All observables (policy inputs) were strictly egocentric; that is, calculated in the local reference frame of the fly model. We only used feedforward policy and critic networks and did not extensively sweep network architectures.

We trained the model in episodes of a finite number of time steps. An episode ends either (i) when the episode time limit is successfully reached or (ii) when an early termination condition, indicating failure, is met. Details of the termination conditions are provided in the corresponding RL task sections in the Supplementary Information. In the first case, the agent estimates the remaining infinite-horizon future rewards (beyond the episode's final step) by bootstrapping from the value of the state at the end of the episode. In the second case, the reward sequence is truncated by setting the future reward to zero. The loss of the future infinite-horizon rewards is an unfavourable outcome and the agent will try to learn to avoid events that trigger early episode termination.

## Modelling flight behaviour

**Flight physics parameters.** We used the following procedure to fit the flight physics parameters. We started with a wing motion trajectory recorded previously from a hovering *D. melanogaster*[13] (https://github.com/willdickson/fmech). We placed the model in a hovering position and actuated the wings to reproduce the real wing trajectory by using the real wing angles as target angles for the wing actuators. We then iteratively adjusted (increased) the wing actuator gain to a point at which the mean absolute error between the reference wing angles and the trajectory traversed by the model's wings was below 5% of the wing angle amplitude. At each iteration, we also fitted the wing joint damping coefficient to avoid underdamping and ensuing wing oscillations. We used the same gain value for all three wing actuators (yaw, roll and pitch). Our final values for the gain and damping pair were gainprm = [18, 18, 18], damping = 0.007769.

Having found suitable wing actuator gain and wing joint damping, we adjusted the MuJoCo fluid model coefficients (Supplementary Information and Supplementary Table 20), which scale the drag and lift forces produced by the flapping wings. These (dimensionless) fluid model coefficients are stored in the fluidcoef MuJoCo attribute. We placed the model in a flight position and again drove the wings with the real reference angles as target angles for the wing actuators. We then iteratively found a set of fluid parameters such that the net lift approximately balanced the fly model weight during several wing-beat cycles, fluidcoef = [1.0, 0.5, 1.5, 1.7, 1.0]. The flight physics parameters are summarized in Supplementary Table 21.

We also performed a sensitivity analysis on the viscous drag and Kutta lift, the two dominant forces in the fluid dynamics. We retrained imitation learning of free flight with modified choices for the coefficients associated with viscous drag and Kutta lift, with all other coefficients held fixed. We then evaluated the degree to which imitation learning was able to correctly reproduce ground truth CoM flight trajectories with realistic wing kinematics, as in the original experiments reported in the paper. We also quantified the fraction of trajectories in which the fly crashed to the ground as a second performance measure. Extended Data Figure 4 shows that flight performance is robust to even 20% variation in these parameters.

**WPG.** All our flight tasks use a WPG that produces, in an open-loop manner, a fixed mirror-symmetric cyclic baseline wing trajectory. The WPG generates the baseline pattern by design, with no learning involved. The baseline trajectory closely follows a previously recorded wing pattern of a hovering *D. melanogaster*[13]. The baseline pattern is available in the Figshare supplementary datasets[20]. At each simulation time step, the WPG retrieves and outputs the six wing angles (three per wing; Fig. 2) of the baseline wing pattern for the current wing-beat cycle step. These baseline wing angles get converted to torque action commands for the wing actuators. Although it already produces a realistic-looking wing motion, the fixed baseline wing trajectory alone is not sufficient to support a stable hover, owing to the lack of a feedback loop, approximations in the MuJoCo fluid model and the sim-to-real gap. It is the role of the policy network to provide these missing components. To achieve this, the WPG torque action is combined (additively) with the policy output to produce the final action vector to be sent to the wing actuators. In this way, the policy modulates the fixed baseline pattern and produces flight required by the task at hand; for example, stabilize flight, hover, speed up, turn and so on.

The wing motion produced by the combination of the WPG and the trained policy stays close to the initial WPG baseline pattern. This is achieved by penalizing the magnitude of the policy actions during training with the DMPO agent[36,37]. In this way, the agent is encouraged to discover a physically viable wing motion pattern by using only minimal policy actions without deviating substantially from the baseline pattern. In addition, the WPG can vary the frequency of the output baseline pattern within a predefined range. A single scalar out of the policy action vector is used by the WPG to control the baseline wing-beat frequency. In our setting, the wing-beat frequency was allowed to vary within a 10% range centred at 218 Hz, the *D. melanogaster* average frequency[43]. The WPG is implemented as a lookup table containing the single fixed baseline wing pattern resampled at different wing-beat frequencies within the 10% frequency range. When a frequency change is requested by the policy, the WPG will smoothly connect the wing patterns at the old and new frequencies.

**Reference flight data.** As the flight reference data, we used previously recorded trajectories of freely flying *D. hydei*. The trajectories contain a fly's Cartesian CoM position and body orientation represented as a quaternion. The trajectories were recorded at 7,500 fps. We started with 44 trajectories of spontaneous turns (saccades)[6] and 92 trajectories of evasion manoeuvres[5] in response to visual looming stimuli. Each reference trajectory started with the fly first flying normally and then performing a manoeuvre. During and after the manoeuvre, the fly could fly straight, sideways and backwards. The flies could also ascend and descend. We linearly interpolated the raw trajectories to the flight simulation control step of 0.2 ms. Then we augmented (doubled) the dataset by mirroring the trajectories in a vertical plane, taking proper quaternion reflection into account. This resulted in a dataset of 272 flight trajectories, equivalent to around 53 s of real-time flight. The dataset is available at Figshare (ref. 20). We used 80% of the trajectories for training and the rest for testing. Owing to the small size of the dataset, to maintain balance between left and right turns in the training data,

we split the dataset such that if a trajectory was in the training set, so was its mirrored counterpart. We simulated flight at 0.05-ms physics time steps and 0.2-ms control time steps (Supplementary Table 15).

## Modelling walking behaviour

**Reference walking data.** We obtained single-camera top-down-view videos of several freely behaving *Drosophila* individuals with 2D key-point tracking[44] (Extended Data Fig. 8a). In brief, groups of ten walking flies (*w+;BPp65ADZp (attP40); BPZpGDBD (attP2), 20XUAS-GtACR1-EYFP* (*attP2*); refs. 68,69) were recorded at 150 fps in a shallow, flat-bottomed, 50-mm-diameter arena. The 2D positions of 17 key points were predicted with the Animal Part Tracker (APT)[70]. From nine such videos, we prepared the walking reference dataset. We used 13 of the 17 key points: 3 on the head, 3 on the thorax, one at the tip of abdomen and the 6 leg tips, as shown in Extended Data Fig. 8. We selected female flies and isolated walking trajectory segments based on the following criteria. At each frame, we required: (i) the distance to the other flies in the arena is larger than one body length; and (ii) the velocity component parallel to the fly body is larger than perpendicular to the body. Then we required (iii) a snippet duration of at least 20 frames (133 ms; roughly corresponds to one fly step); and (iv) a ratio of mean leg-tip speed to mean CoM speed smaller than 1.5. This produced a set of trajectory snippets with flies walking at different speeds, turning and standing still. The average walking speeds per snippet are distributed approximately in the range [0, 4] cm s$^{-1}$ (Fig. 3e, inset). We then linearly interpolated the walking snippets from 6.7-ms time steps to the walking simulation control of 2-ms time steps.

The 13 key points tracked in 2 dimensions, however, are not sufficient for the RL task reward calculation (Supplementary Information), which requires complete specification of the model's pose, position and orientation. Specifically, this full-body representation should include all joint angles, joint axis orientations, joint velocities, body position and orientation and leg-tip positions. Obtaining the full-body data for the model from the experimental data required, first, lifting the 2D walking snippets by complementing the horizontal $x$, $y$ coordinates with the third vertical $z$ dimension. Based on a separate side-view video of a walking fly[71], and our fly model's default standing position, we approximated the body height and pitch angle during walking by a single fixed value. From this video, we also estimated the amplitudes of the arcs traversed by leg tips during swing motion. The amplitudes were $A = 0.086$, 0.047 and 0.051 cm for the T1, T2 and T3 legs, respectively. We approximated the $z$ coordinate of the leg-tip swing arcs by the sine function as $z = A\sin(x)$, with $x$ going from 0 to $\pi$ for each single leg swing. Using the 2D coordinates of the leg-tip key points, we separated leg swings from stances based on leg-tip horizontal velocities in the fly's egocentric reference frame. Then we added the approximate sine-arcs to the swing segments of the leg-tip trajectories, keeping $z = 0$ for stances. This procedure produced 3D coordinates for the 13 key points in the walking snippets selected earlier.

As a final step, we computed the full-body reference poses for each frame in all snippets. We added 13 key-point sites to the fly model and performed inverse kinematics fitting of the whole model body to the fly poses in the 3D walking snippets. For each snippet, we rescaled the reference key points to match the size of the fly model. In each frame, we simultaneously fit all 13 key points by minimizing the following objective with respect to the model joint angles, $\mathbf{q} = (q_1, q_2,\ldots)$:

$$\min_{\mathbf{q}}\left[\sum_{i=1}^{13} \|\mathbf{s}_i(\mathbf{q}) - \mathbf{s}_i^\star\|^2 + \lambda \, \|\mathbf{q} - \mathbf{q}_0\|^2\right],$$

where $\mathbf{s}_i(\mathbf{q})$ and $\mathbf{s}_i^\star$ are the 3D Cartesian coordinates of the 13 key points of the model and the fitting target pose, respectively. We used gradient descent to minimize this objective. To use the time continuity across frames, we used the final pose fitted for the previous frame to initialize the fitting procedure for each subsequent frame. Because the 13 key

points (only 6 of which specify the leg tips) do not fully define the leg postures in space, we added a small regularization term to encourage fitting poses that are closer to the default standing pose of the fly model. The default pose is specified by a vector of model joint angles, $\mathbf{q}_0$, a vector of zeros in our case. The regularization strength is $\lambda = 1 \times 10^{-4}$ cm$^2$ rad$^{-2}$. Having found the joint angles $\mathbf{q}$ for the reference poses in each frame, we also computed joint velocities, d$\mathbf{q}$/d$t$, using finite differences.

This procedure resulted in a complete full-body representation (joint angles, positions, orientations and velocities) of the reference walking trajectories for the walking imitation task. In total, the walking dataset comprises around 16,000 walking snippets, amounting to around 80 min of fly walking behaviour. The dataset is available at Figshare (ref. 20).

**Adhesion, friction and contact forces.** The fly model's ability to attach to and walk on inclined surfaces is enabled by the combination of adhesion, friction and contact forces. In this section we describe the details of the adhesion mechanism in MuJoCo and how the fly model uses the leg adhesion actuators. Let us consider a simple example of the stationary fly on an inclined plane. When a contact is detected between a tarsal claw collision geom and the floor surface, MuJoCo computes the (constraint) contact force, which is the ground reaction force in this case (Extended Data Fig. 3a). Within the Coulomb friction model, as long as the contact force vector $\mathbf{f}_{\text{contact}}$ is within the friction cone boundaries, the tangential component of the net external force ($\mathbf{f}_{\text{weight}}$, the fraction of the total fly weight supported by the given leg in this simplified example), which acts to produce slipping motion, will be balanced by the tangential component $\mathbf{f}_{\text{contact}}^{\parallel}$ of the contact force in the opposite direction. The friction cone (our fly model uses elliptic friction cones; MuJoCo also supports pyramidal friction cones) includes all contact force vectors satisfying $\mathbf{f}_{\text{contact}}^{\parallel} \leq \mu\mathbf{f}_{\text{contact}}^{\perp}$, where $\mu$ is the static friction coefficient. Outside of the friction cone, that is, for contact forces beyond the threshold $\mu\mathbf{f}_{\text{contact}}^{\perp}$, slipping motion will occur. Note that within this friction model, the cone angle is a function of the friction coefficient alone and is given by $\theta = \tan^{-1}\mu$. In our model, $\mu = 1$ and $\theta = 45°$.

In MuJoCo, the action of an adhesion actuator is equivalent to injecting force in the normal contact direction (Extended Data Fig. 3a), effectively acting to push the fly's claw into the floor. In response, the normal component $\mathbf{f}_{\text{contact}}^{\perp}$ of the contact force will increase by the same amount. Although there is no change in the tangential contact force component owing to the adhesion, note how the net contact force vector is now further away from the friction cone boundary, thus providing a larger slip-resisting margin, which can be used for forward walking propulsion, for example. Beyond its role in the Coulomb friction mechanism, the adhesion force can also directly counteract gravity to enable walking on arbitrarily oriented surfaces, such as vertical walls or ceilings.

Extended Data Figure 3b shows the time-lapse of the walking task in which the fly model was trained to use adhesion to overcome bumpy terrain. This task is similar to the walking imitation task described above (Fig. 3). The fly is required to imitate a single real-data walking snippet of walking straight at a fixed speed, 2.7 cm s$^{-1}$. In the fly's way, we introduced a sine-like bump obstacle that cannot be overcome without adhesion. The bump obstacle is procedurally regenerated at each training episode with the bump's height and length varying in the ranges [0, 2] and [2, 4] cm, respectively. Thus, the bump inclination angle was between 0° and 72°. We also added a small action penalty, epsilon_penalty = $3 \times 10^{-4}$, through the DMPO agent mechanism, to encourage the agent to prefer economic actions, including the adhesion action. We recorded the adhesion action and contact forces during a trained policy rollout on a bump with a maximum inclination angle of around 45°, as shown in Extended Data Fig. 3c–f.

The adhesion forces produced by the leg adhesion actuators while overcoming the obstacle are shown in Extended Data Fig. 3c. The model's use of adhesion increases as the terrain angle becomes steeper. The fly model learnt to use mostly the T1 and T2 leg pairs on the way uphill, and mostly T3 on the way downhill. Owing to the lack of constraints, there is an asymmetry (degeneracy) between the left and right leg adhesion use, which we did not attempt to resolve. In our model, the largest adhesion force per leg is one fly body weight, which is also shown in the figure for comparison. The norm of the corresponding contact force vectors $|\mathbf{f}_{contact}|$ for each leg is shown in Extended Data Fig. 3d. The effect of creating a larger slip-resisting margin—moving $\mathbf{f}_{contact}$ further away from the friction cone boundary—with increasing adhesion is shown in Extended Data Fig. 3e. Without the adhesion, most of the leg-floor contacts would not have been able to counteract the slipping force load, especially in the 'driving' T1, T2 legs on the way up, and T3 on the way down (Extended Data Fig. 3f).

## Reporting summary

Further information on research design is available in the Nature Portfolio Reporting Summary linked to this article.

## Data availability

Confocal imaging stack, flight and walking imitation datasets, base wing-beat pattern, grooming pose data, and trained controller networks are available at Figshare (ref. 20).

## Code availability

The fly model and code are publicly available at https://github.com/TuragaLab/flybody

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

**Acknowledgements** This work was supported by the Howard Hughes Medical Institute and Google DeepMind. We thank P. Trochim for writing the Blender-to-MuJoCo export plug-in; B. Moran for open-sourcing the plug-in; N. Gileadi for reviewing code submissions to dm_control; T. Rymer, E. Tenshaw and T. Paterson for performing APT annotations; J. Lappalainen, D. Deb and members of the laboratories of J. Tuthill and B. Brunton for critical reading of the manuscript and discussions; and K. Arrington and S. Preibisch for their contributions to the collaborative process. Z.S. was supported by the German Research Foundation through SPP 2041, Germany's Excellence Strategy (EXC number 2064/1, project number 390727645), the German Federal Ministry of Education and Research (Tübingen AI Center; FKZ 01IS18039A) and the European Union (ERC, DeepCoMechTome, 101089288). Z.S. is a member of the International Max Planck Research School for Intelligent Systems.

**Author contributions** J.M., K.M.B., M.M.B., A.A.R., G.M.C. and M.B.R. conceived the project. I.S. performed body imaging and built the Blender model. Y.T., R.V., A.A.R., G.M.C. and J.M. built the MuJoCo body model. Y.T. and G.N. designed the MuJoCo fluid model. Y.T. developed the MuJoCo adhesion model. A.A.R., C.M., G.M.C., K.M.B. and R.V. performed kinematics analysis and generated walking behaviour data. R.V. and G.-J.B. performed RL, with advice from S.C.T., J.M., K.M.B., M.M.B., A.A.R., G.M.C. and M.B.R. Z.S. contributed to the code base. R.V. and K.M.B. performed flight and walking behaviour analysis with advice from S.C.T., M.B.R., G.M.C., M.M.B. and A.A.R. S.C.T. and R.V. wrote the manuscript, with contributions from all authors.

**Competing interests** The authors declare no competing interests.

**Additional information**
**Correspondence and requests for materials** should be addressed to Yuval Tassa or Srinivas C. Turaga.

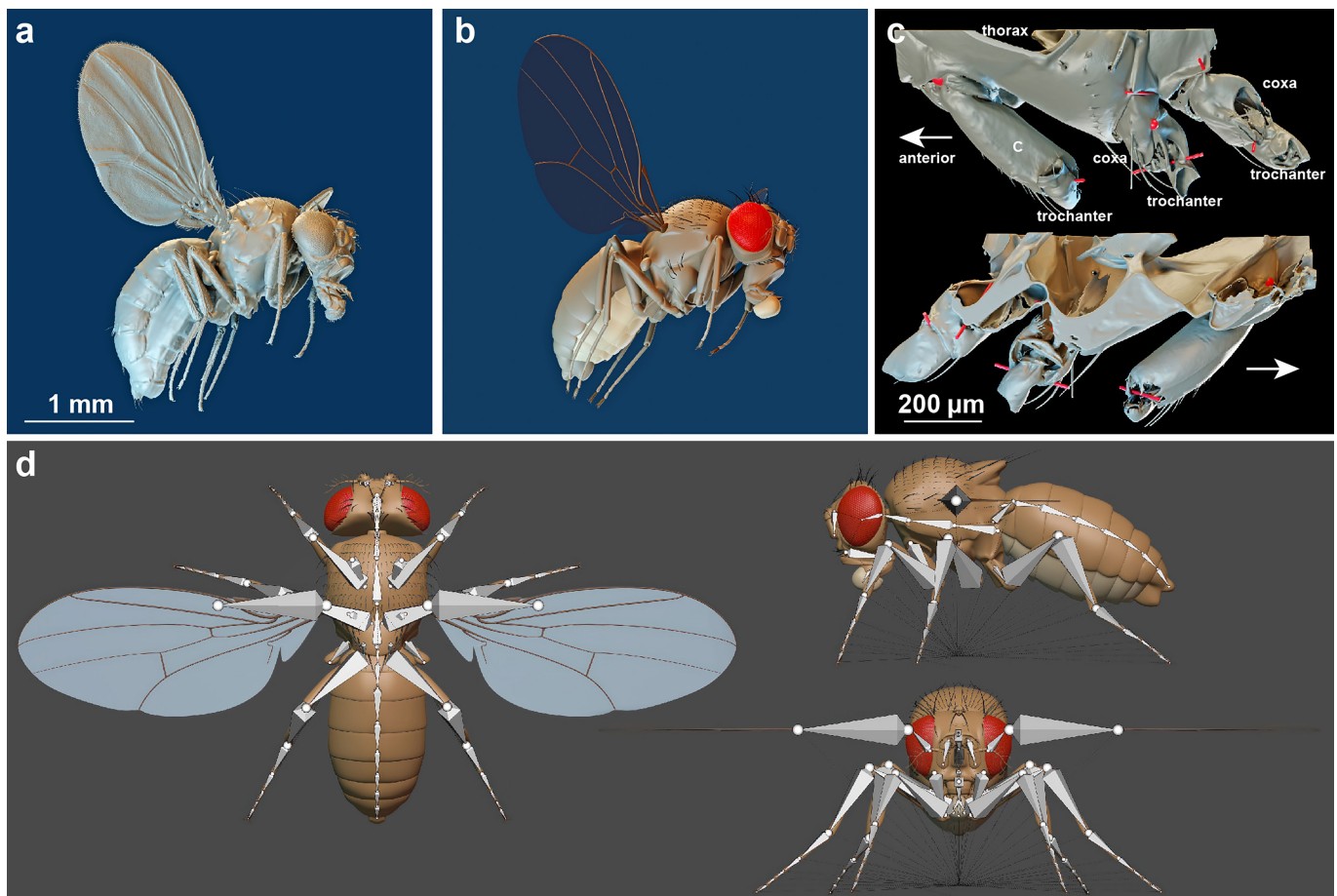

**Extended Data Fig. 1 | Constructing a 3D model of a female fruit fly from confocal data. a**, Hi-polygon (around 22.6 million faces) model of the fly reconstructed in Blender. **b**, Simplified low-polygon model (around 20,000 faces). **c**, 3D model of the left-side coxae based on the confocal data. Red bars represent hinge joints, spheres represent ball joints. Arrows indicate anterior. **d**, Dorsal, sagittal and frontal views of the rigged Blender model in the rest position, the elements of the armature called 'bones' shown as elongated octahedrons.

**a**  Legs T1, T2, T3 in rest position       **b**  Legs T1, T3 in grooming position

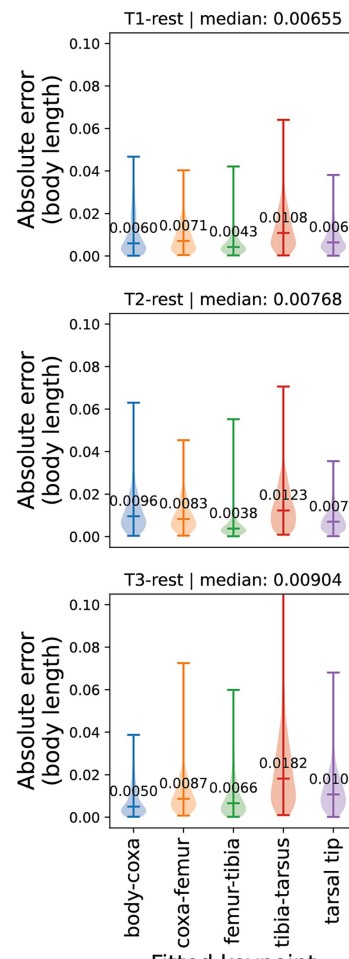

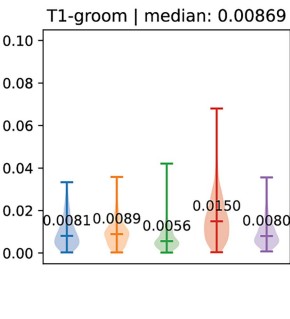

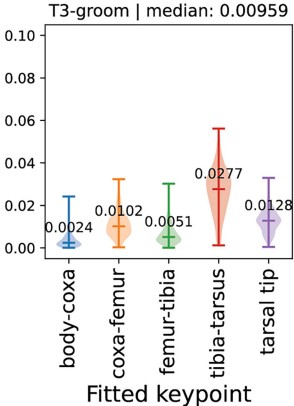

**Extended Data Fig. 2 | Leg DoF analysis.** Inverse kinematics fits of the fly model legs to real *Drosophila* poses during grooming behaviour (392 poses in total.) To separate the effect of DoFs from fly-to-fly size variability, the model legs were rescaled to match the leg lengths in each individual reference pose frame. Individual model legs were fitted separately by simultaneously matching five leg key points located at four leg joints and leg tip. **a,b**, Absolute fitting errors for each leg key point, in units of body length, are shown for (**a**) all legs in rest position, (**b**) leg pairs T1 and T3 in grooming positions. Horizontal bars and corresponding values are median errors for each key point. Median errors across all key points in each leg pair are also indicated.

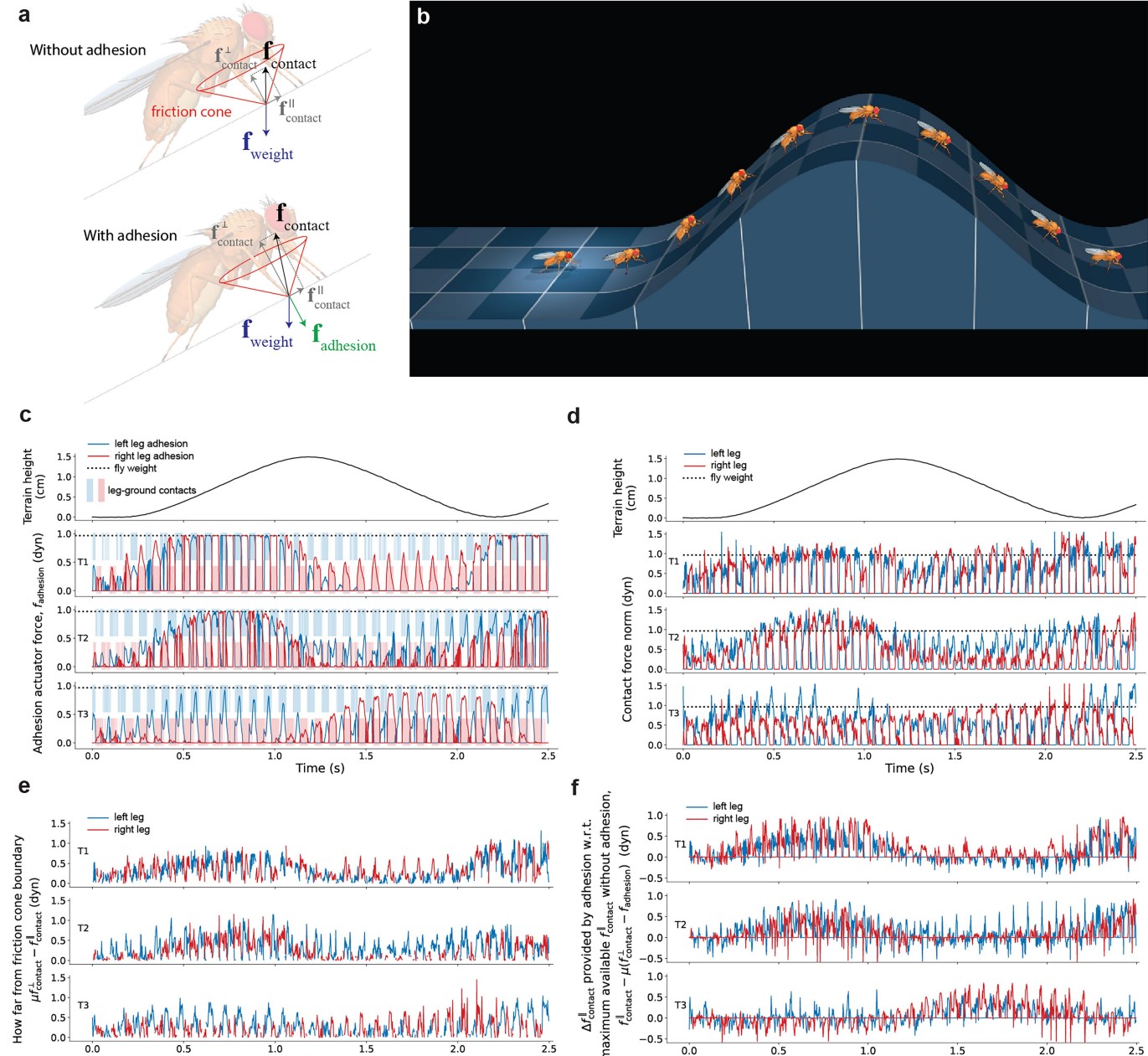

**Extended Data Fig. 3 | Adhesion and contact forces during walking on hilly terrain. a**, Schematic diagram of leg-floor contact forces for the fly model standing on an inclined surface. The adhesion actuator injects force in the normal direction, which in response increases the normal component of the contact force. This creates a larger margin between the tangential contact force component which resists slipping and the slip threshold (the friction cone boundary). **b**, Time-lapse of the trained policy rollout of the RL task where the fly model learns to use the adhesion mechanism to overcome sine-like hills. All the following panels correspond to this policy rollout. **c**, Adhesion actuator forces generated by the fly's claws during the policy rollout shown in **b**. In our model, the largest adhesion force per leg is one fly body weight. The fly body weight, mg = 0.96 dyn, is shown for comparison. The leg-floor contacts are shown for clarity. **d**, Contact force norm during the policy rollout. The fly weight is shown for comparison. **e**, The difference between the slip threshold force and the tangential component of the contact force. This is the "margin" available to resist slipping under external forces and propulsion generation. **f**, The difference between the actual tangential contact force and the largest tangential contact force that would have been available without adhesion. Positive means the contact would have slipped without adhesion. Negative means the contact would not have slipped without adhesion (that is, the contact is inside the friction cone already without adhesion).

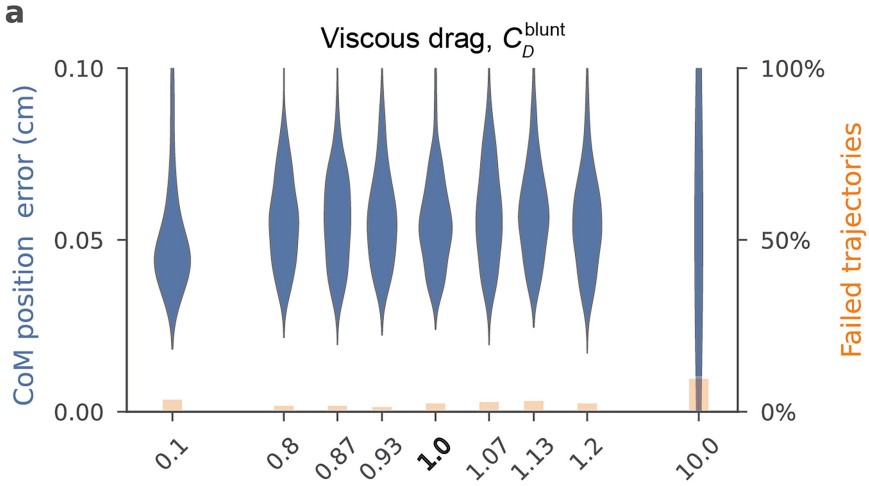

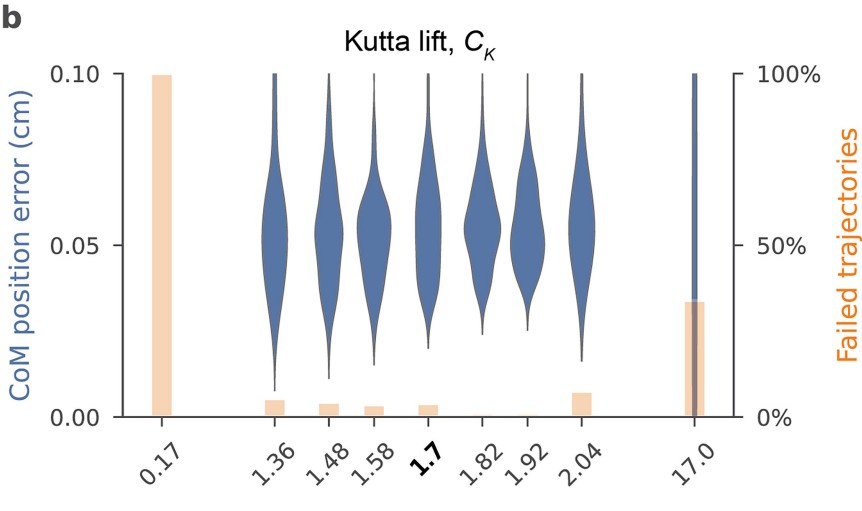

**Extended Data Fig. 4 | Sensitivity analysis of the dimensionless fluid model coefficients. a,b,** For viscous drag (**a**) and Kutta lift (**b**) components. Accuracy of imitation learning is shown in blue violins (body CoM error, 5 training runs, same test trajectories as in Fig. 2) and trajectory failure rate in orange bars (percentage of trajectories that crashed) for simulations varying each coefficient by up to 10×. Default values of the two coefficients are shown in bold. See the Supplementary Information and Supplementary Table 20 for fluid model details.

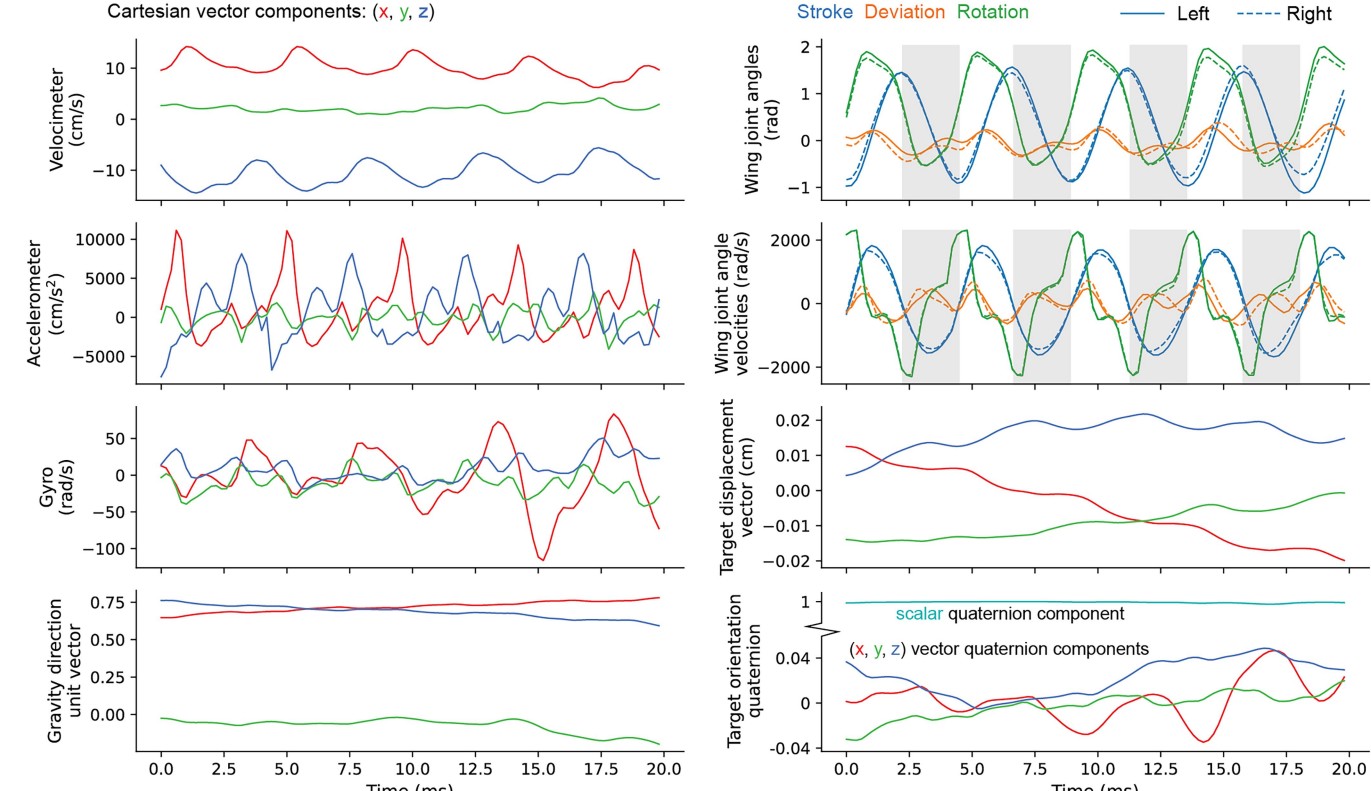

**Extended Data Fig. 5 | Example sensory input.** Sensory inputs to the flight imitation policy in Fig. 2. Vestibular sensors (velocimeter, accelerometer, gyro, gravity direction), proprioception (joint angles and joint velocities), and steering commands (target displacement and orientation) are shown. For clarity, we only show wing joint angles and omit the future preview of the steering command. More details in the Supplementary Information and Supplementary Table 6. Most of the observables are Cartesian vectors and their *xyz* components are correspondingly colour-coded as RGB. All inputs are represented in the fly's egocentric reference frame.

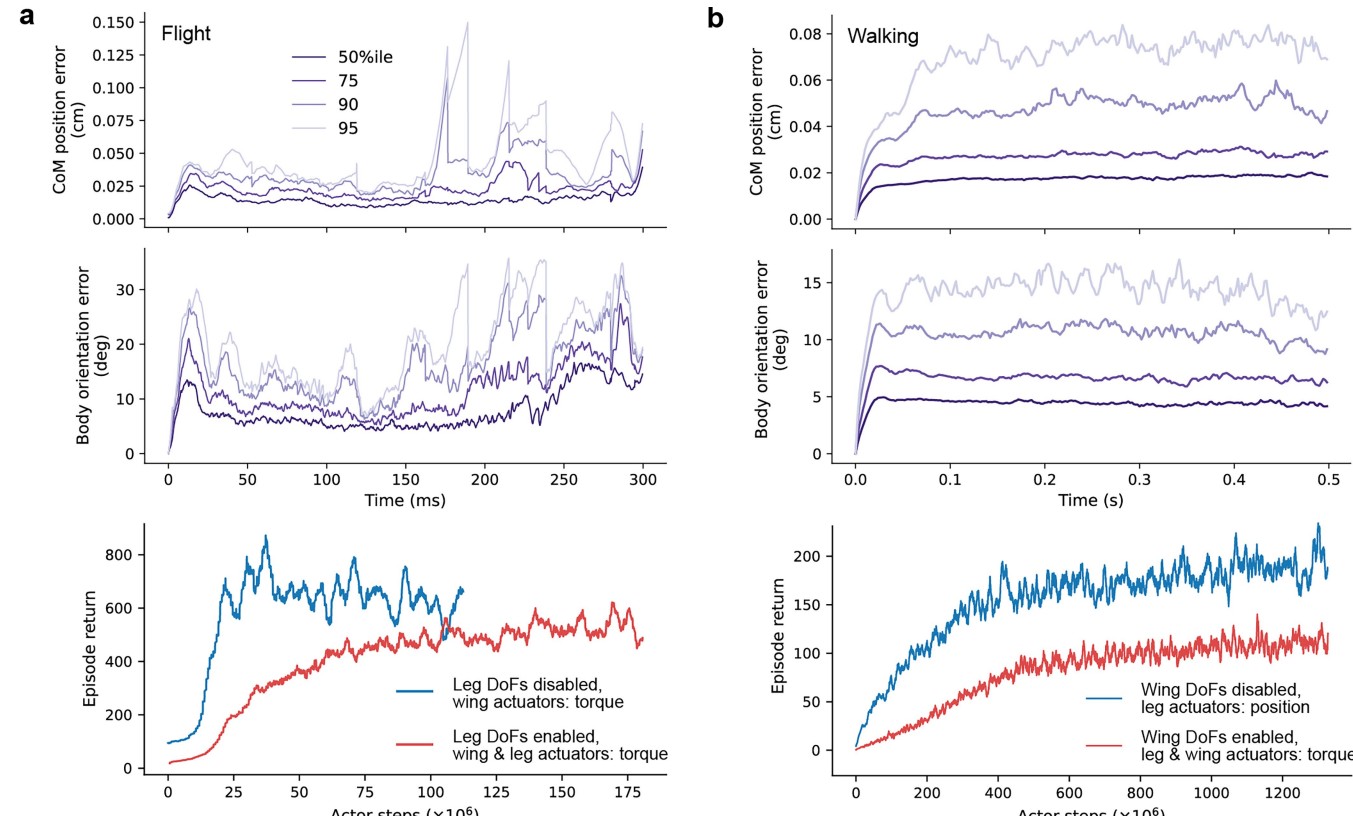

**Extended Data Fig. 6 | Alternative DoFs and actuators.** Performance of the modified fly model with all 102 DoFs enabled and position actuators replaced with torque actuators. **a**,**b**, Flight imitation task, same as in Fig. 2 (**a**) and walking imitation task, same as in Fig. 3 (**b**). Top, Middle: Percentiles of errors between the fly model and target fly CoM position and body orientation. Bottom: Learning curve comparison between the original (blue) and modified (red) fly model. Episode return (e.g., cumulative episode reward) vs MuJoCo control steps during training is shown. The training is slower for the modified fly model. For flight, the episode return at end of training is similar in both cases. For walking, an additional multiplicative reward term is required to keep the (now enabled) wing DoFs in folded position and it causes most of the discrepancy between the two learning curves. This reward term is only approximately satisfied, causing a reduction of the episode return by a factor of ~0.69 in the trained model.

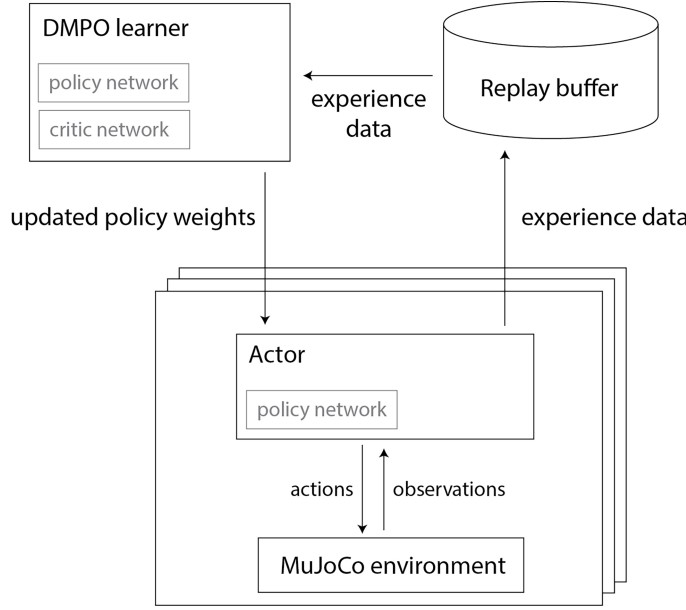

**Extended Data Fig. 7 | Distributed RL training architecture.** Multiple replicas of actors in MuJoCo environments collect experiences and feed them to a single replay buffer. The DMPO learner samples experiences from the replay buffer, updates the policy and critic network weights, and sends the updated weights to the actors' copies of the policy.

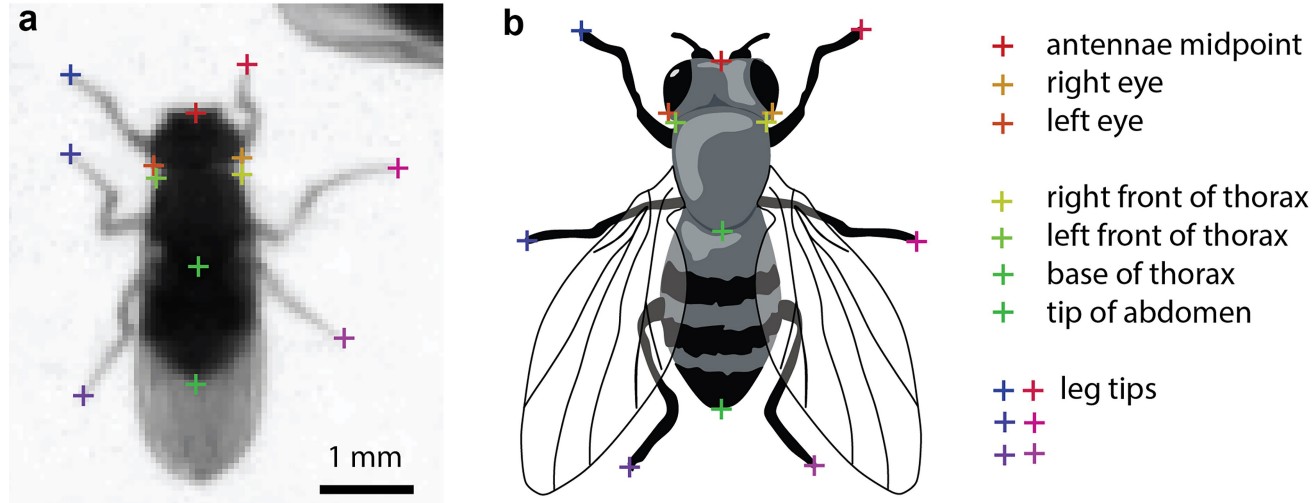

**a**

1 mm

**b**

+ antennae midpoint
+ right eye
+ left eye

+ right front of thorax
+ left front of thorax
+ base of thorax
+ tip of abdomen

+ + leg tips
+ +
+ +

**Extended Data Fig. 8 | Two-dimensional key points used to track fly walking. a**, A top-view video frame of a walking fly with key points inferred with the APT. **b**, Key-point definition. The fly schematic is provided by https://scidraw.io/ and is available at https://doi.org/10.5281/zenodo.3926097.

# Reporting Summary

## Statistics

For all statistical analyses, confirm that the following items are present in the figure legend, table legend, main text, or Methods section.

| n/a | Confirmed | |
|---|---|---|
| ☐ | ☒ | The exact sample size (*n*) for each experimental group/condition, given as a discrete number and unit of measurement |
| ☒ | ☐ | A statement on whether measurements were taken from distinct samples or whether the same sample was measured repeatedly |
| ☒ | ☐ | The statistical test(s) used AND whether they are one- or two-sided *Only common tests should be described solely by name; describe more complex techniques in the Methods section.* |
| ☒ | ☐ | A description of all covariates tested |
| ☒ | ☐ | A description of any assumptions or corrections, such as tests of normality and adjustment for multiple comparisons |
| ☐ | ☒ | A full description of the statistical parameters including central tendency (e.g. means) or other basic estimates (e.g. regression coefficient) AND variation (e.g. standard deviation) or associated estimates of uncertainty (e.g. confidence intervals) |
| ☒ | ☐ | For null hypothesis testing, the test statistic (e.g. *F*, *t*, *r*) with confidence intervals, effect sizes, degrees of freedom and *P* value noted *Give P values as exact values whenever suitable.* |
| ☒ | ☐ | For Bayesian analysis, information on the choice of priors and Markov chain Monte Carlo settings |
| ☒ | ☐ | For hierarchical and complex designs, identification of the appropriate level for tests and full reporting of outcomes |
| ☒ | ☐ | Estimates of effect sizes (e.g. Cohen's *d*, Pearson's *r*), indicating how they were calculated |

*Our web collection on statistics for biologists contains articles on many of the points above.*

## Software and code

Policy information about availability of computer code

| Data collection | All code pertaining to the manuscript is publicly available at https://github.com/TuragaLab/flybody |
|---|---|
| Data analysis | All code pertaining to the manuscript is publicly available at https://github.com/TuragaLab/flybody |

For manuscripts utilizing custom algorithms or software that are central to the research but not yet described in published literature, software must be made available to editors and reviewers. We strongly encourage code deposition in a community repository (e.g. GitHub). See the Nature Portfolio guidelines for submitting code & software for further information.

## Data

Policy information about availability of data

All manuscripts must include a data availability statement. This statement should provide the following information, where applicable:
- Accession codes, unique identifiers, or web links for publicly available datasets
- A description of any restrictions on data availability
- For clinical datasets or third party data, please ensure that the statement adheres to our policy

Confocal imaging stack, flight and walking imitation datasets, base wingbeat pattern, and trained controller networks are available at https://doi.org/10.25378/janelia.25309105

# Research involving human participants, their data, or biological material

Policy information about studies with human participants or human data. See also policy information about sex, gender (identity/presentation), and sexual orientation and race, ethnicity and racism.

| | |
|---|---|
| Reporting on sex and gender | N/A |
| Reporting on race, ethnicity, or other socially relevant groupings | N/A |
| Population characteristics | N/A |
| Recruitment | N/A |
| Ethics oversight | N/A |

Note that full information on the approval of the study protocol must also be provided in the manuscript.

# Field-specific reporting

Please select the one below that is the best fit for your research. If you are not sure, read the appropriate sections before making your selection.

☒ Life sciences ☐ Behavioural & social sciences ☐ Ecological, evolutionary & environmental sciences

For a reference copy of the document with all sections, see nature.com/documents/nr-reporting-summary-flat.pdf

# Life sciences study design

All studies must disclose on these points even when the disclosure is negative.

| | |
|---|---|
| Sample size | No statistical methods were used to pre-determine sample size. |
| Data exclusions | Walking examples for training the walking model were selected based on metrics detailed in the material and methods 'Reference walking data preparation' section. Only data from female flies was used in order to match the anatomical samples used for the body model construction. |
| Replication | The reference data was constructed from 9 replicate videos. |
| Randomization | No randomization was performed. |
| Blinding | No blinding was performed. This was not relevant since only one genotype was used. |

# Reporting for specific materials, systems and methods

We require information from authors about some types of materials, experimental systems and methods used in many studies. Here, indicate whether each material, system or method listed is relevant to your study. If you are not sure if a list item applies to your research, read the appropriate section before selecting a response.

## Materials & experimental systems

| n/a | Involved in the study |
|---|---|
| ☒ | ☐ Antibodies |
| ☒ | ☐ Eukaryotic cell lines |
| ☒ | ☐ Palaeontology and archaeology |
| ☐ | ☒ Animals and other organisms |
| ☒ | ☐ Clinical data |
| ☒ | ☐ Dual use research of concern |
| ☒ | ☐ Plants |

## Methods

| n/a | Involved in the study |
|---|---|
| ☒ | ☐ ChIP-seq |
| ☒ | ☐ Flow cytometry |
| ☒ | ☐ MRI-based neuroimaging |

# Animals and other research organisms

Policy information about studies involving animals; ARRIVE guidelines recommended for reporting animal research, and Sex and Gender in Research

| | |
|---|---|
| Laboratory animals | Drosophila melangaster adults 3-5 days post eclosion were used for the walking reference data and 5-6 days post eclosion were used for the anatomical imaging. |
| Wild animals | N/A |
| Reporting on sex | Due to morphological differences between male and females flies, one sex needed to be selected for the body model. Based on previous research, the availability of the female connectome, and the larger size of female flies, we restricted this work to females. Females were selected based on morphological characteristic such as size, pigmentation, and genitalia shape. |
| Field-collected samples | N/A |
| Ethics oversight | N/A |

Note that full information on the approval of the study protocol must also be provided in the manuscript.

# Plants

| | |
|---|---|
| Seed stocks | N/A |
| Novel plant genotypes | N/A |
| Authentication | N/A |

