## [Peer Review File · Nature]

Whole-body physics simulation of fruit fly locomotion

Corresponding Author: Dr Srinivas Turaga

Version 0:

Reviewer comments:

Referee #1

(Remarks to the Author)

Meta-note re: Melis et al. (2024) and NeuroMechFly

I understand that it is a matter of policy that preprints are not taken into account when making publishing decisions in Nature Portfolio journals. Melis et al. was preprinted on bioRxiv on July 2, 2023 and published in Nature on April 17, 2024, while this article was received by this reviewer on April 12, 2024, suggesting that it was submitted for review prior to the publication of Melis et al. Of note, the two papers share a co-author (Igor Siwanowicz) who made significant contributions to both. I just want to state these facts for posterity and for consideration of how the review comments below should be weighed since many of the points hinge (-) on the similarities and differences between these works. I expect that the overall review stands, regardless of how Melis et al. is factored into the review process (if at all), but I chose to include these comments since they are too essential to the assessment of the science, independent of journal editorial policy.

Similar caveats could be attributed to comparisons to NeuroMechFly since the newest version that added adhesion for terrestrial locomotion (and other features also presented here) is still in preprint stage (Wang-Chen et al., 2023). This is partially mitigated by the fact that the previous iteration was already published (Lobato-Rios et al, 2022).

Meta-note re: Formatting

I attempted to use markdown-style formatting to drastically improve the readability of this review (just paste it into a markdown formatter like HackMD or GitHub). Hopefully this survives the Nature manuscript tracking system, but if not, this comment is for you MTS team!!

Summary

In this manuscript, Vaxenburg et al. present their work on developing a biomechanically-realistic simulation of a fruit fly, its body, ground and flight behavior and neural control of movement. This work is emblematic of the recently described movement towards incorporating embodiment and NeuroAI as an avenue for overcoming the limitations of modeling frameworks in neuroscience. To that end, this work presents a compelling example of the integration of disciplines and techniques, including: advanced microscopy for whole body imaging, biomechanical model development, sophisticated fluid simulation in a physics engine, motion capture and registration of real behavioral kinematics, and deep reinforcement learning (DRL) for kinematic imitation and control using artificial neural networks. Furthermore, the authors demonstrate that this blend of approaches enables simulation of terrestrial and flight behavior, including in vision-guided tasks that real flies are able to solve, providing evidence of the validity and potential utility of this framework. Finally, the scientific software engineering standards and practices in this work are exemplary and should be considered a gold standard: accessible and functional open-source code, operationalized data artifacts (body model and model checkpoints), and contributions of new capabilities developed as part of this work to a broadly used scientific software library (MuJoCo).

Despite the technical sophistication of this work, in its current state, there are a number of substantive (but addressable) concerns that limit its impact and contributions. Described in further detail below, the gist of it is that a number of choices and features of the biomechanical, fluid simulation and DRL/NeuroAI modeling could be better justified, validated and/or explicitly connected to the resulting limitations. These are particularly salient in the context of other related works (preprint

policy notwithstanding) which, if not impinging on novelty, certainly serve as instructive examples of how some of these concerns could be better addressed.

Major concerns

1. Simplification of the body model: Body actuators

The body model (other than the wing, discussed in point 2 below) largely uses position and leg adhesion actuators. This is a significant design choice since it simplifies practical optimization and body model development, but comes with trade-offs regarding the (meaningful) sacrifices in biological fidelity.

This reviewer deeply appreciates the difficulty and complexity of building a body model that incorporates muscles and recognizes the value of representational abstraction. That said, the use of position actuators constrains the potential of this work for immediate use. This becomes relevant for mechanistic studies in fields like motor neuroscience where accurate representation of muscles is essential for establishing homology between model and biological system components, and where the neural control signals (which vary drastically based on actuator choice) are the objects of interest.

While this limits the applicability of this work in neuroscience, it can still be useful for biomechanics research interested in grosser level properties such as kinematics. To this end, and to address other concerns:

1.1. Demonstrate the utility of the adhesion actuators to substantiate the impact of this contribution.

In the manuscript, there is a statement that "However, we observed that the agent generally learned to activate adhesion when legs were in stance (on the ground)" (L281-283), but I don't see any data on this. Showing that this is the case during different phases and gait modes of locomotion would help to show that this addition to the model is useful as it enables estimation of a modality of motor control that is difficult to measure *in vivo* during free behavior, and can be compared to previous work on the role of adhesion during locomotion (Ramdya et al., 2017).

Another way to show the significance of this contribution would be to show that it enables complex 3D navigation. Right now this is only demonstrated qualitatively in Fig. 1h. Showing this in action in a task (imitation or goal-driven) that would be challenging without adhesion actuators would prove the point, and is achievable by devising an environment that features navigation across hilly terrains (as in Merel et al., 2019; Wang-Chen et al., 2023) or steep surfaces (Ramdya et al., 2017).

(A task that uses the adhesion actuator added to the labrum would also enhance the significance of this contribution.)

1.2. Discuss and/or show evidence to justify the use of position actuators over torque actuators.

Position control less accurately reflects the way muscles generate forces and control movement in biological systems. As mentioned, this may be an acceptable limitation in some cases but not others. Torque-based actuators better mimic the way that muscles generate forces and influence joint movement, providing a more biomechanically accurate representation of muscle function. Given that torque actuators are used in other parts of the body model (e.g., wing hinge), it stands to reason that they could be used to model the other joints as well.

The authors could address this limitation by (i) elaborating on their reason for this design choice (e.g., robustness to instabilities, tractability of optimization, etc.); and/or (ii) trying out torque actuators in a limited experiment to show that the differences are insignificant (or at least bounded) in some conditions. In either case, an explicit discussion on the downstream implications of using this actuator type is essential.

1.3. Discuss how big of a gap the current work leaves before muscle actuators can be introduced.

The authors state on L76-79: "The dataset also enables the identification of origin and insertion sites of muscles, locations of the proprioceptive hair plates of the neck, coxae, trochanters, wing base and halteres — information that can be incorporated into future versions of the model."

Since the absence of muscles is a major limitation of this work, providing an estimate of the size of the gap between the current model and one that incorporates these improvements will be helpful to readers that are unfamiliar with the field to gauge the significance of the current work in a broader context.

A major hurdle overcome by this work (and a major contribution) is that it shows the feasibility of reconstructing the body model from confocal imaging data, a modality of microscopy that is much more accessible than some alternatives like XNH or EM.

If the current dataset is of sufficient resolution to recover details of musculoskeletal anatomy necessary to develop muscle-based actuators, then what else is missing? Image segmentation of morphological features of interest? Manual work in converting these data into MuJoCo-based muscle actuator parameters?

It would also be useful to state how the selected imaging modality compares to micro CT.

2. Simplification of the body model: Wings

In this work, the authors model the wing as a slender ellipsoid controlled by a ball joint driven by torque-based actuators. This reviewer appreciates the morphological and biomechanical complexity of the wing and hinge, but given their importance to flight behavior, I believe it is essential to characterize to what extent the simplifications employed here impinge upon the utility of the model for studying insect flight.

2.1. Describe to what extent these simplifications impact the biological validity and interpretability of the model.

The concerns raised in (1) about the utility of this model for neuroscience in the absence of muscles in general are still applicable here and should be discussed in the context of flight.

Next, caveats about timing/journal policy aside, it is hard to overlook the fact that Melis et al. (2024) takes a similar approach to modeling flight biomechanics in *D. melanogaster*, but in deciding to more accurately model the wing hinge, indirect flight muscles, and sclerites, discovers mechanistic roles and functional consequences of these components.

Given this knowledge, or just drawing from the long history of work on insect flight (Boettiger and Furshpan, 1952; Pringle, 1957; Miyano and Ewing, 1985; Ennos et al., 1987; Dickinson and Tu, 1997; etc.), a discussion of how the design choices of the wing model in the current work impacts the biological or biomechanical validity of flight behavior is highly warranted.

Specifically, a description (or, preferably, a demonstration) of how the current wing model could be used to generate empirically testable hypotheses would go a long way towards providing evidence for its utility, and provide a path towards measuring validity.

To be explicit: Figs. 2 and 5 start going towards this, but do not explicitly demonstrate applicability towards empirically testable hypothesis generation, nor do they emphasize the design choices made in the biomechanical model of the wing without conflation with the control model.

2.2. As in (1.3), provide a gap analysis to justify and reinforce the significance of the contributions.

In addition to describing what is needed to accurately reconstruct the wing and hinge (is confocal microscopy, and this dataset specifically, sufficient?), it would be helpful to understand to what extent the wing model is limited by the fluid dynamics simulation.

For example, is it a hard requirement to use an ellipsoid geom to represent the wing due to how the fluid simulation is implemented, or could other geoms be employed? Is this a fundamental limitation of the fluid algorithms, requiring new math to describe interactions with other shapes? Or is this a tractability issue, where other geoms are simply too slow to compute against (i.e., could be solved by new approximation algorithms or more compute resources) or too inaccurate/unstable (i.e., could be solved by more numerically precise implementations)?

I'll also point out that the reported performance estimates of the physics step (55.5 ms for 10 ms of simulation on a single CPU core; L149) would indicate that there is plenty of headroom for a more faithful wing model – assuming that simulation speed is (one of) the constraints.

Describing (and ideally providing evidence, theoretical or empirical) for the need for this simplification would be helpful to contextualize the significance of this contribution.

3. Phenomenological fluid model for insect flight

I will preface this section with a disclaimer that I am not, by any stretch of the imagination, an expert in fluid mechanics or its simulation. That said, allow me to restate the contributions introduced here as far as I've come to understand them after extensive background reading:

- Added mass: The added mass accounts for the inertia of the surrounding air during wing movements. This is important if we want to estimate energetics and forces needed to displace air around the wing.
- Viscous drag: This captures the forces that oppose wing motion and depends on factors including wing velocity, shape and air viscosity. This component is essential for modeling energy dissipation and is most important at high Reynolds numbers (fast moving objects) such as during fast flight.
- Viscous resistance (aka linear drag): Similar to viscous drag, but better captures models interactions at low Reynolds numbers (slow moving objects), such as during slow/precise flight maneuvers and hovering.
- Magnus force: Flies rotate/flap their wings while moving forward to generate a Magnus force perpendicular to the wing's path. This force comes into play in lift generation and turning maneuvers.
- Kutta lift: The Kutta condition accounts for the lift generated by the circulation of air around the fly's wings, especially at the leading edge. It is the primary(?) generator of lift forces in *Drosophila* flight.

As disclaimed, I am deeply uneducated in the nuances of aerodynamics or fluid mechanics models, but I feel that I may not be alone in that given the target audience for this work. Given that that's the case, additional clarity on the following concerns may help to emphasize the importance of the contributions presented in this work to a broader audience.

3.1. Describe why the model parameters are entirely phenomenological and what potential alternatives might be.

The authors describe the math and physics behind each of the fluid model components in great detail, many of which have been objects of study dating back centuries. Given the elegance of this physical modeling, it is a bit disconcerting to then see that the models are then parametrized by a set of suspiciously exact coefficients (`fluidcoef = [1.0, 0.5, 1.5, 1.7, 1.0]` (L562)) that seem to have been hand-tuned until the fly was able to hover in place.

This is further compounded by having a hand-engineered model of wingbeat generation that controls a simplified model of the wing through torque based actuation. The wingbeat generation model, while derived from data, is already a phenomenological simplification of the true biological control mechanism, and issues with the body model and actuators are discussed in (2). So now we have three complex interacting components with unknown degrees of inaccuracy, all being used in concert to derive physical constants in a seemingly guess-and-check approach?

I hope that my domain ignorance here is biasing my reading of this procedure, but I would encourage the authors to dedicate some text to justifying why this is a reasonable thing to do. For example:

3.1.1. Why can't the `fluidcoef` parameters be inferred from known physical constants or derived from empirical data in other systems outside of insect flight? Surely common physical properties like air viscosity have been measured in other settings?

3.1.2. To what extent are the choices made in deriving this phenomenological model a constraint of the generalization of the Andersen et al. (2005) approximation model (a key contribution of this work)? As this new fluid model is a novelty of this work, it seems within scope to describe its limitations (e.g., it may require phenomenological estimation of key constants due to change of units or simplifications introduced in the approximation).

3.2. Describe how the fluid dynamics model interacts with properties of the wing biomechanical model.

Since these are both novel contributions of this work, it would be instructive to domain-ignorant readers such as myself to understand how the modeling choices in the physics affect how I can use the biomechanical model and what uncertainties they introduce.

As concrete (but possible naive) examples:

3.2.1. How are the added mass and viscous drag components affected by using a simplified wing model? Does the absence of realistic sclerite interactions and muscles affect the accuracy of how these components impact estimates of energy or forces? Do these inaccuracies manifest only at high Reynolds number conditions? What properties of the resulting kinematics are affected (e.g., velocity? acceleration?)?

3.2.2. How are the Magnus force and Kutta lift components affected by using a simplified wing model? Are the ball joint/torque actuators sufficient to capture the realistic dynamics of wing rotation and angle of attack in a way that impacts calculation of lift forces?

It would be especially informative to understand how the answers to these questions apply to low/high Reynolds numbers conditions, such as during hovering versus extreme maneuvers.

4. Reuse of imitation learned low level controller modules

As an exemplar of combining embodiment with NeuroAI, this work leverages imitation learning (via DRL) to train neural network controllers to reproduce the kinematics observed during real behavior using motion capture.

Specifically, the authors train steerable low-level controllers (LLCs) which are intended to translate intention into motor commands and can be thought of as homologous to the ventral nerve cord (VNC) in flies (L163-165).

As these modules are trained on motion capture data, they enable alignment to biology which in turn has the potential to improve the chances that learned artificial neural controllers are consistent with those afforded to biological controllers through billions of years of evolution. This is, in fact, the thesis and motivation behind much of this work, as cited throughout the text via references to Zador et al. (2023).

Despite this motivation and potential major strength of this work, the authors do not demonstrate how the trained models can be reused for anything other than kinematic replay. This is a major missed opportunity to demonstrate the utility of imitation learning in realizing the purported advantages of embodied NeuroAI.

I appreciate that the emphasis of this work is on the development of the body model, but a more convincing demonstration of the capabilities enabled by accurate biomechanical modeling would go a long way towards providing evidence for the

significance of this work.

This could be achieved by, for example:

4.1. Show that imitation learning combined with embodied NeuroAI enables estimation of sensory inputs and motor outputs during kinematic replay.

A major advantage of a physics simulator is that we can infer the sensory inputs, including perceived forces and visual inputs, that are useful to neuroscientists interested in studying the neural basis of visually-guided behavior, proprioception, mechanosensation, feedback, and etc. Inference of muscle activations is limited by the choice of actuators, but even the more abstract action space may contain relevant information about the manifold of low level motor control signals.

Showing that you can estimate these parameters during kinematic replay demonstrates a key use case of the model and framework that would not have been possible otherwise.

4.2. Show that imitation learning facilitates reusability of motor skills in goal-driven tasks.

Another major practical capability enabled by this work is the ability to bootstrap policy learning in goal-driven tasks (such as the visually-guided tasks in Fig 4) by reusing the LLC network weights, even if the high-level controller (HLC) needs to be trained from scratch.

Ideally, an architecture would be used that is more amenable to reuse while minimizing the domain shift, such as in CoMic (Merel et al., 2020; cited in the text), but even without adopting advanced techniques, at a minimum the network weights could be reused for conventional weight initialization-based transfer learning.

This could be easily implemented for the bumps and trench tasks, and would presumably make task training converge faster and more stably, as has been shown in previous work adopting this strategy (e.g., Liu et al., Science Robotics, 2022).

Demonstrating these advantages would demonstrate value for the robotics/CS members of the audience as a practical matter/engineering trick for training complex policies based on biological constraints derived from MoCap, as well as increase the significance and impact of the work by providing support for the importance of embodiment in NeuroAI (Zador et al., 2023).

Minor concerns

5. Reference motion capture data: Walking

Since imitation requires registration of motion capture data to the biomechanical model, standard algorithms for this inverse kinematics step (as used here) require the reference data to be in 3D. It can be difficult to do 3D pose estimation in walking flies owing to their size and the need to multiple camera views for accurate 3D triangulation.

That said, the approach employed here is pretty unusual and introduces the need for extra validation (partially addressed in ED Fig 2). Namely, the authors use a 2D pose-tracked dataset of flies walking and "lift" it to 3D using a hand-crafted model of the fly gait cycle. To do this, they use manually estimated parameters that describe the z-coordinate of the leg tips during the walking cycle from manually annotated videos in a previous paper (Akitake et al., 2015).

This is super weird as: (i) this fails to capture the variability outside of the typical trajectories; (ii) is biased and constrained to walking bouts or idle behavior; (iii) there exist easy-to-use and well-validated methods for doing deep learning-based lifting of 2D poses, including in walking flies specifically (Gosztolai et al, 2021); (iv) there exist plenty of datasets and approaches for 3D pose tracking of walking flies (e.g., Günel et al., 2019; Karashchuk et al., 2021).

A minimal effort improvement to this step could be to apply LiftPose3D (open-source, easy to use in this reviewer's experience, has fly-specific checkpoints) to obtain 3D trajectories, or use one of the existing datasets that were derived from multi-view videos.

6. Reference motion capture data: Flight

I'm a bit confused as to the use of the *Drosophila hydei* datasets from Muijres et al. (2014, 2015). Is it because of the free flight tracking as opposed to simulated flight on a pin?

I would think that there are *D. melanogaster*-specific datasets available (and ample methods to generate this data) available in the below refs:

Tammero and Dickinson (2002)
Fry et al. (2003)
Ristroph et al. (2009)
Straw et al. (2010)

van Breugel and Dickinson (2014)
Ben-Dov and Beatus (2022)
Melis et al. (2024) (Dataset doi: 10.22002/aypcy-ck464)

Is there really a strong reason to use *D. hydei* data for such a central experiment? It would be helpful to elaborate on this choice considering the preponderance of historical studies on free flight in *D. melanogaster*. Switching to using *D. melanogaster* data would also significantly strengthen the evidence for the validity of the current work.

7. Overstatement of potential short-term capabilities

The authors describe some of the short-term capabilities enabled by this work:

"In the **short term**, our body model and imitation learning framework can enable the model-based investigation of the neural underpinnings of sensory-motor behaviors such as escape invoked by looming stimuli [Card, 2012], gaze-stabilization [Cruz and Chiappe, 2023], the control of movement by the ventral nerve cord [Lesser et al., 2023, Azevedo et al., 2022, Cheong et al., 2023, Marin et al., 2023, Takemura et al., 2023]." (L391-396)

However, realistically, it seems unlikely that the framework in its current state could actually be employed in investigations across all of these behavioral domains.

For example, escape behavior requires a transition from terrestrial locomotion to flight, punctuated by the complex dynamics involved in take-off. In the current work, there are separate controllers for walking and flight, and effectively different biomechanical models due to the specializations employed in each setting, such as turning off legs during flight. The enhancements that would need to be made to the framework are non-trivial, involving (at least): (i) training combined neural controllers or coming up with an MoE-style switching scheme; (ii) investigating and likely improving the physics models for take-off that combine both ground forces and complex aerodynamics (where turbulence is more important?); (iii) overcoming intractability of high DoFs if all actuators are active or engineering capabilities for mid-simulation toggling of actuators in MuJoCo + dynamically changing the action space in neural network controllers. I'm not even sure what would need to happen with the WBG.

As another example, control of movements by the VNC, such as for the grooming sequence, would require significant additional work to: (i) tweak the body model to angle ranges and actuator DoFs for complex leg movements like mid- and posterior-thorax grooming; (ii) capture the dynamics of the interaction between legs and soft tissues like the wing (currently not modeled as a deformable structure), antennae and arista, as well as how the adhesion actuators come into play; and (iii) add sensory modalities relevant to this behavior, such as olfaction or fine-scale mechanosensation (e.g., dust) which are external drivers of the motivation to initiate and engage in grooming.

These could be addressed by either removing or requalifying the statement "short term", by more accurately describing the gaps, or by just highlighting specific classes of behaviors that are actually enabled by this work (e.g., locomotion and flight).

I also think that the extent to which "neural underpinnings" are accessible via this model is highly debatable given the limitations outlined in this review and without providing any empirical evidence of predictability of neural correlates (e.g., from simultaneously recorded in vivo activity as in Melis et al and others).

8. Typos

L125-126: Should "Most important..." be part of the previous sentence or is there a clause missing?

L250: "manuevers" -> "maneuvers"

Nature checklist

A. Summary of the key results

Provided in the beginning sections.

B. Originality and significance: if not novel, please include reference

Comments on novelty provided all throughout the review. Major concerns are overlap with Melis et al. (2024) and Wang-Chen et al. (2023) – refer to sections at the top of the review for specific comments on these.

C. Data & methodology: validity of approach, quality of data, quality of presentation

Key concerns revolve around validity and are elaborated on throughout the review. Quality of data is largely ok, with some concerns about the MoCap reference data described in points (5) and (6).

D. Appropriate use of statistics and treatment of uncertainties

Not much of this, but it also is not too applicable for this work.

****E. Conclusions: robustness, validity, reliability****
Discussed inline.

****F. Suggested improvements: experiments, data for possible revision****
Discussed inline.

****G. References: appropriate credit to previous work?***

Undoubtedly, there are missing references given how many fields this work integrates but it seems mostly okay as far as I can tell. It could use an enhanced discussion of recent work already cited in this paper (Melis et al., 2024; Wang-Chen et al., 2023).

****H. Clarity and context: lucidity of abstract/summary, appropriateness of abstract, introduction and conclusions****
Great! Well written and clear :)

Accountability

This review was written by Talmo Pereira <talmo@salk.edu> who opts to forsake anonymity in favor of ethical reviewing through accountability and transparency.

(Remarks on code availability)

The code is great! The authors could do better with the conda environment file (explicitly specify channels, impose some version constraints), but for interested audiences who would actually make use of the code, it's definitely in great shape (until Jax updates of course ;)).

They could document and streamline artifact download (e.g., data and model checkpoints needed to run the examples), but this can be overcome pretty easily by motivated users.

This reviewer was able to install and run the code and examples successfully!

Referee #2

(Remarks to the Author)

A. Summary of key results

The manuscript provides a great tool for future work on the integration of models of the muscular and neural control of sensorimotor behavior in an embodied context, i.e. a biomechanical model of the body of a fruit fly and its interaction with the surrounding physical world. The whole-body model is implemented in the open source MuJoCo (Multi-Joint dynamics with Contact) physics engine and elegantly incorporates both, realistic information on the geometry, weight, and degrees of freedom of 67 body parts as well as interactions of the 67 body parts with each other and interactions with an approximation of the physical environment during locomotion. Based on the assumption of uniform density MuJoCo estimates the body parts moments of inertia, and actuators are added to control joint movements and positions. Interactions with the physical world are approximated by a phenomenological fluid model that generalizes and extends previous work. The body model is then equipped with idealized (non-biological) sensorimotor and visual feedback controllers that can be trained to mimic locomotor movements of real flies. An artificial neural network (ANN) receives MuJoCo simulated sensory input and is trained by reinforcement learning in closed loop to drive the actuators in the body model. The main finding is that the highly flexible artificial neural network can indeed be trained to produce the simulation of locomotor movements that closely resemble basic walking and flight tasks as observed in the biological system.

B. Originality and significance

The model is all done on a high technical level and can be used real-time, the code is well documented and freely available. Additionally, forces like the Kutta lift and the Magnus effect that have not been part of previous models have now been implemented. Additional information on muscles and sensory biology can be incorporated in the future, so that the model is general purpose and extensible. Therefore, in the long run, if smartly paired with other approaches, data, and experiments, this could really help to gain novel insight into the neural control of sensorimotor behavior. However, as it stands the study does neither provide novel insight into the neural control of sensorimotor behavior as the model lacks biological neural components, muscles, material properties, and sensors, nor does it rigorously test the impact of the new forces implemented on locomotion or the limits of the whole-body model. In sum, the generation of the fine biomechanical model and the accompanying proof of principle that it can be used in a simulation to train an artificial neural network to mimic movements as macroscopically observed during basic aspects of locomotion represent strong methodological advances but little advances to our understanding of the control of locomotion.

C. Data & methodology: validity of approach, quality of data, quality of presentation

The data presented are high quality and the model is finest state of the art. However, as it stands, this approach does not seem to provide key novel biological insight into the neural control of sensorimotor behavior.

1. The finding that an artificial neural network (ANN) can be trained to produce walking- and flight-like locomotor patterns

that mimic movements produced by real animals does not mean that it employs similar rules that are used by the biological system. This is somehow implied in the manuscript, but it has not been tested. For example, it is stated that the project models the neural control of sensorimotor behavior, but the artificial neural network is not based on experimental insight on neural connectivity or physiology, and thus remains a black box. Can different types of ANNs learn to produce the flight and walking tasks shown? If yes, this would indicate that multiple strategies/network structures can be used to accomplish the task, but it remains unclear which ones are employed by the brain. See point 1 under suggested experiments (section F).

2. There are some concerns that the ‘anatomically-detailed biomechanical whole-body model’ as presented here may be too much of a simplification to impose realistic requirements to the neural controller, particularly the lack of any stretch-activated elements or other material properties and the fact that the controller needs to only learn deviations to a “nominal wing beat pattern” with no feedback or resonance. The concern is that the ANN can learn to realistically simulate locomotion of a biomechanical whole-body model in a physical environment, but the demands to the neural controller are so different from the real biology that one does not learn much about the nature of the biological neural networks that normally control locomotion. Below are a few examples substantiating this concern. First, it is well known that differential elastic and other mechanical properties of the exoskeleton are highly important to locomotor movements, such as the resonating properties of specific parts of the thoracic cuticle and the wing hinge during flight. During walking, joint stiffness as mediated by material properties and co-contractions of antagonistic muscles are at least as important as the movements of the leg joints, but only the joint movements and positions are actuated. Material properties of the exoskeleton, joints, and tendons are not considered in the biomechanical whole-body model. Second, the properties of the skeletal muscles have not been included. In particular during flight, resonating properties and stretch activation of flight power muscles are extremely important for setting the relationship between NS output and locomotor movement. In the case of stretch activated asynchronous muscles, the NS outputs only every 10 to 40 contractions of the muscles. By contrast, flight steering muscle contractions are related very differently to NS output, in some cases 1:1 between motor neuron firing and muscle contraction. Therefore, wing power and wing steering are coupled to the neural controller in entirely different ways. However, in the simulation presented the neural controller gives output signals to actuators that drive joint movements and positions in the same way for both. Such obvious deviations from the biology of the system are neither acknowledged nor discussed. Third, degrees of freedom and joint position ranges that are used to actuate the wings or the leg joints are based on movements observed during grooming movements. The joint kinematics during grooming might be in a similar working range as those during walking, but for sure not as those during flight. It is clear that not all biological details can be implemented in the body model, but a critical evaluation of the potential benefit of implementing some key additional aspects would be very helpful. This requires additional experiments and a critical discussion of the potential consequences of what is omitted.

3. What do we really learn about the biology of flight and walking? For example, how much better would the simulation do with more sensors, what is the minimal number of sensors to have adequate learning of the tasks tested? Testing this experimentally could give some information as to which types of sensory feedback are most essential for locomotor control. What is the relative impact of the forces that have not been implemented before (e.g. magnus force, Kutta lift), and most importantly, how complex can the behavior get and still be successfully trained? Real flies do rapid saccades and fast changing, complex flight trajectories. Real flies walk not only on horizontal surfaces, but can climb various slopes, walk upside down on the ceiling, and climb obstacles. What is the most complex locomotor behavior that can be learned by the ANN in this model? Can a neural network controller trained on grooming climb a slope? Testing this would give deeper insight into both the limits and the potential power of the model for future studies.

D. Appropriate use of statistics and treatment of uncertainties

The statistics used to make the model seem state of the art. However, as in any model, not all biology can be incorporated, Therefore, remaining uncertainties are the nature of the approach and must be accepted. However, these should be spelled out more clearly.

1. More detailed information on how well the model sensory system matches the real sensory system would be highly useful. Table 20 lists sensors known and studied in real flies and next to this, the corresponding model sensors. However, it remains unclear how well the model sensors represent the measurements taken by the biological sensors. How well does the model haltere gyroscope represent the measurements of Coriolis forces, balance, and equilibrium that are so far known to be provided by real halteres? It is clear that the biological sensors are far from completely understood and that the model can and does unlikely need to fully copy the biological sensors, but some assessment of the mechanosensory experiences of the model would be very useful.

2. Related to the above, during specific tasks sensors can be switched off and on, and their properties can be modified. Do some tasks require switching off certain sensors, such as switching off resistance reflexes during locomotion in real insects? Has this been systematically analyzed (not just recorded)? How important was changing sensor properties for which task, and how well do these changes reflect known changes of the properties of sensors in biological systems? Again, it is clear that not all of this can be addressed, but some assessment of how important and how realistic task specific changes in sensor properties and availability would help judging how realistic and how general purpose the model sensory system is.

3. It is mentioned that some actuators may be switched off in a task specific manner, such as leg actuators during flight tasks. Is it important to switch these off to maintain stable flight, e.g. because feedback to leg actuators would produce movements that would interfere with the flight task? If yes, it would be important to see systematic analyses of which actuators are switched off during which task. It would also be important to learn how joints and appendages are logged into position during specific tasks in biological systems, and how that relates to switching off actuators in the body model.

E. Conclusions: robustness, validity, reliability

1. The paper states that generality of the model is demonstrated by realistically simulating two forms of locomotion, flight in air and walking on land. However, the controllers seem to have a quite different design (wing beat generator plus trainable multi-layer perceptron for flight, versus no pattern generator but only a single multi-layer perceptron policy network for walking). Together with the large differences in sensory input for both tasks and the fact that the legs are fixed into one position during flight with disabled actuators and the wings for walking, models for flight and walking seem quite different. Consequently, the statement on model generality should be put into perspective accordingly.

2. The opening statement that the body determines how the nervous system controls behavior (abstract, first sentence) does not quite capture the core of the study. The way the NS controls behavior is indeed influenced by the geometries, weights, and movements of the body parts (which has been considered here) as well as their interaction with the physical environment (much of which has been considered here), but also by the material properties of the body parts, the properties and dimensions of the muscles, metabolic state, and numerous other parameters (which have not been considered here). Therefore, the promise that the embodied context provided by this study is key to understanding how the NS produces behavior seems an overstatement. It is one of many building blocks required to address this question in sufficient detail.

3. The discussion lists multiple ways the model can be built on in the future, but this list includes mainly anatomical additions, such as XNH X-ray imaging and confocal imaging of body parts to add further detail to the 67 rigid parts body model, recent mapping of proprioceptors to include more realistic positions of the sensors, or pose-tracking algorithms to retrieve more detailed kinematic information from high-speed video of flies. This may all help to further improve the existing model, but it will not bring an entirely different quality to the model. Incorporating the physiology of the neuromuscular and sensory systems would indeed cross a milestone.

4. Although the ms claims that future work can leverage connectomic mapping of the fly nervous system to model the neural circuits that control locomotion more realistically, the avenue to get there is only partially sketched. The artificial network remains a black box and would have to be entirely replaced by a connectome- and physiology constrained neural network model. How realistic is this? And how to make the biological neural network control the biomechanical body model? At least in biology, the fly neural controller does not need to learn to use the body for locomotion through the physical world. Upon using biological neural networks, is there still training envisioned, and if yes, what kind of training?

5. Often the text suggests that the model resembles the biology of the nervous system. For example, the training of 'steerable' low level controllers that are used to generalize from training trajectories to locomotion along novel trajectories are classified as analogous to the ventral nerve cord in that they are responsible for translating descending command signals to low level motor control signals (lines 160-165). It is correct that the biological system for sure uses descending commands for the decision to fly somewhere, and that the descending input is integrated with other information in ventral nerve cord networks. However, this does not suffice to state that the model low level controllers are analogous to the ventral nerve cord. It is not entirely clear whether all joint movements are driven by low level command signals, and most importantly, it is not clear whether VNC networks can be trained. Therefore, it is suggested to remove suggestive statements on possible analogies between the ANN and the biological networks (which have not been studied here) throughout the text. The study does not gain much from these statements unless they are further substantiated.

6. The discussion repeats what has been done and lists possible future directions that can build upon the model. However, it does not refer to any novel insight into the biology of animal locomotion. Any important novel insight on the sensorimotor control of locomotion that this study has provided must be explained in the discussion.

F. Suggested improvements: experiments, data for revisions

1. A first step toward identifying rules used by the black box ANN would be to compare different types of artificial neural networks in their ability to be trained for generating locomotor patterns. This could potentially help identifying specific network motives that are particularly important for successful ANN training. It would require thorough analyses of different types of ANNs trained to control the whole-body model to then distinguish network design principles that are particularly useful from network motives that are less important.

2. As mentioned above (section C3) additional experiments could give valuable information about the biology of locomotion control. First, manipulating the numbers of sensors required for adequate task learning. Second, quantifying the relative impact of the forces neglected in previous models (e.g. magnus force, Kutta lift). Third, testing, how complex the behavioral task can get to still be successfully trained, e.g. rapid saccades, climbing different slopes and obstacles.

G. References: appropriate credit to previous work?

1. The importance of Kutta and Magnus forces has been discussed by Sane (2003, JEB, 206 (23): 4191–4208). Can the importances of the inclusion of these forces be assessed?

2. The idea to use reduced approximate fluid dynamics simulations that are fast enough for repetitive evaluation in optimization procedures on the flight system has been proposed in Berman and Wang (2007, J. Fluid Mech 582:153-168). How does your phenomenological fluid model based on Anderson (2007) relate to what has been proposed by Berman and Wang?

3. Reid et al (2019) argues that wing flexibility plays a role in energetically optimal flight. This is not part of the proposed, posture-based biomechanical model. Some reference to wing flexibility and other material property-based mechanisms should be made and the expected consequences of omitting these known factors should be discussed.

4. The discussion should include some reference to other biological systems in which whole-body simulations are performed, such as worms and jellyfish.

5. Previous models often focused on one movement modality, for example Lobato-Rios for walking or Dickson 2008 for flight. The reason is that the two modes are very distinct in their kinematics and neural control. What is the exact merit of having all in one model and does this come with limitations? In particular, if the ranges and angles for flight and walking are both taken from a third motor behavior, grooming?

H. Clarity of content

The ms is clearly written and well presented. As mentioned above, similarities and differences between the model and biology should be pointed out more clearly throughout the ms.

(Remarks on code availability)

The code is a highly valuable and usable resource for the community. It is done highly professionally and indeed all open-source. We were able to install it, and we ran the "getting- started script" without any problems.

Version 1:

Reviewer comments:

Referee #1

(Remarks to the Author)

Major concerns

1. Simplification of the body model: Body actuators

1.1. Demonstrate the utility of the adhesion actuators to substantiate the impact of this contribution.

> Thank you for this suggestion. We have now added the results of a new task with a fly trained to walk on hilly terrain. We analyzed the resulting adhesion and contact forces generated while walking. Please see Extended Data Fig. 6.

Fantastic! The hilly terrain experiment and analysis in Extended Data Figure **7** perfectly addresses the concern. I particularly like the diagram in panel a, rollout traces in panel c, and the slipping analysis in f. Well done!

1.2. Discuss and/or show evidence to justify the use of position actuators over torque actuators.

> We clarify that the goal of our model is to accurately predict what forces the body must produce to generate realistic behavior, but not how these forces are generated. Adequate measurements do not yet exist across the entire body, of the muscles which drive movement and the proprioceptors which report sensory feedback. In the absence of these measurements, we cannot yet accurately model the detailed mechanism of force production. For this reason, we can only accurately model the forces that are produced by each body part, but not the muscle activations.

I appreciate the difficulty of modeling muscles and thank the authors for the response to concern 1.3, but here we're talking about position vs torque.

As I suggested in my original review:

> _The authors could address this limitation by (i) elaborating on their reason for this design choice (e.g., robustness to instabilities, tractability of optimization, etc.); and/or (ii) trying out torque actuators in a limited experiment to show that the differences are insignificant (or at least bounded) in some conditions. **In either case, an explicit discussion on the downstream implications of using this actuator type is essential.** _

To that end, I don't see any of the following pieces of information in the revised manuscript:

> Imitation learning can be used regardless of actuator type. These early results also suggest the possibility that position actuators lead to faster training and more accurate imitation (as also suggested by Chen, Zhang, Mueller, Rai, Sreenath arxiv 2022).

Thanks for providing this reference! It would make for an excellent citation **in the revised manuscript**, maybe around L104-106?

> However, we do not recommend interpreting the control signal sent to an actuator since a muscle is neither a position actuator nor a torque actuator, but rather only suggest interpreting the output torques produced by the actuator.

This is probably the MOST important statement to include, particularly for the biomechanically-uninformed masses such as this reviewer :)

> We now demonstrate through new experiments...

Thanks for providing this data. I would suggest including these experiments as another extended data figure as it is a tremendously relevant data point for practitioners that want to understand the trade-offs explored here.

In addition to their inclusion, I'm a little confused about the experimental conditions being evaluated here. There are two modeling choices being tested: (i) enabling/disabling unused DoFs, and (ii) torque vs position actuators. Could the authors disentangle these? If I'm a user that wants to achieve faster training, it's not clear whether I should disable legs/wing DoFs, switch to all position actuators, or both.

In the flight task (top row), what happens if I keep leg DoFs disabled but use position actuators for the wings? Or what if I keep leg DoFs enabled and use position actuators (for either? both?).

In walking (bottom row), what if wing DoFs are enabled but legs use position actuators? Does using position actuators for wings help to compensate for the discrepancy in the learning curves introduced by the new rewards term to keep the wings folded?

1.3. Discuss how big of a gap the current work leaves before muscle actuators can be introduced.

> Accurately incorporating muscle actuation involves three steps. [...] We clarify this in our discussion.

Fantastic! Thank you for the detailed description – this is exactly what I was looking for.

What I can't find is this information in the revised manuscript. I could only find:

> ...comprehensive measurements do not yet exist to faithfully model anatomically realistic muscle actuation... (L104-105)

But no description of what those measurements would be (as is detailed in the response to reviewers).

> We intend our open source project to serve as a platform which can be built upon in several ways. Confocal imaging (Extended Data Fig. 1), as well as micro CT [Lobato-Rios et al., 2022] and XNH [Kuan et al., 2020] imaging of the body support detailed musculo-skeletal measurements across the whole body which can be used to model anatomically detailed muscle actuation. This includes modeling of the neck [Gorko et al., 2024], wing hinge [Melis et al., 2024], and coxa [Kuan et al., 2020, Mamiya et al., 2023] joints. (L417-422)

This, again, does not address my concern about **clearly stating to the readers how big of a gap the current work leaves before muscle actuators can be introduced**.

It is also a bit at odds with the qualitative comparison presented in the review which seems to suggest that micro CT *cannot* "be used to model anatomically detailed muscle actuation".

2. Simplification of the body model: Wings

2.1. Describe to what extent these simplifications impact the biological validity and interpretability of the model.

> We do not intend our work to serve as a definitive analysis of the fluid physics involved in insect flight. [...] Our work is a necessary platform and essential first step for developing and modeling the same mechanistic understanding across the entire body. We clarify this in our introduction and discussion.

I think with the extended descriptions re: gap in biomechanical modeling requested for concern #1, this would be adequately addressed now.

2.2. As in (1.3), provide a gap analysis to justify [...] to what extent the wing model is limited by the fluid dynamics simulation.

> The fluid model was developed to be broadly useful for MuJoCo, and not just for our fly body model. It was developed for ellipsoids because we can compute an approximate estimation of the circulation around the object (and therefore the forces) based on the cited literature.

The revised discussion around L149-152 now addresses the concerns around using an ellipsoid geom and other approximations to the fluid interaction modeling.

3. Phenomenological fluid model for insect flight

3.1. Describe why the model parameters are entirely phenomenological and what potential alternatives might be.

> However, we now demonstrate that relative to the measurements currently available, our model is robust to the exact

choice of these parameters. [...] We now add a new analysis of the contribution of the various fluid forces in our model during free flight (see Fig 2j).

The analysis in Fig 2j is excellent, as well as the revised text that describes it. Thank you for including the sensitivity analysis in ED Fig 5 as well.

3.2. Describe how the fluid dynamics model interacts with properties of the wing biomechanical model.

> We thank you for this suggestion. We now discuss these issues in the manuscript.

Thank you for adding this discussion!

4. Reuse of imitation learned low level controller modules

4.1. Show that imitation learning combined with embodied NeuroAI enables estimation of sensory inputs and motor outputs during kinematic replay.

> Thank you for this suggestion. We now provide a new notebook in our github repo demonstrating this use case. And we now highlight the sensory variables predicted from imitation learning, which is quite analogous to kinematic replay, in Fig 2d, the visual inputs predicted in Fig 4, and the ground contact forces predicted in Extended Data Fig 6.

Beautiful!

4.2. Show that imitation learning facilitates reusability of motor skills in goal-driven tasks.

> We thank you for this excellent suggestion. We have now replaced the controllers trained from scratch for the visually-guided flight tasks (bumps, trench) with a high-level controller which is trained to reuse the pre-trained low-level controller trained using imitation. And we now release code for the reuse of both the flight and walking controllers. These changes can be seen in Fig 4 and the "Vision-guided flight with flight controller reuse" section.

Perfect!

Minor concerns

5. Reference motion capture data: Walking

6. Reference motion capture data: Flight

Both of these concerns are around the availability and use of reference 3D MoCap data.

I am not sure I entirely agree with the justifications presented here, but since properly testing the validity and accuracy of the 3D tracking is a substantial endeavor and since it is a minor concern, I feel that it does not impact the findings or soundness of the main claims of the paper.

7. Overstatement of potential short-term capabilities

> We have now removed the phrase "short-term" and more clearly detailed our vision of the next steps for how our model can be used.

Thank you. The revised discussion I think does a good job of articulating the present capabilities of this work.

Accountability

This review was written by Talmo Pereira (talmo@salk.edu) who opts to forsake anonymity in favor of ethical reviewing through accountability and transparency.

(Remarks on code availability)

Referee #2

(Remarks to the Author)

A. Summary of key results:

The manuscript aims at providing a tool for future work on the integration of models of the muscular and neural control of sensorimotor behavior in an embodied context, i.e. a biomechanical model of the body of a fruit fly and its interaction with the surrounding physical world. The whole-body model is implemented in the open source MuJoCo (Multi-Joint dynamics with Contact) physics engine and elegantly incorporates realistic information on the geometry, weight, and degrees of freedom of 67 body parts as well as geometry based (no muscle properties, no material properties) interactions of the 67-parts-body with each other and with an approximation of the physical environment during locomotion. Based on the assumption of uniform material of all body parts MuJoCo estimates the body part moments of inertia, and actuators are added to control

joint movements and positions. Interactions with the physical world are approximated by a phenomenological fluid model that generalizes and extends previous work, but does not account for turbulences. The body model is then equipped with idealized (non-biological) sensorimotor and visual feedback controllers that can be trained to mimic locomotor movements as filmed in real flies. An artificial neural network (ANN) receives MuJoCo simulated sensory input and is trained by reinforcement learning in closed loop to drive the actuators in the body model. The main finding is that the highly flexible artificial neural network can indeed be trained to produce the simulation of locomotor movements that closely resemble basic walking and flight tasks as observed macroscopically in the biological system. As it stands the body model may be of great utility for robotics and biomechanics research, but its applicability in research on neuromuscular function and the neural control of movement is not clear.

B. Originality and significance:

The model is done on a high technical level and can be used near real-time, the code is well documented and freely available. Additionally, forces like the Kutta lift and the Magnus effect that have not been part of previous models have now been implemented and further analyzed in an embodied context. It is claimed but not demonstrated that additional information on muscles and sensory biology can be incorporated in the future, so that the model is general purpose and extensible. Therefore, in the long run, if smartly paired with other approaches, data, experiments, and significant changes to the model, this could really help to gain novel insight into the control of sensorimotor behavior. However, as it stands the study does neither provide novel insight into the neural control of sensorimotor behavior as the model lacks biological neural components, muscles, material properties, and sensors, nor does it rigorously test the limits of the whole-body model. The latter might be a problem, because it remains unclear whether some design choices decrease the power of the model for studying the neural control of locomotion. In sum, the generation of the fine biomechanical model and the accompanying proof of principle that it can be used in a simulation to train an artificial neural network to mimic movements as macroscopically observed during basic aspects of locomotion represent strong methodological advances but little advances to our understanding of the control of locomotion.

C. Data & methodology: validity of approach, quality of data, quality of presentation:

The data presented are high quality and the model is finest state of the art. However, as it stands this approach does not seem to provide novel key biological insight into the neural control of sensorimotor behavior. The 3 points that I had raised in this context have not been addressed satisfactorily. In fact, the authors clearly state in their rebuttal that investigating the neural basis of locomotion is beyond the scope of the study, yet, the text still strongly suggests novel insight into the neural basis of locomotion. One text example is "detailed modeling of the sensorimotor control of behavior requires a detailed model of the body. Here we contribute a body model.....enabling the simulation of diverse fly behaviors (abstract). For a broad audience this implies novel insight into the neural basis of locomotion. There are many such statements throughout the text. These are all disconnected from the point that understanding the neural basis of behavior is beyond the scope of this study.

Similarly, the rebuttal letter states that the authors believe that this model will serve as a platform for all future studies modeling the generation of fly behavior. The main text is a good reflection of this belief, but should really be down-toned throughout for the following reason. First, the fact that an ANN can be trained to produce movement patterns as seen in animals does not mean that it employs similar rules that are used by the biological system. There might be numerous different solutions to the problem. Second, the whole-body model contains numerous simplifications. The resulting concern is that the ANN can learn to realistically simulate movement of a biomechanical whole-body model in a virtual physical environment, but the demands to the neural controller are so different from the real biology that one does not learn much about the nature of the biological neural networks that normally control locomotion. For example, the demands to neural output are significantly affected by muscle and exoskeletal properties that are not included in the model and that cannot be modelled by position actuators (see original review for more detail). Such obvious deviations from the biology of the system are neither acknowledged nor discussed, but a critical discussion would be important for adequate use in future models. Third, even without more realistic biology the model lacks some physical forces that have been shown to be important in insect flight (e.g. turbulences are not adequately represented in the model). And finally, more realistic biological properties of muscles and exoskeletal elements may reveal forces that are not yet considered in the model (e.g. the elastic bending of real wings). In sum, the claim that this one model will be the basis for all future modeling studies seems largely overstated. Although it is clear that not all of these details can be implemented in the body model, a critical evaluation of the potential benefits of implementing more detailed biology or of the current model's limits is required.

D. Appropriate use of statistics and treatment of uncertainties

The statistics used seem state of the art. However, in the model large parts of the biology have not been incorporated. Therefore, remaining uncertainties are the nature of the approach and must be accepted. However, these should be spelled out more clearly.

1. The previous suggestion to include more detailed information on how well the model sensory system matches the real sensory system has been dismissed with the argumentation that the goal was to model what quantities are sensed by the fruit fly and not how these are sensed. But there are no analyses of what quantities the fruit fly senses. Some assessment of the mechanosensory experiences of the model would have been informative.

2. There is still no information on whether different tasks require different sets of sensors or different sensor properties, or whether this has been systematically analyzed (not just recorded). No further data or experiments have been added. It is stated that this is discussed in the revised manuscript, but it remains unclear where and how this is discussed.

3. The figure that has been added to the rebuttal letter contains highly informative new analyses on model performance with modified actuator properties and on the potential necessity of task-specific actuator use. However, the figure is difficult to read (many overlapping solid and dotted lines), and neither the figure nor the information have been included in the revised ms.

E. Conclusions: robustness, validity, reliability:

1. Additional data showing convincingly that a single common model setting can reproduce walking and flight has been generated. This nicely underscores the generality of the model in reproducing different types of movements as observed on the macroscopic level in high-speed videos of walking and flying *Drosophila*. It should still be pointed out with clarity to the reader that it remains unclear how these movements are produced, i.e. the neural and biomechanical mechanisms that produce these movements remains unclear, and many different mechanisms can produce the same movement. For example, an extension of a leg joint can be produced by the contraction of an extensor muscle or by a combination of elastic exoskeletal properties and flexor muscle contraction. Of course, it is exciting to analyze which fluid dynamic forces act upon the body during flight and walking movements, but novel key insight into motor control would require also how these forces are generated. Therefore, it has to be clarified more explicitly throughout the ms that how the body generates forces is not addressed, or the ms must be targeted to a more specialized audience.

2. First sentence of the abstract has been corrected to 'the body influences how the nervous system controls behavior'. Though correct it remains misleading because it suggests that the manuscript presents novel data on the nervous system, which is not the case. Moreover, the start of the introduction nullifies the correction made in the abstract by stating that the body dictates precisely how motor commands from the nervous system are translated into action, and that this all is captured by the model. It must be clear to the reader from the beginning on that neither muscle nor exoskeletal properties are part of the 'detailed biomechanical model'.

3. The discussion remains unclear when it comes to future research that can build on the model. Instead of critically dissecting what the model is most useful for and what its current limitations are, it is stated that this work lays the foundation for studying the basis of sensorimotor behavior in the adult fruit fly. The power of the model for understanding nearly everything from brain and muscle function to body properties and biomechanics to animal interactions with the physical world seems largely overstated.

4. Although the ms claims that future work can leverage connectomic mapping of the fly nervous system to model the neural circuits that control locomotion more realistically, the avenue to get there remains sketchy, without a critical discussion of the anticipated challenges when integrating connectomes.

5. As in the original version, the text in the revised ms suggests that the model resembles nervous system biology. Given that the ms and model are targeted not only to neuroscientists, but also to computer scientists and engineers, as stated in the response letter, it seems particularly important to remove suggestive statements on neuroscience that are misleading to a broad readership. The examples from the original review are not repeated here, though not resolved. As the model stands, it may have fantastic potential for future research, but it really does not provide any novel insight into nervous system function. To not mislead computer scientists and engineers, or disappoint neuroscientists, this must be made absolutely clear to the reader.

F. Suggested improvements: experiments, data for revisions:

Some of the previously suggested experiments for revisions have been done and provide novel insight, but others were not done.

1. A possible suggestion toward identifying rules used by the black box ANN was to compare different types of artificial neural networks in their ability to be trained for generating locomotor patterns. This suggestion was judged to be out of scope for this study and would indeed require substantial additional effort. However, without additional analyses it must be clearly pointed out in the text that the current analyses do not provide novel information on the rules used by the ANN, on the usefulness of specific network design principles best suited this task, or on the biology of the real flight system.

2. Additional experiments were suggested to provide information about the biology of locomotion control. These included manipulating the numbers of sensors required for adequate task learning as well as the relative impact of the forces neglected in previous models (e.g. Magnus force, Kutta lift). The number of sensors were not varied, but the contribution of various fluid forces were analyzed and presented in Fig. 2J. This provides valuable new insight on the forces, but there is still little information on the importance of numbers, types, and properties of sensors for adequate task learning. The suggestion of testing how complex the behavioral task can get to still be successfully trained (e.g rapid saccades, climbing different slopes and obstacles) has been partially addressed by training a fly to walk on a hilly terrain, but the limits of what the model can be trained on have not been analyzed in detail.

G. References: appropriate credit to previous work?:

All previous suggestions have been incorporated in the revised manuscript

Other points:

- please clarify what *Drosophila* species has been used for table 1

- please provide standard deviation for values in table 1
- delete 'to' in line 148
- delete 'in' in line 150
- please carefully double check the text for typos

(Remarks on code availability)

The code is a highly valuable and usable resource for the community. It is done highly professionally and indeed all open source. We were able to install it and run the getting started skript without problems.

Srinivas C Turaga, PhD
Group Leader
19700 Helix Drive
Ashburn, VA 20147
+1 (339) 221-1958

Jul 30, 2024

Dear Editor,

We thank you and the reviewers for your thoughtful and detailed comments on Nature manuscript 2024-04-06671 by Vaxenburg et al, titled “Whole-body simulation of realistic fruit fly locomotion with deep reinforcement learning”. Please find attached our revision of this paper, which I hope you will agree is now substantially strengthened in response to excellent suggestions from the reviewers.

Our paper describes a new general-purpose physics simulator of fly behavior enabled by the development of a new anatomically detailed body model, physics simulation of contact forces and fluid forces, and an imitation learning framework capable of learning closed-loop sensorimotor control of realistic locomotor behavior both walking on land and flight in the air. Below we first summarize a few general points and then give a detailed response to the reviewers.

Several comments by the reviewers touch on the biological realism in our model of how the body produces forces (actuation) and in the sensory system. We clarify that comprehensive measurements across the entire body do not yet exist to accurately model muscle actuation and the sensory transduction. Therefore, we only accurately model *what* forces are generated by the body in interaction with its environment, but not *how* these forces are generated by the body through a combination of active (muscle) and passive (biomechanical) mechanisms. Similarly, we only model idealized sensors which report *what* quantities the body can measure, but not *how* these quantities are measured and reported by sensory neurons. Comprehensively and accurately modeling this interface between the body and the nervous system is exciting future work, which awaits new measurements and in many cases also new measurement technology. We also now clarify this point throughout our manuscript.

In response to excellent suggestions by the reviewers, our revision makes the following major changes:

1. Expanded presentation and discussion of the fluid model along with analysis of fluid forces and a stability analysis of model parameters.

2. Similarly, a new simulation and analysis of adhesion and ground reaction forces.
3. Demonstration of interchangeability of torque vs position actuation.
4. Demonstration of universal imitation learning without task-specific specification of controllable degrees of freedom or controller architectures.
5. Demonstration of reuse of low-level controllers for visually guided navigation.

Sincerely, and on behalf of all authors,

Srinivas C Turaga

Referee #1:

Meta-note re: Melis et al. (2024) and NeuroMechFly

I understand that it is a matter of policy that preprints are not taken into account when making publishing decisions in Nature Portfolio journals. Melis et al. was preprinted on bioRxiv on July 2, 2023 and published in Nature on April 17, 2024, while this article was received by this reviewer on April 12, 2024, suggesting that it was submitted for review prior to the publication of Melis et al. Of note, the two papers share a co-author (Igor Siwanowicz) who made significant contributions to both. I just want to state these facts for posterity and for consideration of how the review comments below should be weighed since many of the points hinge (;-)] on the similarities and differences between these works. I expect that the overall review stands, regardless of how Melis et al. is factored into the review process (if at all), but I chose to include these comments since they are too essential to the assessment of the science, independent of journal editorial policy.

Similar caveats could be attributed to comparisons to NeuroMechFly since the newest version that added adhesion for terrestrial locomotion (and other features also presented here) is still in preprint stage (Wang-Chen et al., 2023). This is partially mitigated by the fact that the previous iteration was already published (Lobato-Rios et al, 2022).

Here we clarify the relationship of our paper to the work recently described in Melis et al 2024 and Wang-Chen et al 2023. Our work is highly novel and adds significantly to previous work described in Wang-Chen 2023 and Melis 2024.

Relationship to Melis et al 2024

Our work is complementary. Melis 2024 focuses on measuring and modeling muscle actuation of the wing hinge during flight. It describes exciting new measurements of muscle activation during tethered flight which are then used to model muscle actuation of the wing hinge during free flight trajectories. It is worth pointing out that this model can only simulate flight behaviors as they only model the wings.

In contrast, our anatomically detailed body model is a general-purpose model capable of the broadest range of behaviors including walking, flight, grooming. However, as we clarify, we do not yet model muscle actuation. Data from Melis 2024 can be used to add wing muscles to a future version of our model which includes detailed muscle actuation. Further, we contribute an imitation learning framework and use it to build steerable

controllers for flight and walking in a data-driven manner. In contrast, Melis 2024 only uses model-predictive control to generate pre-determined flight trajectories and does not have a steerable sensorimotor controller.

Relationship to Wang-Chen et al 2023 and Lobato-Rios et al 2022

NeuroMechFly (Lobato-Rios et al 2022) and NeuroMechFly v2 (Wang-Chen et al 2023) report an anatomically detailed body based on microCT, first in the PyBullet simulator (NeuroMechFly) and more recently in MuJoCo (NeuroMechFly v2). In both papers, they only report unrealistic heuristic hand-designed low-level walking controllers. In Wang-Chen 2023, a combination of reinforcement learning and hand-design is used to create a variety of high-level controllers to perform a variety of sensory-guided tasks including olfaction and vision.

In our work, we have independently constructed an equally detailed new body model in MuJoCo, based on a new confocal microscopy based technique we developed for imaging the body anatomy. This model is capable of simulating a much broader repertoire of fly behaviors thanks to our new fluid simulation. Our model can both walk on the ground and fly in the air. Further, we contribute a data-driven imitation learning framework for training low-level steerable controllers and demonstrate that it produces biologically realistic walking and flight along arbitrary trajectories and across the full range of speeds. Our physics simulation is therefore capable of a superset of the behaviors produced by NeuroMechFly and our imitation learning enables these behaviors in a more realistic manner than the hand-designed low-level gait controllers used by NeuroMechFly.

We would like to highlight here how our open science policies have supported the work described Wang-Chen 2023. The pre-publication open-source release of both code and models by our team were essential for two of the central results described in their paper. The adhesion actuator reported in their paper was actually developed by our team for this project, and open-sourced as part of MuJoCo prior to our publication. A second main contribution reported in their paper is the integration of their body model with a connectome-constrained model network model of the fly visual system. This model was developed by my lab and open-sourced alongside Lappalainen et al biorxiv 2023 and now published in Lappalainen et al Nature 2024. Therefore our work should be seen as synergistic to the work described in Wang-Chen 2023, with our pre-publication release enabling exciting cross-pollination.

Major concerns

1. Simplification of the body model: Body actuators

The body model (other than the wing, discussed in point 2 below) largely uses position and leg adhesion actuators. This is a significant design choice since it simplifies practical optimization and body model development, but comes with trade-offs regarding the (meaningful) sacrifices in biological fidelity.

This reviewer deeply appreciates the difficulty and complexity of building a body model that incorporates muscles and recognizes the value of representational abstraction. That said, the use of position actuators constrains the potential of this work for immediate use. This becomes relevant for mechanistic studies in fields like motor neuroscience where accurate representation of muscles is essential for establishing homology between model and biological system components, and where the neural control signals (which vary drastically based on actuator choice) are the objects of interest.

While this limits the applicability of this work in neuroscience, it can still be useful for biomechanics research interested in grosser level properties such as kinematics. To this end, and to address other concerns:

See detailed responses below.

1.1. Demonstrate the utility of the adhesion actuators to substantiate the impact of this contribution.

In the manuscript, there is a statement that "However, we observed that the agent generally learned to activate adhesion when legs were in stance (on the ground)" (L281-283), but I don't see any data on this. Showing that this is the case during different phases and gait modes of locomotion would help to show that this addition to the model is useful as it enables estimation of a modality of motor control that is difficult to measure in vivo during free behavior, and can be compared to previous work on the role of adhesion during locomotion (Ramdya et al., 2017).

Another way to show the significance of this contribution would be to show that it enables complex 3D navigation. Right now this is only demonstrated qualitatively in Fig.

1h. Showing this in action in a task (imitation or goal-driven) that would be challenging without adhesion actuators would prove the point, and is achievable by devising an environment that features navigation across hilly terrains (as in Merel et al., 2019; Wang-Chen et al., 2023) or steep surfaces (Ramdya et al., 2017).

(A task that uses the adhesion actuator added to the labrum would also enhance the significance of this contribution.)

Thank you for this suggestion. We have now added the results of a new task with a fly trained to walk on hilly terrain. We analyzed the resulting adhesion and contact forces generated while walking. Please see Extended Data Fig. 6.

1.2. Discuss and/or show evidence to justify the use of position actuators over torque actuators.

Position control less accurately reflects the way muscles generate forces and control movement in biological systems. As mentioned, this may be an acceptable limitation in some cases but not others. Torque-based actuators better mimic the way that muscles generate forces and influence joint movement, providing a more biomechanically accurate representation of muscle function. Given that torque actuators are used in other parts of the body model (e.g., wing hinge), it stands to reason that they could be used to model the other joints as well.

The authors could address this limitation by (i) elaborating on their reason for this design choice (e.g., robustness to instabilities, tractability of optimization, etc.); and/or (ii) trying out torque actuators in a limited experiment to show that the differences are insignificant (or at least bounded) in some conditions. In either case, an explicit discussion on the downstream implications of using this actuator type is essential.

We clarify that the goal of our model is to accurately predict **what** forces the body must produce to generate realistic behavior, but not **how** these forces are generated. Adequate measurements do not yet exist across the entire body, of the muscles which drive movement and the proprioceptors which report sensory feedback. In the absence of these measurements, we cannot yet accurately model the detailed mechanism of force production. For this reason, we can only accurately model the forces that are produced by each body part, but not the muscle activations. Whether these forces are produced internally by a position actuator or a torque actuator is merely a convenience

for the machine learning. We have now made it possible for a user to choose whether they wish to use position or torque actuators through a configuration option. However, we do not recommend interpreting the control signal sent to an actuator since a muscle is neither a position actuator nor a torque actuator, but rather only suggest interpreting the output torques produced by the actuator. We now demonstrate through new experiments two points:

1. Imitation learning can be used regardless of actuator type. These early results also suggest the possibility that position actuators lead to faster training and more accurate imitation (as also suggested by Chen, Zhang, Mueller, Rai, Sreenath arxiv 2022).
2. Relatedly (addressing points by both reviewers below), while leg degrees of freedom can be frozen during flight and wing DoFs frozen during walking, this is again merely a convenience but not strictly required. A single general-purpose model parametrization with all degrees of freedom and sensors enabled can also be trained successfully to produce all behaviors, both walking and flight.

Figure. Comparison between the performance of the original fly model and the modified model (all DoFs enabled, torque actuators replacing position actuators) trained on flight (top row) and walking (bottom row) imitation tasks. (a, b) Percentiles of CoM and orientation quaternion tracking errors in the flight imitation task. Solid lines: original model, dotted lines: modified model. (c) Flight imitation learning curve comparison: average episode returns vs actor steps during training. (d, e, f) Same as (a, b, c) but for walking imitation. In (f), an additional reward term is required to keep the (now enabled) wing DoFs in folded position and it causes most of the discrepancy between the two learning curves. This reward term is only approximately satisfied, causing a reduction of the episode return by a factor of ~ 0.69 in the trained model.

1.3. Discuss how big of a gap the current work leaves before muscle actuators can be introduced.

The authors state on L76-79: "The dataset also enables the identification of origin and insertion sites of muscles, locations of the proprioceptive hair plates of the neck, coxae, trochanters, wing base and halteres — information that can be incorporated into future versions of the model."

Since the absence of muscles is a major limitation of this work, providing an estimate of the size of the gap between the current model and one that incorporates these improvements will be helpful to readers that are unfamiliar with the field to gauge the significance of the current work in a broader context.

A major hurdle overcome by this work (and a major contribution) is that it shows the feasibility of reconstructing the body model from confocal imaging data, a modality of microscopy that is much more accessible than some alternatives like XNH or EM.

If the current dataset is of sufficient resolution to recover details of musculoskeletal anatomy necessary to develop muscle-based actuators, then what else is missing? Image segmentation of morphological features of interest? Manual work in converting these data into MuJoCo-based muscle actuator parameters?

It would also be useful to state how the selected imaging modality compares to micro CT.

Accurately incorporating muscle actuation involves three steps. First, identification of all the muscles and their insertion sites, revealing the degrees of freedom actuated by each muscle. Second, each muscle must be modeled in MuJoCo, yielding anatomically correct muscle-actuation across the body. This modeling step is non-trivial and requires some experimentation to determine the best way to implement complex muscle and

tendon wrapping that can be seen in the anatomy with the limited modeling capabilities of MuJoCo. In some cases, such as the wing hinge and neck joints which are simplified in our model and are challenging to implement in full detail, virtual muscles might be needed to map from muscle activations directly to torques on a ball joint. Third, system identification must be performed to approximately constrain the dynamics of each muscle across the entire body. Ideally, this would involve correlated measurements of muscle activity and kinematics, as in Melis et al 2024. As a first step, however, system identification of muscle parameters could be performed by inverse dynamics/imitation, as we have done for the non-muscle actuators used in this work. The work described in Melis et al 2024 indeed points the way to the next steps for virtual muscles, however completing this across the entire body with explicit muscles is a significant undertaking.

We clarify this in our discussion.

Finally, here is a qualitative comparison of our confocal microscopy protocol with micro CT imaging of the fly. The superior resolution and staining of confocal microscopy significantly aids the identification of musculoskeletal elements.

(Left) Partial projection through a confocal stack showing morphology of the drosophila hind leg joints (yellow arrowheads) and tendons. (Right) Partial projection through a stack generated from a μ CT dataset. Positions of the joints between the leg segments, as well as tendons, could not be unambiguously determined.

2. Simplification of the body model: Wings

In this work, the authors model the wing as a slender ellipsoid controlled by a ball joint driven by torque-based actuators. This reviewer appreciates the morphological and biomechanical complexity of the wing and hinge, but given their importance to flight behavior, I believe it is essential to characterize to what extent the simplifications employed here impinge upon the utility of the model for studying insect flight.

See detailed responses below.

2.1. Describe to what extent these simplifications impact the biological validity and interpretability of the model.

The concerns raised in (1) about the utility of this model for neuroscience in the absence of muscles in general are still applicable here and should be discussed in the context of flight.

Next, caveats about timing/journal policy aside, it is hard to overlook the fact that Melis et al. (2024) takes a similar approach to modeling flight biomechanics in *D. melanogaster*, but in deciding to more accurately model the wing hinge, indirect flight muscles, and sclerites, discovers mechanistic roles and functional consequences of these components.

Given this knowledge, or just drawing from the long history of work on insect flight (Boettiger and Furshpan, 1952; Pringle, 1957; Miyan and Ewing, 1985; Ennos et al., 1987; Dickinson and Tu, 1997; etc.), a discussion of how the design choices of the wing model in the current work impacts the biological or biomechanical validity of flight behavior is highly warranted.

Specifically, a description (or, preferably, a demonstration) of how the current wing model could be used to generate empirically testable hypotheses would go a long way towards providing evidence for its utility, and provide a path towards measuring validity.

To be explicit: Figs. 2 and 5 start going towards this, but do not explicitly demonstrate applicability towards empirically testable hypothesis generation, nor do they emphasize the design choices made in the biomechanical model of the wing without conflation with the control model.

We have developed a model which can accurately simulate the broadest range of fruit fly behavior in a single (necessarily approximate) physics simulation, which is fast enough to enable reinforcement learning. To this end, it is necessary that our model contain all the body parts and degrees of freedom, as well all the physics necessary, to simulate a fly that can walk, fly, groom, etc.

We do not intend our work to serve as a definitive analysis of the fluid physics involved in insect flight. We therefore developed a polynomial approximation of the true fluid dynamics that approximates the forces generated sufficiently well to accurately replicate the wing and body kinematics across a wide range of flight maneuvers. This, we demonstrated by modeling extreme evasion maneuvers.

Even without muscle actuation, our existing model can be used for neuroscience research in many ways including, for instance:

1. Simulating visual inputs during flight and the resulting neural signal generated by the optic lobes.
2. Modeling the neural basis of visually guided flight maneuver decision making.
3. Modeling muscle basis of flight by using the same virtual muscle model used in Melis et al 2024 – using their black box neural network to predict torques resulting from muscle activation, without explicit physics modeling of muscles.

As you point out, the work of Melis et al 2024 revealed the mechanistic basis for muscle actuation of the wing. We can also incorporate these measurements, in concert with further detailed musculoskeletal modeling of the wing hinge mechanism to build explicit physics modeling of muscle actuation. Our work is a necessary platform and essential first step for developing and modeling the same mechanistic understanding across the entire body.

We clarify this in our introduction and discussion.

2.2. As in (1.3), provide a gap analysis to justify and reinforce the significance of the contributions.

In addition to describing what is needed to accurately reconstruct the wing and hinge (is confocal microscopy, and this dataset specifically, sufficient?), it would be helpful to understand to what extent the wing model is limited by the fluid dynamics simulation.

For example, is it a hard requirement to use an ellipsoid geom to represent the wing due to how the fluid simulation is implemented, or could other geoms be employed? Is this a

fundamental limitation of the fluid algorithms, requiring new math to describe interactions with other shapes? Or is this a tractability issue, where other geoms are simply too slow to compute against (i.e., could be solved by new approximation algorithms or more compute resources) or too inaccurate/unstable (i.e., could be solved by more numerically precise implementations)?

I'll also point out that the reported performance estimates of the physics step (55.5 ms for 10 ms of simulation on a single CPU core; L149) would indicate that there is plenty of headroom for a more faithful wing model – assuming that simulation speed is (one of) the constraints.

Describing (and ideally providing evidence, theoretical or empirical) for the need for this simplification would be helpful to contextualize the significance of this contribution.

The fluid model was developed to be broadly useful for MuJoCo, and not just for our fly body model. It was developed for ellipsoids because we can compute an approximate estimation of the circulation around the object (and therefore the forces) based on the cited literature.

In principle, a purpose-built model for the wing could be constructed with effective fluid forces represented by:

- drag (which opposes the velocity of the plate on the plane)
- lift (which is normal to drag)
- rotational drag

Experimental measurements or numerical simulations providing the correlation between lift/drag and angle of attack drag and lift could be used to determine the parameters of this model. For instance, see Brunton & Rowley AIAA 2012.

3. Phenomenological fluid model for insect flight

I will preface this section with a disclaimer that I am not, by any stretch of the imagination, an expert in fluid mechanics or its simulation. That said, allow me to restate the contributions introduced here as far as I've come to understand them after extensive background reading:

- Added mass: The added mass accounts for the inertia of the surrounding air during wing movements. This is important if we want to estimate energetics and forces needed to displace air around the wing.

- Viscous drag: This captures the forces that oppose wing motion and depends on factors including wing velocity, shape and air viscosity. This component is essential for modeling energy dissipation and is most important at high Reynolds numbers (fast moving objects) such as during fast flight.
- Viscous resistance (aka linear drag): Similar to viscous drag, but better captures models interactions at low Reynolds numbers (slow moving objects), such as during slow/precise flight maneuvers and hovering.
- Magnus force: Flies rotate/flap their wings while moving forward to generate a Magnus force perpendicular to the wing's path. This force comes into play in lift generation and turning maneuvers.
- Kutta lift: The Kutta condition accounts for the lift generated by the circulation of air around the fly's wings, especially at the leading edge. It is the primary(?) generator of lift forces in Drosophila flight.

As disclaimed, I am deeply uneducated in the nuances of aerodynamics or fluid mechanics models, but I feel that I may not be alone in that given the target audience for this work. Given that that's the case, additional clarity on the following concerns may help to emphasize the importance of the contributions presented in this work to a broader audience.

Thank you for this suggestion and your excellent description of the components of our phenomenological fluid model. We have now significantly expanded our presentation of the fluid model.

3.1. Describe why the model parameters are entirely phenomenological and what potential alternatives might be.

The authors describe the math and physics behind each of the fluid model components in great detail, many of which have been objects of study dating back centuries. Given the elegance of this physical modeling, it is a bit disconcerting to then see that the models are then parametrized by a set of suspiciously exact coefficients (`fluidcoef = [1.0, 0.5, 1.5, 1.7, 1.0]` (L562)) that seem to have been hand-tuned until the fly was able to hover in place.

This is further compounded by having a hand-engineered model of wingbeat generation that controls a simplified model of the wing through torque based actuation. The wingbeat generation model, while derived from data, is already a phenomenological simplification of the true biological control mechanism, and issues with the body model and actuators are discussed in (2). So now we have three complex interacting

components with unknown degrees of inaccuracy, all being used in concert to derive physical constants in a seemingly guess-and-check approach?

I hope that my domain ignorance here is biasing my reading of this procedure, but I would encourage the authors to dedicate some text to justifying why this is a reasonable thing to do. For example:

3.1.1. Why can't the `fluidcoef` parameters be inferred from known physical constants or derived from empirical data in other systems outside of insect flight? Surely common physical properties like air viscosity have been measured in other settings?

3.1.2. To what extent are the choices made in deriving this phenomenological model a constraint of the generalization of the Andersen et al. (2005) approximation model (a key contribution of this work)? As this new fluid model is a novelty of this work, it seems within scope to describe its limitations (e.g., it may require phenomenological estimation of key constants due to change of units or simplifications introduced in the approximation).

Unfortunately, it is not possible, with the experimental measurements available to us, to determine the coefficients associated with the components of our phenomenological fluid model. Indeed we tuned the parameters to enable the fly to hover in place. Beyond this level of tuning which gets us into the right ballpark, it is not possible to significantly refine our parameter estimates with the available data. However, we now demonstrate that relative to the measurements currently available, our model is robust to the exact choice of these parameters.

We now add a new analysis of the contribution of the various fluid forces in our model during free flight (see Fig 2j). We show that two components dominate: viscous drag and kutta lift. The other forces are 1-2 orders of magnitude smaller and so do not contribute significantly. We therefore conducted a sensitivity analysis of the coefficients associated with these two components and found that flight performance is robust to even 20% variation in these parameters.

In detail, we re-trained imitation learning of free flight with modified choices for the coefficients associated with viscous drag and Kutta lift, with all other coefficients held fixed. We then evaluated the degree to which imitation learning was able to correctly reproduce ground truth center of mass flight trajectories with realistic wing kinematics, as in the original experiments reported in the paper. We also quantified the fraction of trajectories where the fly crashed to the ground as a second performance measure.

We found that these two parameters could be varied by up to 20% without significantly degrading performance. It's worth pointing out that there is individual biological variation in flight behavior as well as in body mass, shape and size. We have also added this discussion to the paper. Further, as you note below, we are comparing experimentally measured free flight trajectories from *D hydei* with our model of *D melanogaster*. Therefore it is not possible to use these measurements to further constrain the parameters of our physics simulation

Figure. Sensitivity analysis of the dimensionless fluid model scaling coefficients for viscous drag (a) and Kutta lift (b) components. Accuracy of imitation learning is shown in blue violins (body center of mass error, 5 training runs, same trajectories as in manuscript) and trajectory failure rate in orange bars (fraction of trajectories which crashed) for simulations varying each coefficient by up to 10x.

3.2. Describe how the fluid dynamics model interacts with properties of the wing biomechanical model.

Since these are both novel contributions of this work, it would be instructive to domain-ignorant readers such as myself to understand how the modeling choices in the

physics affect how I can use the biomechanical model and what uncertainties they introduce.

As concrete (but possible naive) examples:

3.2.1. How are the added mass and viscous drag components affected by using a simplified wing model? Does the absence of realistic sclerite interactions and muscles affect the accuracy of how these components impact estimates of energy or forces? Do these inaccuracies manifest only at high Reynolds number conditions? What properties of the resulting kinematics are affected (e.g., velocity? acceleration?)?

3.2.2. How are the Magnus force and Kutta lift components affected by using a simplified wing model? Are the ball joint/torque actuators sufficient to capture the realistic dynamics of wing rotation and angle of attack in a way that impacts calculation of lift forces?

It would be especially informative to understand how the answers to these questions apply to low/high Reynolds numbers conditions, such as during hovering versus extreme maneuvers.

We thank you for this suggestion. We now discuss these issues in the manuscript. We also clarify that our goal for this paper is to accurately model **what** forces must be generated by the wings in order for realistic wing movements to result in realistic body movement through air. However, we do not yet model **how** these forces are generated by the body, through a combination of passive (detailed wing, wing hinge biomechanics + interactions with fluid) and active (muscle) forces.

4. Reuse of imitation learned low level controller modules

As an exemplar of combining embodiment with NeuroAI, this work leverages imitation learning (via DRL) to train neural network controllers to reproduce the kinematics observed during real behavior using motion capture.

Specifically, the authors train steerable low-level controllers (LLCs) which are intended to translate intention into motor commands and can be thought of as homologous to the ventral nerve cord (VNC) in flies (L163-165).

As these modules are trained on motion capture data, they enable alignment to biology which in turn has the potential to improve the chances that learned artificial neural controllers are consistent with those afforded to biological controllers through billions of

years of evolution. This is, in fact, the thesis and motivation behind much of this work, as cited throughout the text via references to Zador et al. (2023).

Despite this motivation and potential major strength of this work, the authors do not demonstrate how the trained models can be reused for anything other than kinematic replay. This is a major missed opportunity to demonstrate the utility of imitation learning in realizing the purported advantages of embodied NeuroAI.

I appreciate that the emphasis of this work is on the development of the body model, but a more convincing demonstration of the capabilities enabled by accurate biomechanical modeling would go a long way towards providing evidence for the significance of this work.

This could be achieved by, for example:

4.1. Show that imitation learning combined with embodied NeuroAI enables estimation of sensory inputs and motor outputs during kinematic replay.

A major advantage of a physics simulator is that we can infer the sensory inputs, including perceived forces and visual inputs, that are useful to neuroscientists interested in studying the neural basis of visually-guided behavior, proprioception, mechanosensation, feedback, and etc. Inference of muscle activations is limited by the choice of actuators, but even the more abstract action space may contain relevant information about the manifold of low level motor control signals.

Showing that you can estimate these parameters during kinematic replay demonstrates a key use case of the model and framework that would not have been possible otherwise.

Thank you for this suggestion. We now provide a new notebook in our github repo demonstrating this use case. And we now highlight the sensory variables predicted from imitation learning, which is quite analogous to kinematic replay, in Fig 2d, the visual inputs predicted in Fig 4, and the ground contact forces predicted in Extended Data Fig 6.

4.2. Show that imitation learning facilitates reusability of motor skills in goal-driven tasks.

Another major practical capability enabled by this work is the ability to bootstrap policy learning in goal-driven tasks (such as the visually-guided tasks in Fig 4) by reusing the LLC network weights, even if the high-level controller (HLC) needs to be trained from scratch.

Ideally, an architecture would be used that is more amenable to reuse while minimizing the domain shift, such as in CoMic (Merel et al., 2020; cited in the text), but even without adopting advanced techniques, at a minimum the network weights could be reused for conventional weight initialization-based transfer learning.

This could be easily implemented for the bumps and trench tasks, and would presumably make task training converge faster and more stably, as has been shown in previous work adopting this strategy (e.g., Liu et al., Science Robotics, 2022).

Demonstrating these advantages would demonstrate value for the robotics/CS members of the audience as a practical matter/engineering trick for training complex policies based on biological constraints derived from MoCap, as well as increase the significance and impact of the work by providing support for the importance of embodiment in NeuroAI (Zador et al., 2023).

We thank you for this excellent suggestion. We have now replaced the controllers trained from scratch for the visually-guided flight tasks (bumps, trench) with a high-level controller which is trained to reuse the pre-trained low-level controller trained using imitation. And we now release code for the reuse of both the flight and walking controllers. These changes can be seen in Fig 4 and the “Vision-guided flight with flight controller reuse” section.

Minor concerns

5. Reference motion capture data: Walking

Since imitation requires registration of motion capture data to the biomechanical model, standard algorithms for this inverse kinematics step (as used here) require the reference data to be in 3D. It can be difficult to do 3D pose estimation in walking flies owing to their size and the need to multiple camera views for accurate 3D triangulation.

That said, the approach employed here is pretty unusual and introduces the need for extra validation (partially addressed in ED Fig 2). Namely, the authors use a 2D pose-tracked dataset of flies walking and "lift" it to 3D using a hand-crafted model of the fly gait cycle. To do this, they use manually estimated parameters that describe the z-coordinate of the leg tips during the walking cycle from manually annotated videos in a previous paper (Akitake et al., 2015).

This is super weird as: (i) this fails to capture the variability outside of the typical trajectories; (ii) is biased and constrained to walking bouts or idle behavior; (iii) there exist easy-to-use and well-validated methods for doing deep learning-based lifting of 2D poses, including in walking flies specifically (Gosztolai et al, 2021); (iv) there exist plenty of datasets and approaches for 3D pose tracking of walking flies (e.g., Günel et al., 2019; Karashchuk et al., 2021).

A minimal effort improvement to this step could be to apply LiftPose3D (open-source, easy to use in this reviewer's experience, has fly-specific checkpoints) to obtain 3D trajectories, or use one of the existing datasets that were derived from multi-view videos.

Existing 3D datasets of fruit flies are largely inadequate for our purposes. They are either limited to flies unnaturally constrained to walk on a ball, or do not contain a sufficient diversity and quantity of freely walking trajectories. We thank you for the suggestion of using LiftPose3D. However, we discovered that we cannot easily apply it to our data since it needs a different number of keypoints to what we have in our data, and since it was trained on data collected from a ventral view while our data was collected from a dorsal view.

We should add that our approach, which uses inverse kinematics based on our body model to infer 3D from 2D, is both the best possible model-based estimate and another nice use-case of our body model.

6. Reference motion capture data: Flight

I'm a bit confused as to the use of the *Drosophila hydei* datasets from Muijres et al. (2014, 2015). Is it because of the free flight tracking as opposed to simulated flight on a pin?

I would think that there are *D. melanogaster*-specific datasets available (and ample methods to generate this data) available in the below refs:

Tammero and Dickinson (2002)
Fry et al. (2003)
Ristroph et al. (2009)
Straw et al. (2010)
van Breugel and Dickinson (2014)
Ben-Dov and Beatus (2022)
Melis et al. (2024) (Dataset doi: 10.22002/aypcy-ck464)

Is there really a strong reason to use *D. hydei* data for such a central experiment? It would be helpful to elaborate on this choice considering the preponderance of historical studies on free flight in *D. melanogaster*. Switching to using *D. melanogaster* data would also significantly strengthen the evidence for the validity of the current work.

We have a similar situation with the available 3D pose tracking data for flight. Existing datasets, including those cited here, are again either limited to *D. melanogaster* flies constrained to tethered flight, only report a very small number of free flight trajectories, are not public, or do not report body orientation. We note that even Melis et al 2024 used the same *D. hydei* dataset of free flight trajectories to simulate free flight noting:

*It is difficult to make precise one-to-one comparisons with the flight manoeuvres of real flies, because **the most relevant free flight data were collected on a slightly larger species (Drosophila hydei)** and even relatively stereotyped manoeuvres such as spontaneous body saccades vary extensively from event to event [Mujires 2014, Mujires 2015]. — Melis et al 2024*

7. Overstatement of potential short-term capabilities

The authors describe some of the short-term capabilities enabled by this work:

"In the ****short term****, our body model and imitation learning framework can enable the model-based investigation of the neural underpinnings of sensory-motor behaviors such as escape invoked by looming stimuli [Card, 2012], gaze-stabilization [Cruz and Chiappe, 2023], the control of movement by the ventral nerve cord [Lesser et al., 2023, Azevedo et al., 2022, Cheong et al., 2023, Marin et al., 2023, Takemura et al., 2023]." (L391-396)

However, realistically, it seems unlikely that the framework in its current state could actually be employed in investigations across all of these behavioral domains.

We have now removed the phrase “short-term” and more clearly detailed our vision of the next steps for how our model can be used.

As we detail in our response below to Review 2, we believe that our body model can be fruitfully combined with a recent approach [Lappalainen et al 2024], which demonstrates that task-optimized and connectome-constrained network models of the fruit fly visual system accurately predict neural activity with single neuron resolution. The same approach can be taken to build connectome-constrained models of the fruit fly VNC, using imitation learning as the task in order to build models of fruit fly locomotion capable of predicting the role of individual neurons in the VNC. Preliminary work from my lab modeling the male fly song production circuitry in the VNC already shows promise.

For example, escape behavior requires a transition from terrestrial locomotion to flight, punctuated by the complex dynamics involved in take-off. In the current work, there are separate controllers for walking and flight, and effectively different biomechanical models due to the specializations employed in each setting, such as turning off legs during flight. The enhancements that would need to be made to the framework are non-trivial, involving (at least): (i) training combined neural controllers or coming up with an MoE-style switching scheme; (ii) investigating and likely improving the physics models for take-off that combine both ground forces and complex aerodynamics (where turbulence is more important?); (iii) overcoming intractability of high DoFs if all actuators are active or engineering capabilities for mid-simulation toggling of actuators in MuJoCo + dynamically changing the action space in neural network controllers. I'm not even sure what would need to happen with the WBG.

We clarify that the differences in neural network controller architectures including WBG, and degrees of freedom + observation space, between flight and walking behaviors were simply for convenience and not strictly necessary. We demonstrate this point by training a common network architecture with the same model settings for each of the two tasks of walking and flight with similar imitation accuracy. We detail this further in our response to Reviewer 2.

As another example, control of movements by the VNC, such as for the grooming sequence, would require significant additional work to: (i) tweak the body model to angle ranges and actuator DoFs for complex leg movements like mid- and posterior-thorax grooming; (ii) capture the dynamics of the interaction between legs and soft tissues like the wing (currently not modeled as a deformable structure), antennae and aristae, as well as how the adhesion actuators come into play; and (iii) add sensory modalities

relevant to this behavior, such as olfaction or fine-scale mechanosensation (e.g., dust) which are external drivers of the motivation to initiate and engage in grooming.

We agree that grooming will require additional modeling. However, we clarify that joint angle ranges are already optimized based on extreme postures generated during grooming behavior. Further, it is very easy in our software to change joint limits, if needed.

These could be addressed by either removing or requalifying the statement "short term", by more accurately describing the gaps, or by just highlighting specific classes of behaviors that are actually enabled by this work (e.g., locomotion and flight).

I also think that the extent to which "neural underpinnings" are accessible via this model is highly debatable given the limitations outlined in this review and without providing any empirical evidence of predictability of neural correlates (e.g., from simultaneously recorded in vivo activity as in Melis et al and others).

Addressed above.

8. Typos

L125-126: Should "Most important..." be part of the previous sentence or is there a clause missing?

L250: "manuevers" -> "maneuvers"

Thank you. We have corrected these typos.

Nature checklist

****A. Summary of the key results****

Provided in the beginning sections.

****B. Originality and significance: if not novel, please include reference****

Comments on novelty provided all throughout the review. Major concerns are overlap with Melis et al. (2024) and Wang-Chen et al. (2023) – refer to sections at the top of the review for specific comments on these.

****C. Data & methodology: validity of approach, quality of data, quality of presentation****

Key concerns revolve around validity and are elaborated on throughout the review. Quality of data is largely ok, with some concerns about the MoCap reference data described in points (5) and (6).

****D. Appropriate use of statistics and treatment of uncertainties****

Not much of this, but it also is not too applicable for this work.

****E. Conclusions: robustness, validity, reliability****

Discussed inline.

****F. Suggested improvements: experiments, data for possible revision****

Discussed inline.

****G. References: appropriate credit to previous work?****

Undoubtedly, there are missing references given how many fields this work integrates but it seems mostly okay as far as I can tell. It could use an enhanced discussion of recent work already cited in this paper (Melis et al., 2024; Wang-Chen et al., 2023).

****H. Clarity and context: lucidity of abstract/summary, appropriateness of abstract, introduction and conclusions****

Great! Well written and clear :)

Accountability

This review was written by Talmo Pereira <talmo@salk.edu> who opts to forsake anonymity in favor of ethical reviewing through accountability and transparency.

Referee #1 (Remarks on code availability):

The code is great! The authors could do better with the conda environment file (explicitly specify channels, impose some version constraints), but for interested audiences who would actually make use of the code, it's definitely in great shape (until Jax updates of course ;)).

They could document and streamline artifact download (e.g., data and model checkpoints needed to run the examples), but this can be overcome pretty easily by motivated users.

This reviewer was able to install and run the code and examples successfully!

Thank you!

Referee #2 (Remarks to the Author):

A. Summary of key results

The manuscript provides a great tool for future work on the integration of models of the muscular and neural control of sensorimotor behavior in an embodied context, i.e. a biomechanical model of the body of a fruit fly and its interaction with the surrounding physical world. The whole-body model is implemented in the open source MuJoCo (Multi-Joint dynamics with Contact) physics engine and elegantly incorporates both, realistic information on the geometry, weight, and degrees of freedom of 67 body parts as well as interactions of the 67 body parts with each other and interactions with an approximation of the physical environment during locomotion. Based on the assumption of uniform density MoJoCo estimates the body parts moments of inertia, and actuators are added to control joint movements and positions. Interactions with the physical world are approximated by a phenomenological fluid model that generalizes and extends previous work. The body model is then equipped with idealized (non-biological) sensorimotor and visual feedback controllers that can be trained to mimic locomotor movements of real flies. An artificial neural network (ANN) receives MuJoCo simulated sensory input and is trained by reinforcement learning in closed loop to drive the actuators in the body model. The main finding is that the highly flexible artificial neural network can indeed be trained to produce the simulation of locomotor movements that closely resemble basic walking and flight tasks as observed in the biological system.

B. Originality and significance

The model is all done on a high technical level and can be used real-time, the code is well documented and freely available. Additionally, forces like the Kutta lift and the Magnus effect that have not been part of previous models have now been implemented. Additional information on muscles and sensory biology can be incorporated in the future, so that the model is general purpose and extensible. Therefore, in the long run, if smartly paired with other approaches, data, and experiments, this could really help to gain novel insight into the neural control of sensorimotor behavior. However, as it stands the study does neither provide novel insight into the neural control of sensorimotor behavior as the model lacks biological neural components, muscles, material properties, and sensors, nor does it rigorously test the impact of the new forces implemented on locomotion or the limits of the whole-body model. In sum, the generation of the fine

biomechanical model and the accompanying proof of principle that it can be used in a simulation to train an artificial neural network to mimic movements as macroscopically observed during basic aspects of locomotion represent strong methodological advances but little advances to our understanding of the control of locomotion.

C. Data & methodology: validity of approach, quality of data, quality of presentation

The data presented are high quality and the model is finest state of the art. However, as it stands, this approach does not seem to provide key novel biological insight into the neural control of sensorimotor behavior.

1. The finding that an artificial neural network (ANN) can be trained to produce walking- and flight-like locomotor patterns that mimic movements produced by real animals does not mean that it employs similar rules that are used by the biological system. This is somehow implied in the manuscript, but it has not been tested. For example, it is stated that the project models the neural control of sensorimotor behavior, but the artificial neural network is not based on experimental insight on neural connectivity or physiology, and thus remains a black box. Can different types of ANNs learn to produce the flight and walking tasks shown? If yes, this would indicate that multiple strategies/network structures can be used to accomplish the task, but it remains unclear which ones are employed by the brain. See point 1 under suggested experiments (section F).

We believe our general-purpose fly body model, capable of accurately modeling the broadest range of fly behaviors, will serve as the platform of choice for all future studies modeling the generation of fly behavior. To this end, we demonstrated both of our contributions: the body model and the imitation learning framework on locomotor behaviors both on land and in the air. However, investigating the neural basis of locomotion is out of the scope of our work.

However, we believe that our imitation learning framework is key to demonstrating two important results. First, to demonstrate that our physics simulation is sufficiently accurate and that our body model is sufficiently rich in terms of degrees of freedom and actuation as well as its sensors to accurately generate natural locomotion in a sensorimotor closed-loop. Second, to demonstrate a general-purpose machine learning framework enabling the optimization of parametric controllers – an essential

computational tool for future work building data-driven computational models of neural control.

2. There are some concerns that the 'anatomically-detailed biomechanical whole-body model' as presented here may be too much of a simplification to impose realistic requirements to the neural controller, particularly the lack of any stretch-activated elements or other material properties and the fact that the controller needs to only learn deviations to a "nominal wing beat pattern" with no feedback or resonance. The concern is that the ANN can learn to realistically simulate locomotion of a biomechanical whole-body model in a physical environment, but the demands to the neural controller are so different from the real biology that one does not learn much about the nature of the biological neural networks that normally control locomotion. Below are a few examples substantiating this concern. First, it is well known that differential elastic and other mechanical properties of the exoskeleton are highly important to locomotor movements, such as the resonating properties of specific parts of the thoracic cuticle and the wing hinge during flight. During walking, joint stiffness as mediated by material properties and co-contractions of antagonistic muscles are at least as important as the movements of the leg joints, but only the joint movements and positions are actuated. Material properties of the exoskeleton, joints, and tendons are not considered in the biomechanical whole-body model. Second, the properties of the skeletal muscles have not been included. In particular during flight, resonating properties and stretch activation of flight power muscles are extremely important for setting the relationship between NS output and locomotor movement. In the case of stretch activated asynchronous muscles, the NS outputs only every 10 to 40 contractions of the muscles. By contrast, flight steering muscle contractions are related very differently to NS output, in some cases 1:1 between motor neuron firing and muscle contraction. Therefore, wing power and wing steering are coupled to the neural controller in entirely different ways. However, in the simulation presented the neural controller gives output signals to actuators that drive joint movements and positions in the same way for both. Such obvious deviations from the biology of the system are neither acknowledged nor discussed. Third, degrees of freedom and joint position ranges that are used to actuate the wings or the leg joints are based on movements observed during grooming movements. The joint kinematics during grooming might be in a similar working range as those during walking, but for sure not as those during flight. It is clear that not all biological details can be implemented in the body model, but a critical evaluation of the potential benefit of implementing some key additional aspects would be very helpful. This requires additional experiments and a critical discussion of the potential consequences of what is omitted.

We wish to distinguish between **which** forces are required by the body from **how** the forces are generated by the body. For instance, our model only aims to predict total upwards fluid forces generated by realistic movements of the wings in air, and therefore correctly predicts the resulting translation of the body during free flight. However, we do not yet model how the body generates these forces, which have both passive and active components. We agree that the addition of anatomically detailed muscles, stretch receptors, material properties, resonance, etc will enhance the realism of how our body model generates forces. We now discuss the consequences of ignoring these details in our manuscript. We also discuss our vision that our body model can serve as a platform for future studies focused on modeling such components in detail by individual labs with defined expertise in these areas.

However, we wish to clarify two points:

First, the design of our wing-beat pattern generator based controller for free flight is simply one architectural choice to constrain and train such a controller. It is not an essential component of the body model or the imitation learning framework and indeed can be replaced by any other ANN architecture.

Second, the joint limits were set to the outer limits of the possible ranges determined by inverse kinematics to enable all possible behaviors with a common set of joint limits. Indeed during any particular behavior the joint angles explored might be limited to a smaller range inside these limits, but these do not constitute joint limits in our model.

3. What do we really learn about the biology of flight and walking? For example, how much better would the simulation do with more sensors, what is the minimal number of sensors to have adequate learning of the tasks tested? Testing this experimentally could give some information as to which types of sensory feedback are most essential for locomotor control. What is the relative impact of the forces that have not been implemented before (e.g. magnus force, Kutta lift), and most importantly, how complex can the behavior get and still be successfully trained? Real flies do rapid saccades and fast changing, complex flight trajectories. Real flies walk not only on horizontal surfaces, but can climb various slopes, walk upside down on the ceiling, and climb obstacles. What is the most complex locomotor behavior that can be learned by the ANN in this model? Can a neural network controller trained on grooming climb a slope? Testing this would give deeper insight into both the limits and the potential power of the model for future studies.

We thank you for these excellent suggestions. We now report the relative contributions of the fluid forces during free flight in Fig 2. We also now analyze and report the contributions of the adhesion forces during walking on sloped surfaces in Extended Data Fig. 6.

We clarify that we did indeed successfully train our steerable locomotion controllers to produce the most complex locomotion behaviors. We trained the steerable flight controller to imitate extreme flight trajectories involving saccades and rapid evasive maneuvers. We trained the steerable walking controller to imitate the near complete repertoire of complex walking trajectories generated during free walking behavior in a social context. Further, we demonstrated via inverse-kinematics that it is possible to match grooming behaviors as well. While we agree that our demonstration does not yet cover the complete range of fly behaviors, it is nonetheless the most comprehensive demonstration to date in a single model.

D. Appropriate use of statistics and treatment of uncertainties

The statistics used to make the model seem state of the art. However, as in any model, not all biology can be incorporated, Therefore, remaining uncertainties are the nature of the approach and must be accepted. However, these should be spelled out more clearly.

1. More detailed information on how well the model sensory system matches the real sensory system would be highly useful. Table 20 lists sensors known and studied in real flies and next to this, the corresponding model sensors. However, it remains unclear how well the model sensors represent the measurements taken by the biological sensors. How well does the model haltere gyroscope represent the measurements of Coriolis forces, balance, and equilibrium that are so far known to be provided by real halteres? It is clear that the biological sensors are far from completely understood and that the model can and does unlikely need to fully copy the biological sensors, but some assessment of the mechanosensory experiences of the model would be very useful.

We should clarify that our goal for this version of the body model was to build sensors representing **what** quantities are sensed by the fruit fly body, but we do not claim to faithfully model **how** the sensors report these quantities. Therefore these sensors should be considered idealized sensors, which we now further clarify in the manuscript. As is the case with modeling muscle actuation, faithfully modeling the ways in which the sensory system functions is left for future work. We do however provide an exhaustive literature survey in Table 20, which we now also discuss in the manuscript.

2. Related to the above, during specific tasks sensors can be switched off and on, and their properties can be modified. Do some tasks require switching off certain sensors, such as switching off resistance reflexes during locomotion in real insects? Has this been systematically analyzed (not just recorded)? How important was changing sensor properties for which task, and how well do these changes reflect known changes of the properties of sensors in biological systems? Again, it is clear that not all of this can be addressed, but some assessment of how important and how realistic task specific changes in sensor properties and availability would help judging how realistic and how general purpose the model sensory system is.

3. It is mentioned that some actuators may be switched off in a task specific manner, such as leg actuators during flight tasks. Is it important to switch these off to maintain stable flight, e.g. because feedback to leg actuators would produce movements that would interfere with the flight task? If yes, it would be important to see systematic analyses of which actuators are switched off during which task. It would also be important to learn how joints and appendages are logged into position during specific tasks in biological systems, and how that relates to switching off actuators in the body model.

We agree that these questions regarding the necessity and sufficiency of sensors and actuators for particular behaviors are exciting directions which can be addressed with our modeling framework. We now discuss these questions in the manuscript.

E. Conclusions: robustness, validity, reliability

1. The paper states that generality of the model is demonstrated by realistically simulating two forms of locomotion, flight in air and walking on land. However, the controllers seem to have a quite different design (wing beat generator plus trainable multi-layer perceptron for flight, versus no pattern generator but only a single multi-layer perceptron policy network for walking). Together with the large differences in sensory input for both tasks and the fact that the legs are fixed into one position during flight with disabled actuators and the wings for walking, models for flight and walking seem quite different. Consequently, the statement on model generality should be put into perspective accordingly.

Thank you for raising this concern. We should clarify that these modifications we made to the sensory inputs and motor actuation for walking and for flight were simply for convenience to simplify the reinforcement learning problem and not strictly necessary. We now demonstrate that a single common model setting with all the sensors and actuators enabled can accurately reproduce both flight and walking.

Figure. Comparison between the performance of the original fly model and the modified model (all DoFs enabled, torque actuators replacing position actuators) trained on flight (top row) and walking (bottom row) imitation tasks. (a, b) Percentiles of CoM and orientation quaternion tracking errors in the flight imitation task. Solid lines: original model, dotted lines: modified model. (c) Flight imitation learning curve comparison: average episode returns vs actor steps during training. (d, e, f) Same as (a, b, c) but for walking imitation. In (f), an additional reward term is required to keep the (now enabled) wing DoFs in folded position and it causes most of the discrepancy between the two learning curves. This reward term is only approximately satisfied, causing a reduction of the episode return by a factor of ~ 0.69 in the trained model.

2. The opening statement that the body determines how the nervous system controls behavior (abstract, first sentence) does not quite capture the core of the study. The way the NS controls behavior is indeed influenced by the geometries, weights, and movements of the body parts (which has been considered here) as well as their interaction with the physical environment (much of which has been considered here), but also by the material properties of the body parts, the properties and dimensions of the muscles, metabolic state, and numerous other parameters (which have not been considered here). Therefore, the promise that the embodied context provided by this study is key to understanding how the NS produces behavior seems an overstatement. It is one of many building blocks required to address this question in sufficient detail.

We agree. We have now re-worded this statement.

3. The discussion lists multiple ways the model can be build on in the future, but this list includes mainly anatomical additions, such as XNH X-ray imaging and confocal imaging of body parts to add further detail to the 67 rigid parts body model, recent mapping of proprioceptors to include more realistic positions of the sensors, or pose-tracking algorithms to retrieve more detailed kinematic information from high-speed video of flies. This may all help to further improve the existing model, but it will not bring an entirely different quality to the model. Incorporating the physiology of the neuromuscular and sensory systems would indeed cross a milestone.

We agree that the future model suggested in this comment is indeed exciting. We can spell out two steps to get there. First, we must map the **geometry (anatomy)** of the actuation and sensory systems, which reveals which joints and body parts are coupled by which musculoskeletal components. However, these anatomical measurements can be made post-mortem using high-resolution X-ray and confocal imaging. Second, we must model the **dynamics (physiology)** of these systems which determines, for instance, how quantitatively muscle activation translates to force production. While it is possible that pose-tracking combined with imitation learning can help constrain these parameters somewhat, ideal would be correlated measurements of kinematics and sensory neuron + muscle activity throughout the body during behavior. This feat was recently performed by Melis et al 2024 for the wing hinge muscle system.

4. Although the ms claims that future work can leverage connectomic mapping of the fly nervous system to model the neural circuits that control locomotion more realistically, the avenue to get there is only partially sketched. The artificial network remains a black box and would have to be entirely replaced by a connectome- and physiology constrained neural network model. How realistic is this? And how to make the biological neural network control the biomechanical body model? At least in biology, the fly neural controller does not need to learn to use the body for locomotion through the physical world. Upon using biological neural networks, is there still training envisioned, and if yes, what kind of training?

We now further clarify this point in the manuscript. Lappalainen et al 2024 recently demonstrated that task and connectome constrained network models of the fly visual system can combine measurements of connectivity of a circuit with knowledge of the input-output function of the circuit in order to estimate the unknown dynamical parameters of individual neurons and synapses resulting in accurate predictions of the neural activity of individual neurons in the circuit. The goal of the task-optimization training is not to emulate any learning process in the animal, or even to estimate the weights of the network (which are constrained by measurement of the connectome), but

instead to estimate the dynamical properties of single neurons (time constant and firing threshold) and synapses (unitary synapse strength).

We can imagine a similar strategy here. We can use the imitation learning task to train a connectome-constrained network model of the fly VNC to enable the model fly to walk and fly in exactly the same way that the real fly does. As in Lappalainen et al 2024, the parameters estimated would not be the weights of the network but instead dynamics of single neurons and synapses. The success of this approach will depend on training the model to accurately reproduce the broadest diversity of fly behaviors with the VNC network whose weights are strongly constrained by the connectome. For this reason, we believe that imitation learning will be an important component of how our model will be used to build models of the nervous system.

We now spell this out in the discussion.

5. Often the text suggests that the model resembles the biology of the nervous system. For example, the training of ‘steerable’ low level controllers that are used to generalize from training trajectories to locomotion along novel trajectories are classified as analogous to the ventral nerve cord in that they are responsible for translating descending command signals to low level motor control signals (lines 160-165). It is correct that the biological system for sure uses descending commands for the decision to fly somewhere, and that the descending input is integrated with other information in ventral nerve cord networks. However, this does not suffice to state that the model low level controllers are analogous to the ventral nerve cord. It is not entirely clear whether all joint movements are driven by low level command signals, and most importantly, it is not clear whether VNC networks can be trained. Therefore, it is suggested to remove suggestive statements on possible analogies between the ANN and the biological networks (which have not been studied here) throughout the text. The study does not gain much from these statements unless they are further substantiated.

Thank you for pointing this out. We have targeted our model and paper to a broader community beyond fruit fly neuroscience, including computer science, engineering, biomechanics. For this reason, we have attempted a high-level presentation of the biology – we have now refined these statements.

6. The discussion repeats what has been done and lists possible future directions that can build upon the model. However, it does not refer to any novel insight into the biology of animal locomotion. Any important novel insight on the sensorimotor control of locomotion that this study has provided must be explained in the discussion.

F. Suggested improvements: experiments, data for revisions

1. A first step toward identifying rules used by the black box ANN would be to compare different types of artificial neural networks in their ability to be trained for generating locomotor patterns. This could potentially help identifying specific network motives that are particularly important for successful ANN training. It would require thorough analyses of different types of ANNs trained to control the whole-body model to then distinguish network design principles that are particularly useful from network motives that are less important.

2. As mentioned above (section C3) additional experiments could give valuable information about the biology of locomotion control. First, manipulating the numbers of sensors required for adequate task learning. Second, quantifying the relative impact of the forces neglected in previous models (e.g. magnus force, Kutta lift). Third, testing, how complex the behavioral task can get to still be successfully trained, e.g. rapid saccades, climbing different slopes and obstacles.

Thank you for these suggestions. We have now conducted new experiments and analyses as discussed above. However, we consider the analysis of ANN architectures to be out of scope for this work, which focused on the substantial effort of developing and validating a general-purpose body model and imitation learning.

G. References: appropriate credit to previous work?

1. The importance of Kutta and Magnus forces has been discussed by Sane (2003, JEB, 206 (23): 4191–4208). Can the importances of the inclusion of these forces be assessed?

Thank you for this suggestion. We now plot the contributions of each component of our phenomenological fluid model during free flight in Fig 2j. We find that just two components – viscous drag and Kutta lift – dominate and are 1-2 orders of magnitude larger than the other components.

2. The idea to use reduced approximate fluid dynamics simulations that are fast enough for repetitive evaluation in optimization procedures on the flight system has been proposed in Berman and Wang (2007, J. Fluid Mech 582:153-168). How does your phenomenological fluid model based on Anderson (2007) relate to what has been proposed by Berman and Wang?

Addressed above.

3. Reid et al (2019) argues that wing flexibility plays a role in energetically optimal flight. This is not part of the proposed, posture-based biomechanical model. Some reference to wing flexibility and other material property-based mechanisms should be made and the expected consequences of omitting these known factors should be discussed.

Addressed above.

4. The discussion should include some reference to other biological systems in which whole-body simulations are performed, such as worms and jellyfish.

Thank you for this suggestion. We now include these references to work by Wang,...,Fairhall PNAS 2023 on the hydra and by Boyle,...,Cohen FCN 2012 on C elegans.

5. Previous models often focused on one movement modality, for example Lobato-Rios for walking or Dickson 2008 for flight. The reason is that the two modes are very distinct in their kinematics and neural control. What is the exact merit of having all in one model and does this come with limitations? In particular, if the ranges and angles for flight and walking are both taken from a third motor behavior, grooming?

Addressed above.

Having a single general-purpose body model can enable a unified understanding of how wing and leg circuitry work together to coordinate the rich diversity of fly behaviors including walking, flight, courtship, and grooming. Looking forward towards modeling the neural circuitry involved in sensory motor control, the same neural circuitry is often involved in multiple behaviors, for instance walking and grooming both involve leg control circuitry, flight and male courtship song production both involve wing control circuitry, male courtship involves coordinating wings and legs in order to pursue the female during song production.

We also addressed this concern by training walking and flight controllers using the same body model without freezing any degrees of freedom. This demonstrates that modality specific modeling choices are not necessary, especially to model behaviors such as the transition from walking to flight, or the use of wings during courtship behavior which occurs during walking on the ground. Indeed the same set of joint limits are used for all behaviors.

H. Clarity of content

The ms is clearly written and well presented. As mentioned above, similarities and differences between the model and biology should be pointed out more clearly throughout the ms.

Thank you!

Referee #2 (Remarks on code availability):

The code is a highly valuable and usable resource for the community. It is done highly professionally and indeed all open-source. We were able to install it, and we ran the "getting- started script" without any problems.

Thank you!